# Triiodothyronine (T3) promotes brown fat hyperplasia via thyroid hormone receptor α mediated adipocyte progenitor cell proliferation

Shengnan Liu[1,12], Siyi Shen[1,12], Ying Yan[1], Chao Sun[1], Zhiqiang Lu[2,3], Hua Feng[4], Yiruo Ma[1], Zhili Tang[1], Jing Yu[1], Yuting Wu[1], Balázs Gereben[5], Petra Mohácsik[5], Csaba Fekete[6], Xiaoyun Feng[7], Feixiang Yuan [1], Feifan Guo [1], Cheng Hu[8], Mengle Shao [9], Xin Gao[2,3], Lin Zhao[2,3], Yuying Li[1], Jingjing Jiang [2,3✉] & Hao Ying [1,10,11✉]

The thyroid hormone (TH)-controlled recruitment process of brown adipose tissue (BAT) is not fully understood. Here, we show that long-term treatment of T3, the active form of TH, increases the recruitment of thermogenic capacity in interscapular BAT of male mice through hyperplasia by promoting the TH receptor α-mediated adipocyte progenitor cell proliferation. Our single-cell analysis reveals the heterogeneous nature and hierarchical trajectory within adipocyte progenitor cells of interscapular BAT. Further analyses suggest that T3 facilitates cell state transition from a more stem-like state towards a more committed adipogenic state and promotes cell cycle progression towards a mitotic state in adipocyte progenitor cells, through mechanisms involving the action of Myc on glycolysis. Our findings elucidate the mechanisms underlying the TH action in adipocyte progenitors residing in BAT and provide a framework for better understanding of the TH effects on hyperplastic growth and adaptive thermogenesis in BAT depot at a single-cell level.

[1] CAS Key Laboratory of Nutrition, Metabolism and Food Safety, Shanghai Institute of Nutrition and Health, University of Chinese Academy of Sciences, Chinese Academy of Sciences, and Shanghai Jiao Tong University Affiliated Sixth People's Hospital, Shanghai, China. [2] Department of Endocrinology and Metabolism, Zhongshan Hospital, Fudan University, Shanghai 200032, China. [3] Fudan Institute for Metabolic Diseases, Fudan University, Shanghai, China. [4] Omics Core, Bio-Med Big Data Center, Shanghai Institute of Nutrition and Health, Chinese Academy of Sciences, Shanghai 200031, China. [5] Laboratory of Molecular Cell Metabolism, Institute of Experimental Medicine, Budapest 1083, Hungary. [6] Laboratory of Integrative Neuroendocrinology, Institute of Experimental Medicine, Budapest 1083, Hungary. [7] Department of Endocrinology and Metabolism, Shanghai General Hospital, School of medicine, Shanghai Jiaotong University, Shanghai 200080, China. [8] Shanghai Diabetes Institute, Shanghai Key Laboratory of Diabetes Mellitus, Shanghai Clinical Centre for Diabetes, Shanghai Jiao Tong University Affiliated Sixth People's Hospital, Shanghai 200233, China. [9] The Center for Microbes, Development and Health, Institute Pasteur of Shanghai, Chinese Academy of Sciences, Shanghai 200031, China. [10] Innovation Center for Intervention of Chronic Disease and Promotion of Health, Shanghai, China. [11] Key Laboratory of Food Safety Risk Assessment, Ministry of Health, Beijing 100021, China. [12] These authors contributed equally: Shengnan Liu, Siyi Shen. ✉email: jiang.jingjing@zs-hospital.sh.cn; yinghao@sibs.ac.cn

Obesity increases the risk of metabolic diseases and is a lifestyle-related risk factor for premature death. Unfortunately, interventions aimed at decreasing calorie intake and increasing energy expenditure usually fail in the long term[1]. Innovative strategies are therefore required to combat obesity. Brown adipose tissue (BAT) is specialized for thermoregulation and energy expenditure, which contains numerous mitochondria and is capable to dissipate stored energy through the action of uncoupling protein 1 (UCP1)[2]. Given that metabolically active BAT has been detected in adult humans and its function is impaired in obesity, enhancing the activity of BAT has been considered as a promising strategy for the prevention and treatment of obesity and its associated metabolic diseases[3–6]. Thus, better understanding the molecular mechanisms underlying the BAT development and physiology is of both broad biological interest and high clinical relevance.

About 50 g of active BAT may account for a 20% daily increase in energy expenditure in human, suggesting that BAT serves as a natural defense against cold and obesity[7]. As a plastic tissue, BAT undergoes expansion via hypertrophic (increase in cell size) and/or hyperplasic (increase in cell number) growth in thermogenic responses to cold and diet[8,9]. Since BAT mass is normally associated with its capacity, relatively small increases in BAT mass are likely to result in big increases in energy dissipation. When UCP1 levels are saturated in fully differentiated brown adipocytes, increase in thermogenic capacity eventually results from acquisition of more brown adipocytes by promoting the proliferation and subsequent differentiation of adipocyte progenitor cells (APCs) within BAT depots[8,9]. As a fundamental mechanism of adaptation to cold and hyperphagia, the physiological control of BAT hyperplasia has attracted great attention recently.

As a thermogenic hormone, thyroid hormone (TH) is necessary for a full metabolic response of BAT under maximal demands[10–13]. Apart from its role in regulating thermogenic capacity of mature brown adipocytes either directly or via sympathetic nervous system, TH can enhance the hyperplastic growth of interscapular BAT (iBAT), a classic brown fat depot[14]. However, the underlying mechanisms, especially the TH effects on the adipogenic commitment trajectory and the mitotic proliferation of APCs in iBAT depot, are far from being fully understood. Recently, our understanding of the heterogeneity within the APCs has been greatly advanced by single-cell analysis in white adipose tissue (WAT) depot[15–18]. However, whether the APCs in BAT also exhibit heterogeneity and whether TH can regulate cellular heterogeneity and hierarchy within APCs of iBAT, hence modulating thermogenic capacity, remain largely unknown.

Here, we demonstrate that long-term treatment of triiodothyronine (T3), the active form of TH, could increase the recruitment of thermogenic capacity in iBAT depot of mice through tissue expansion by promoting APC proliferation. Mechanistically, thyroid hormone receptor α (TRα) is the major TR isoform mediating the T3 effect on APC proliferation in iBAT depot. Moreover, our single-cell RNA sequencing (scRNA-seq) analysis revealed the heterogeneous nature and hierarchical trajectory of APCs in iBAT. Further analyses suggest that T3 promotes the cell state transition and cell cycle progression via c-Myc (Myc) in APCs of iBAT. Thus, our results establish T3 as a regulator of APCs in iBAT, providing insights into TH-induced recruitment of thermogenic capacity.

## Results

**T3 boosts the thermogenic capacity by promoting iBAT expansion.** To understand the profound effect of TH on the recruitment process of iBAT, mice were rendered hypothyroid by using MMI (MMI mice) followed by intraperitoneal (i.p.) T3 treatment for 4 h (short-term) or 5 days (long-term) as described in Materials and Methods (Fig. 1a and Supplementary Note 1). We found that long-term treatment of T3 increased not only the dorsal skin temperature ($T_{skin}$) in the interscapular area, but also the core body temperature ($T_{core}$) as assessed via rectal thermometry ($T_{rectal}$) of MMI mice, indicating that long-term T3 treatment is able to enhance the thermogenic capacity of iBAT (Fig. 1b, c). Although significant changes in interscapular $T_{skin}$ and $T_{core}$ were not observed after short-term T3 treatment (Fig. 1b, c), the analysis of UCP1 recruitment in iBAT depot revealed that short-term T3 treatment increased the levels of either UCP1 protein per mg tissue protein or UCP1 protein per depot in these mice (Fig. 1d, e and Supplementary Fig. 1b, c), which might be attributed to the upregulation of Ucp1 mRNA levels by T3 (Supplementary Fig. 1d). These data also indicate that the acute elevation of UCP1 protein levels after short-term T3 treatment, though probably effective, might not be high or timely enough to affect the $T_{rectal}$ and interscapular $T_{skin}$ under current experimental condition.

Interestingly, although decreased rather than increased Ucp1 mRNA levels were observed after long-term T3 treatment (Supplementary Fig. 1d), UCP1 protein levels per mg tissue protein were similarly increased after long-term T3 treatment (Fig. 1d and Supplementary Fig. 1b, c). Notably, although the UCP1 protein levels per mg tissue protein were not further increased by long-term T3 treatment compared to short-term treatment (Fig. 1d), the total amount of UCP1 protein per depot was further increased by long-term T3 treatment (Fig. 1e). As T3 treatment had no effect on the stability of UCP1 protein (Supplementary Fig. 1e), the above results imply that there were more UCP1-expressing cells after long-term T3 treatment. These results also suggest that the sustained elevation of the total amount of UCP1 protein per depot might be responsible for the increase in thermogenic capacity of iBAT observed by surface infrared thermography after long-term T3 treatment.

In accordance with the greatly increased thermogenic capacity, the iBAT depot was enlarged after long-term T3 treatment in MMI mice, as evident from increased size and weight of iBAT (Fig. 1f, Supplementary Fig. 1f and Supplementary Note 1). Importantly, long-term treatment of T3 increased the number of both stromal vascular fraction (SVF) cells and mature adipocytes from the iBAT depot (Fig. 1g, Supplementary Fig. 1g and Supplementary Note 1), indicating that iBAT expands extensively through hyperplasic growth during T3-induced recruitment of thermogenic capacity. In contrast, the number of either SVF cells or mature adipocytes was not significantly increased after short-term T3 treatment (Fig. 1g and Supplementary Fig. 1g). Consistently, the decreases in iBAT size, weight, and number of SVF cells were observed in the iBAT of hypothyroid MMI mice compared to euthyroid control mice, which was accompanied with decreases in either interscapular $T_{skin}$ or $T_{core}$ (Supplementary Fig. 1i–l). Above results suggest that T3 might be of physiological importance in modulating the thermogenic capacity through promoting the hyperplastic growth of iBAT.

**T3 expands the adipocyte progenitor population in iBAT depot.** Given that APCs residing in fat depot are able to undergo mitotic proliferation and adipogenic differentiation, thereby generating a functional adipose depot in vivo[19], we speculated that the enlargement of iBAT depot by long-term T3 treatment involves the proliferation of APCs. To test this hypothesis, we performed flow cytometry and fluorescence-activated cell sorting (FACS) analysis by using the SVF derived from the iBAT depot (Supplementary Fig. 2a)[20]. We found that T3 treatment in MMI mice could increase both the number of APCs (CD45$^-$CD31$^-$Sca1$^+$ cells) and the

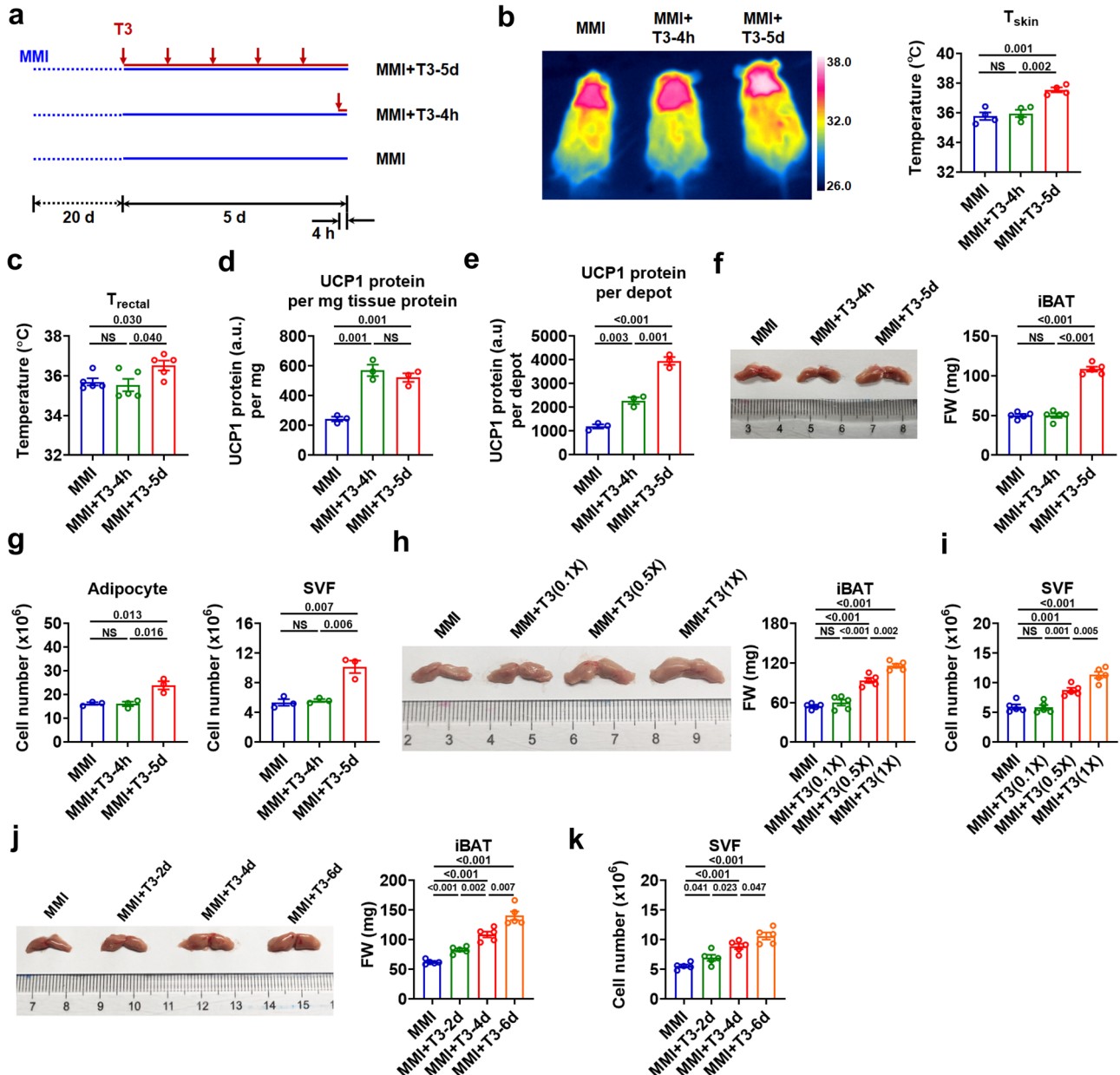

**Fig. 1 T3 increases thermogenic capacity by promoting iBAT expansion. a** Schematic of short-term (MMI + T3-4h) and long-term (MMI + T3-5d) T3 treatments of hypothyroid (MMI) mice. **b** Representative infrared images (left) and quantification of average dorsal skin temperature ($T_{skin}$) (right) of MMI mice and MMI mice with different T3 treatments ($n = 4$). **c** Rectal temperature ($T_{rectal}$) of MMI mice and MMI mice with different T3 treatments ($n = 5$). **d**, **e** UCP1 protein amount per mg iBAT protein (**d**) and total UCP1 protein amount per iBAT depot (**e**) in MMI mice and MMI mice with different T3 treatments ($n = 3$). **f** Representative photograph (left) and fat weight (FW) (right) of iBAT depot from MMI mice and MMI mice with different T3 treatments ($n = 5$). **g** Number of mature adipocytes (left) and stromal vascular fraction (SVF) cells (right) of iBAT per mouse in MMI mice and MMI mice with different T3 treatments ($n = 3$). iBAT was minced and then digested for 40 min. **h** Representative photograph (left) and FW (right) of iBAT depot from MMI mice and MMI mice receiving daily T3 injection for 4 d at indicated doses ($n = 5$). **i** Number of SVF cells of iBAT per mouse in MMI mice and MMI mice receiving daily T3 injection for 4 d at indicated doses ($n = 5$). **j** Representative photograph (left) and FW (right) of iBAT depot from MMI mice and MMI mice receiving daily T3 injection for indicated days at 0.5X standard dose ($n = 5$). **k** Number of SVF cells of iBAT per mouse in MMI mice and MMI mice receiving daily T3 injection for indicated days at 0.5X standard dose ($n = 5$). Means ± SEM are shown. *P* values were calculated by two-tailed unpaired Student's *t* test for **b-k**. NS, not significant. Source data are provided as a Source Data file.

percentage of APCs in either total live cells or CD45− non-hematopoietic (Lin−) SVF cells in a dose- and time-dependent manner (Fig. 2a–f and Supplementary Fig. 2b). Notably, the number and percentage of CD45+ cells in the stromal compartment were increased or tended to be increased dose- and time-dependently by T3 treatment, suggesting that immune cells are also likely involved in the T3-induced iBAT expansion

(Supplementary Fig. 2c–f). Consistent with above results, decreases in the number and percentage of APCs were observed in the iBAT of hypothyroid mice compared to euthyroid mice, suggesting that T3 might be of physiological importance in maintaining the APC pool in iBAT depot (Supplementary Fig. 2g).

Furthermore, 5-bromo-2-deoxyuridine (BrdU) or 5-ethynyl-2-deoxyuridine (EdU) label retention was employed to identify

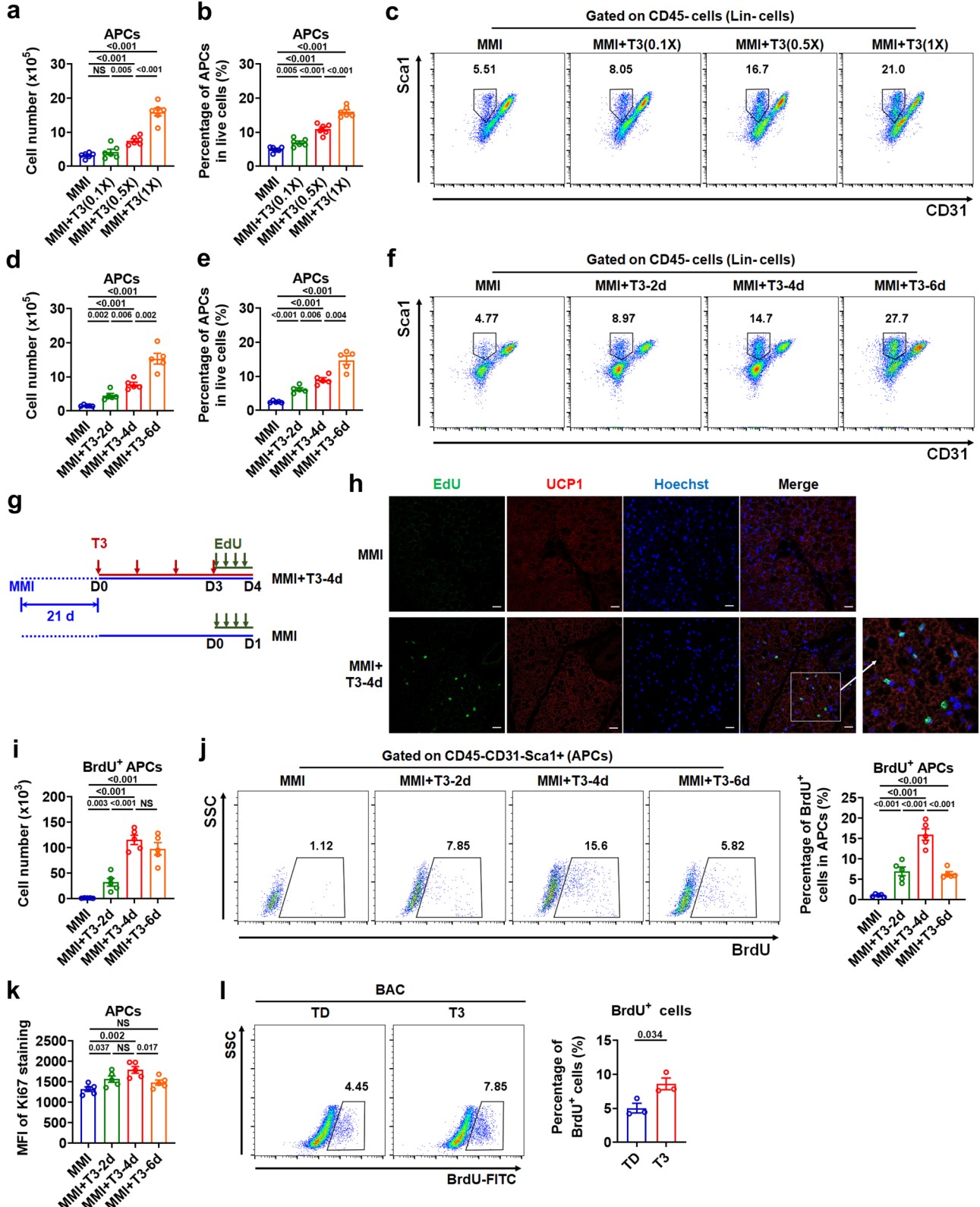

newly generated cells in the iBAT of MMI mice after T3 treatment (Fig. 2g). Fluorescence microscopy analysis revealed that EdU[+] cells could be observed after T3 treatment but barely detectable in the iBAT of MMI mice without T3 treatment (Fig. 2h). Analysis of in vivo BrdU labeling by flow cytometry revealed a gradual increase in both the number and the percentage of BrdU[+] APCs after daily T3 injections in MMI

mice (Fig. 2i, j and Supplementary Fig. 2h). Notably, the number of BrdU[+] APCs reached the highest point when the percentage of BrdU[+] APCs reached a peak at day 4 after T3 injections in MMI mice. Accordingly, staining intensity of proliferation marker Ki67 in APCs was increased gradually and reached the highest point after 4 days of T3 treatment (Fig. 2k). These results indicate that T3 induces the proliferation of APCs to expand the progenitor

**Fig. 2 T3 expands adipocyte progenitor population in iBAT depot. a–f** Number of adipocyte progenitor cells (APCs) in iBAT per mouse (**a**, **d**) and percentage of APCs in live SVF cells of iBAT (**b**, **e**) in MMI mice and MMI mice with daily T3 injection for 4 d at indicated doses ($n = 6$) or for indicated days at 0.5X standard dose ($n = 5$). FACS plot of the APCs (**c**, **f**). Cells were pre-gated on live CD45⁻ cells. **g** Schematic of EdU labeling experiment. **h** Immunofluorescence staining of UCP1 and Hoechst with EdU labeling on iBAT sections. Scale bar, 20 μm. **i–k** Number of BrdU⁺ APCs in iBAT per mouse (**i**), FACS plot and quantification of BrdU⁺ APCs in iBAT (**j**), and mean fluorescence intensity (MFI) of Ki67 staining in APCs of iBAT (**k**) in MMI mice and MMI mice with daily T3 injection for indicated days at 0.5X standard dose ($n = 5$). Cells were pre-gated on live CD45⁻CD31⁻Sca1⁺ cells. **l** BrdU incorporation analysis of brown adipocyte precursor cell line (BAC) cultured in TH-deficient (TD) medium after T3 treatment ($n = 3$). Means ± SEM are shown. *P* values were calculated by two-tailed unpaired Student's *t* test for **a**, **b**, **d**, **e**, **i-l**. NS, not significant. Source data are provided as a Source Data file.

population in iBAT depot. Consistently, increases in the percentage of BrdU⁺ cells and the Ki67 staining intensity were observed in a T3-treated brown adipocyte precursor cell line (BAC) (Fig. 2l and Supplementary Fig. 2i). These data and those from the experiments performed at 30°C, using adrenergic receptor antagonists, with intracerebroventricular injection of T3, and using microneedles for local delivery of T3 to iBAT region (Supplementary Fig. 3a–i and Supplementary Note 2) suggest that T3 can target APCs resided in iBAT directly and promoting APC proliferation in a cell-autonomous manner.

**Deletion of TRα with Myf5-Cre attenuates the T3 effect on hyperplasia of iBAT**. TRα and TRβ mediate most actions of T3, which are expressed in a tissue-specific and developmentally-regulated manner. To determine which TR isoform is responsible for the T3 effect on the hyperplastic growth of iBAT, especially the APC proliferation, the expression of TRα and TRβ in the cultured SVF cells, APCs derived from iBAT, as well as SVF-derived mature brown adipocytes was examined. We found that the mRNA levels of TRα are 10 times higher than those of TRβ in either freshly isolated APCs or cultured APC-enriched SVF cells (Fig. 3a), suggesting that TRα might be the major TR isoform mediating the T3 effect on APCs in iBAT. To investigate the role of TRα in the recruitment process of iBAT, we generated mice lacking the TRα in Myf5-lineage (MTRαKO mice) by crossing TRα flox/flox mice (TRαFloxed mice) with Myf5-Cre mice (Fig. 3b). In agreement with our current knowledge of Myf5-Cre, TRα mRNA levels in both iBAT and skeletal muscle were markedly decreased, but not altered in WATs (Fig. 3c and Supplementary Fig. 4a, b). To evaluate the efficiency of the Cre-mediated recombination in APCs, we took advantage of tdTomato reporter mice and found that Myf5-Cre led to an efficient recombination in APCs of iBAT (Supplementary Fig. 4c, d).

MTRαKO mice displayed normal body weight and muscle weight compared to TRαFloxed mice (Supplementary Fig. 4e, f). As expected, a slight but significant decrease in the interscapular $T_{skin}$ was observed in MTRαKO mice compared to TRαFloxed mice (Fig. 3d, e). It is worth noting that, after 5 days of T3 i.p. injection, the interscapular $T_{skin}$ was also increased in MTRαKO mice but to a lesser extent than that in TRαFloxed mice (Fig. 3d, e). These results suggest that TRα in Myf5-lineage is required for the regulatory effect of T3 on the thermogenic capacity of iBAT. In accordance with the defect in the thermoregulation by TH, we observed decreases in size and weight of iBAT in MTRαKO mice compared to TRαFloxed mice either with or without T3 treatment (Fig. 3f and Supplementary Fig. 4g).

Accordingly, the number and percentage of APCs, as well as the Ki67 staining intensity of APCs were all decreased in the iBAT of MTRαKO mice compared to TRαFloxed mice (Fig. 3g–i and Supplementary Fig. 4h). After i.p. injection of T3, the number of APCs, the percentage of APCs, and the Ki67 staining intensity of APCs were all slightly increased in MTRαKO mice but to a much lesser extent than those in TRαFloxed mice (Fig. 3g–i and Supplementary Fig. 4h), suggesting that deletion of TRα in Myf5-lineage contributes to the blunted response of APCs to T3.

Consistently, BrdU incorporation assay confirmed that the percentage of newly generated APCs was markedly reduced in the iBAT of MTRαKO mice after i.p. injection of T3 (Fig. 3j). Collectively, these data indicate that the TRα in Myf5-lineage mediates the T3 effect on iBAT hyperplasia. In addition, as treatment of β3-AR antagonist totally diminished the effect of i.p. injection of T3 on APC proliferation in the iBAT of MTRαKO mice, we speculated that β3-AR mediated adrenergic activation is indispensable for the preserved T3 action in MTRαKO mice (Supplementary Fig. 4i). These results also indicate that β3 adrenergic signaling involved here is independent of the TRα-mediated direct action of T3 on APC proliferation in the iBAT depot.

The effect of TRα on the proliferative capacity of adipocyte progenitors in iBAT was then evaluated in vitro by using APC-enriched SVF from the iBAT depot. Flow cytometry analysis revealed that about 90% of cultured SVF cells were CD45⁻CD31⁻Sca1⁺ cells (APCs) (Supplementary Fig. 4j). As expected, the expression of TRα was very low in cultured SVF cells dissociated from the iBAT of MTRαKO mice (Supplementary Fig. 4k). These iBAT-derived SVF cells from MTRαKO mice grew much slower than control cells derived from TRαFloxed mice (Supplementary Fig. 4l). Accordingly, the Ki67 staining intensity of SVF cells and the percentage of BrdU⁺ SVF cells were both dramatically decreased in SVF cells derived from the iBAT of MTRαKO mice compared to control cells (Supplementary Fig. 4m, n).

Additionally, the results obtained in mice without TRβ in Myf5-lineage (MTRβKO mice), mice with mature brown adipocyte-specific TRα knockout (UTRαKO mice), and mice treated with GC-1, a TRβ-selective agonist, suggest that TRβ in Myf5-lineage is dispensable for T3 induced-hyperplasia of iBAT (Fig. 4a–g, Supplementary Fig. 5a–h, and Supplementary Note 3), while TRα in brown progenitors but not TRα in mature brown adipocytes predominantly mediates the T3 effect on the proliferative capacity of APCs in iBAT (Fig. 4h–l, Supplementary Fig. 5i–k, and Supplementary Note 3).

**ScRNA-seq analysis reveals the heterogeneity of APCs in iBAT depot.** Given that adipocyte progenitors are a functionally heterogeneous cell population[15–18], to explore the role of T3 in APCs of iBAT, we performed scRNA-seq using MMI mice with and without long-term T3 treatment (Supplementary Note 4). Graph-based clustering identified the presence of 11 distinct clusters, each containing unique marker gene profiles (Fig. 5a, Supplementary Data 1 and Supplementary Fig. 6a). To query the similarity of subpopulations, we performed unsupervised hierarchical clustering of gene signatures to explore the relatedness among the clusters and then identified 5 groups of clusters (Fig. 5b, c). Further analysis revealed most differentially expressed genes (DEGs) among different groups (Fig. 5d–j, Supplementary Fig. 6b and Supplementary Data 2 and Supplementary Note 4). Group 1 cells expressed high levels of *Pi16* and *Dpp4*, but did not exhibit expression of adipocyte markers, indicating that cells in Group 1 are more primitive undifferentiated APCs. Group 2 cells were

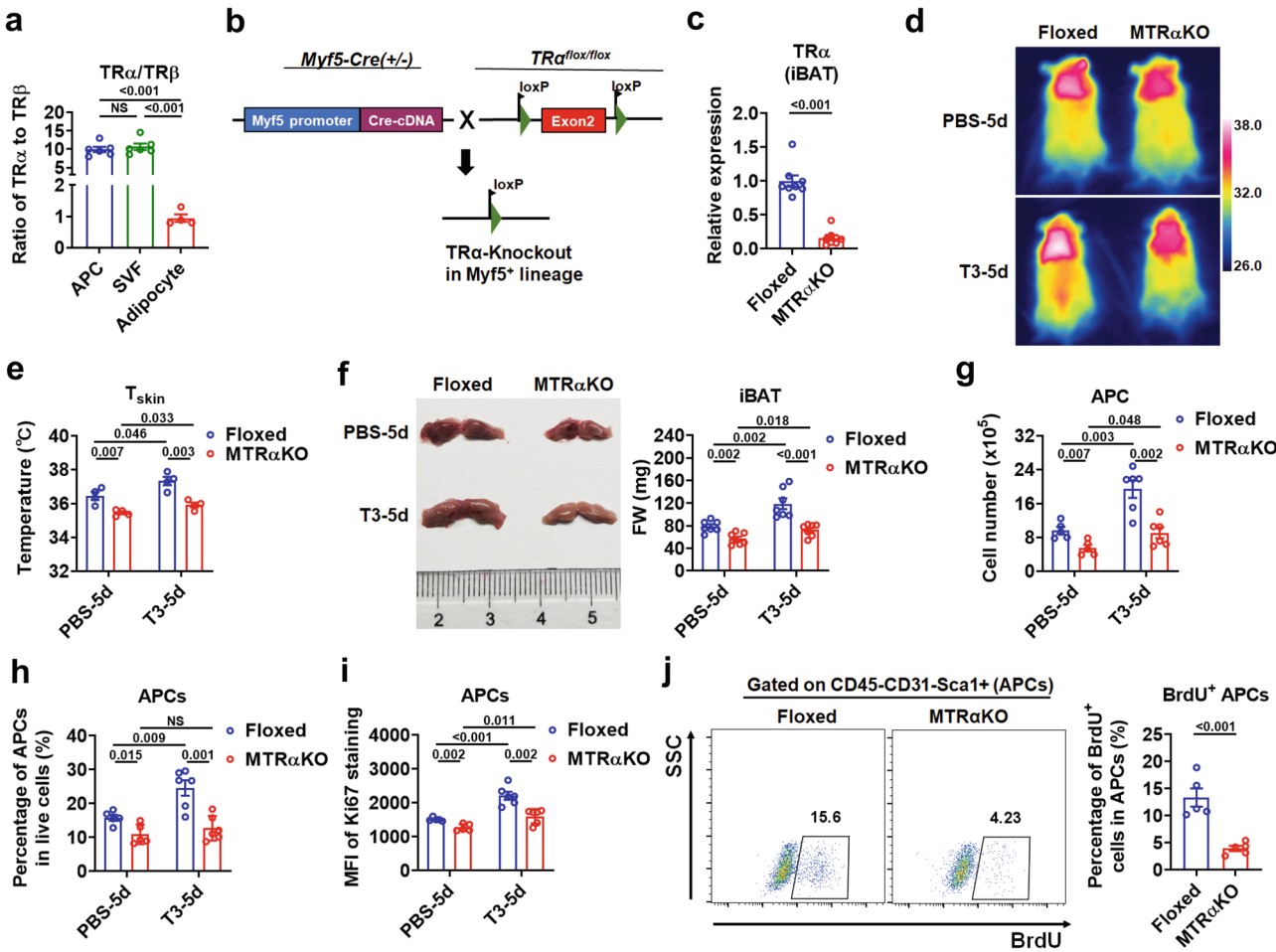

**Fig. 3 Deletion of TRα with Myf5-Cre blunts the T3 effect on iBAT hyperplasia. a** Ratio of the amount of TR transcripts (TRα vs TRβ) assessed by qRT-PCR in APCs (*n* = 6), cultured SVF cells (*n* = 6) and SVF-derived mature brown adipocytes (*n* = 4). **b** Breeding scheme for generating the MTRαKO mice. **c** Relative TRα mRNA levels in iBAT depots of TRαFloxed (Floxed) and MTRαKO mice (*n* = 8). **d**, **e** Representative infrared images (**d**) and quantification (**e**) of average dorsal $T_{skin}$ of Floxed and MTRαKO mice with daily PBS or 0.5X standard dose of T3 injection for 5 d (*n* = 4). **f** Representative photograph (left) and FW of iBAT (right) of Floxed and MTRαKO mice with daily PBS or 0.5X standard dose of T3 injection dose for 5 d (*n* = 7). **g** Number of APCs in iBAT per mouse from Floxed and MTRαKO mice with daily PBS (*n* = 5) or 0.5X standard dose of T3 (*n* = 6) injection for 5 d. **h** Percentage of APCs in live SVF cells of iBAT from Floxed and MTRαKO mice with daily PBS (*n* = 5) or 0.5X standard dose of T3 (*n* = 6) injection for 5 d. **i** MFI of Ki67 staining in APCs of iBAT from Floxed and MTRαKO mice with daily PBS (*n* = 5) or 0.5X standard dose of T3 (*n* = 6) injection for 5 d. **j** FACS plot and quantification of the BrdU⁺ APCs in iBAT from Floxed and MTRαKO mice with daily T3 injection at 0.5X standard dose for 5 d (*n* = 5). Cells were pre-gated on live CD45⁻CD31⁻Sca1⁺ cells. Means ± SEM are shown. *P* values were calculated by two-tailed unpaired Student's *t* test for **a**, **c**, **e**–**j**. NS, not significant. Two-way ANOVA was performed for the data in **g**–**i** additionally, and the results were show in Supplementary Table 1–3. Source data are provided as a Source Data file.

marked by high expression of *Fabp4* and *Plin2*, indicating that these cells are more committed preadipocytes. Group 3 and Group 4 cells exhibited high expression of *Gdf10* and *Fmo2* (Group 3) and high levels of *Cilp* and *Myoc* (Group 4), respectively. Group 5 cells expressed high levels of *Stmn1* and *Cenpa*. Notably, Group 5 cells were enriched for cell cycle-related genes, including *Ccnb1*, *Ccna2*, and *Foxm1* (Supplementary Fig. 6c and Supplementary Data 2). Gene Ontology (GO) analysis of the top 50 up-regulated genes of Group 5 revealed that top enriched GO terms are mitosis-related (Fig. 5k and Supplementary Data 3), suggesting that Group 5 cells are a subpopulation actively undergoing mitotic division. The analysis based on cell cycle phase scoring and calculation of the total mRNA molecules per cell further supports the notion that Group 5 cells are cycling cells in G2/M phase (Fig. 5l, m).

Additionally, the top group-defining genes reported in previous studies by Burl et al[15]., Schwalie et al[18]., and Hepler et al[16]. were overlaid on our scRNA-seq data for comparison, which revealed

similarities for group-defining genes between ours and others (Supplementary Fig. 7–9). Collectively, our results indicate that, similar to APCs in WATs, APCs in iBAT are also a heterogeneous cell population. Although the similarities were observed, our results indicate that cellular APCs landscape differs between iBAT and WAT depots, which might reflect the physiological properties of the depots (Supplementary Note 4). Notably, analysis of our scRNA-seq data and others reported by Burl et al[15]. revealed that TRα is the major TR isoform in the APCs resided in iBAT, iWAT and eWAT (Supplementary Fig. 10a–c and Supplementary Note 4).

**Pseudotemporal trajectory analysis delineates the hierarchy of APCs in iBAT.** To delineate the interrelatedness between the identified APC subpopulations, we organized all APCs using a trajectory inference model. Monocle arranged APCs along a common trajectory that diverged into two distinct branches (Fig. 6a). And all cells could be classified into three states

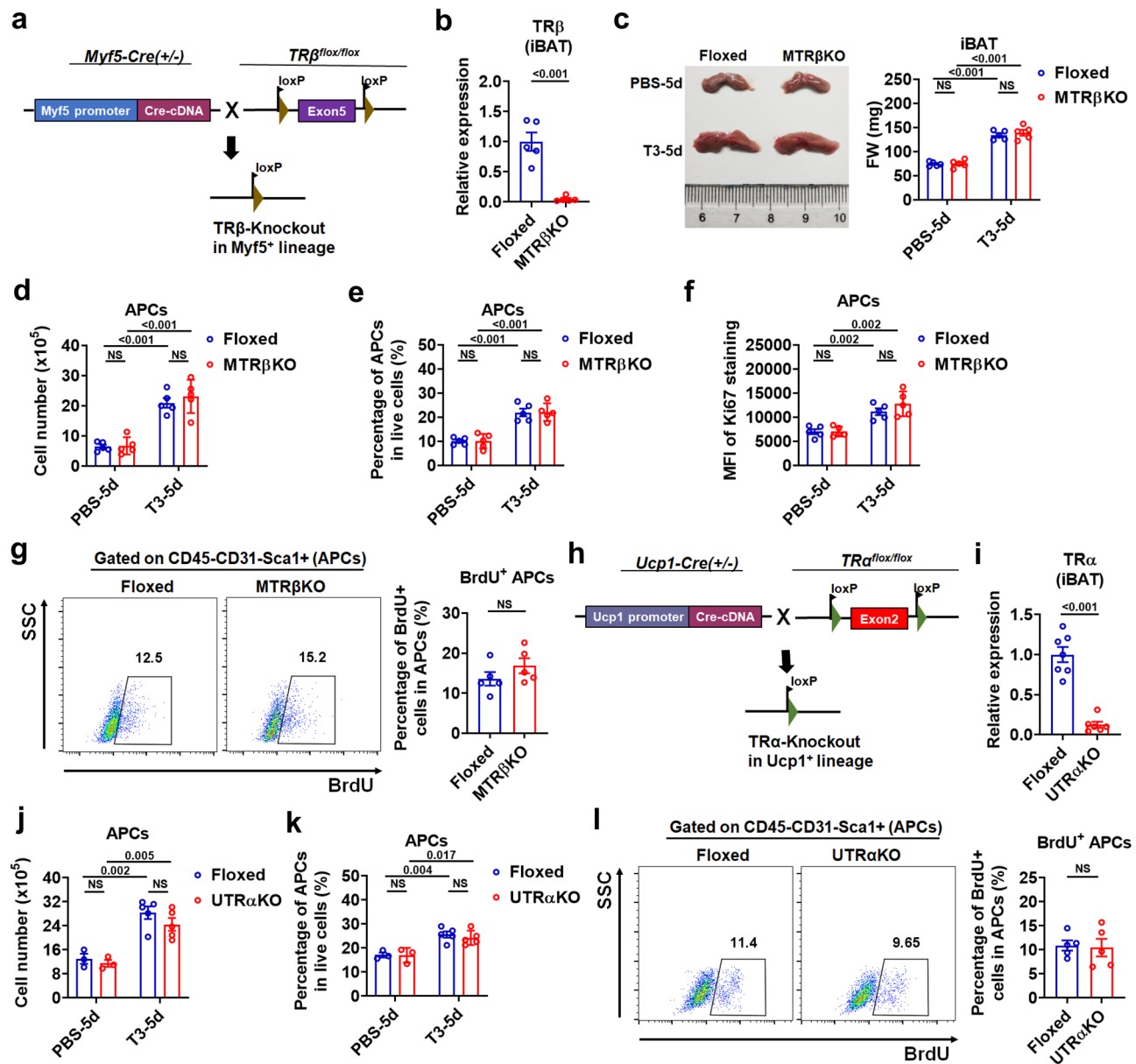

**Fig. 4 Deletion of TRβ by Myf5-cre or TRα by UCP1-Cre barely affects APC proliferation in iBAT. a** Breeding scheme for generating MTRβKO mice. **b** Relative TRβ mRNA levels in iBAT depots of TRβFloxed (Floxed) and MTRβKO mice ($n = 5$). **c** Representative photograph (left) and FW of iBAT (right) of Floxed and MTRβKO mice with daily PBS or 0.5X standard dose of T3 injection for 5 d ($n = 5$). **d, e** Number of APCs in iBAT per mouse (**d**, $n = 5$) and percentage of APCs in live SVF cells of iBAT (**e**, $n = 5$) from Floxed and MTRβKO mice with daily PBS or 0.5X standard dose of T3 injection for 5 d. **f** MFI of Ki67 staining in APCs of Floxed and MTRβKO mice with daily PBS or 0.5X standard dose of T3 injection for 5 d ($n = 5$). **g** FACS plot and quantification of the BrdU$^+$ APCs in iBAT from Floxed and MTRβKO mice with daily PBS or 0.5X standard dose of T3 injection for 5 d ($n = 5$). Cells were pre-gated on live CD45$^-$CD31$^-$Sca1$^+$ cells. **h** Breeding scheme for generating UTRαKO mice. **i** Relative TRα mRNA levels in iBAT depots of TRαFloxed (Floxed) ($n = 7$) and UTRαKO mice ($n = 6$). **j, k** Number of APCs in iBAT per mouse (**j**) and percentage of APCs in live SVF cells of iBAT (**k**) from Floxed and UTRαKO mice with daily PBS ($n = 3$) or 0.5X standard dose of T3 ($n = 5$) injection for 5 d. **l** FACS plot and quantification of the BrdU$^+$ APCs in iBAT from Floxed and UTRαKO mice with daily T3 injection at 0.5X standard dose for 5 d ($n = 5$). Cells were pre-gated on live CD45$^-$CD31$^-$Sca1$^+$ cells. Means ± SEM are shown. $P$ values were calculated by two-tailed unpaired Student's $t$ test for **b–g**, **i–l**. NS, not significant. Source data are provided as a Source Data file.

(Fig. 6b). We found that ~87% of Group 1 cells aggregated in State 1 and accounted for ~90% of cells in State 1, which were plotted tightly at the start of pseudotime (Fig. 6c, d). These results suggest that Group 1 cells are most stem-like in nature. About 90% of Group 2 cells were in State 3 and accounted for ~75% of cells in State 3, which localized primarily towards the end of bottom left branch (Fig. 6c, d). About 63% of Group 3 cells were in State 2 and accounted for ~48% of cells in State 2, which existed primarily towards the end of top branch (Fig. 6c, d).

Group 4 cells did not appear to have any spatial bias, suggesting that Group 4 cells represent a transient state (Fig. 6c, d). Group 5 cells, a subpopulation most closely related to Group 2 according to the Spearman correlation analysis ($r = 0.939875$), were mainly in State 3 and more restricted to the bottom left branch (Fig. 6c, d).

To further define the ordering of cells in pseudotime, we analyzed gene-expression patterns corresponding to different states (Fig. 6e). Consistent with the distribution of different group

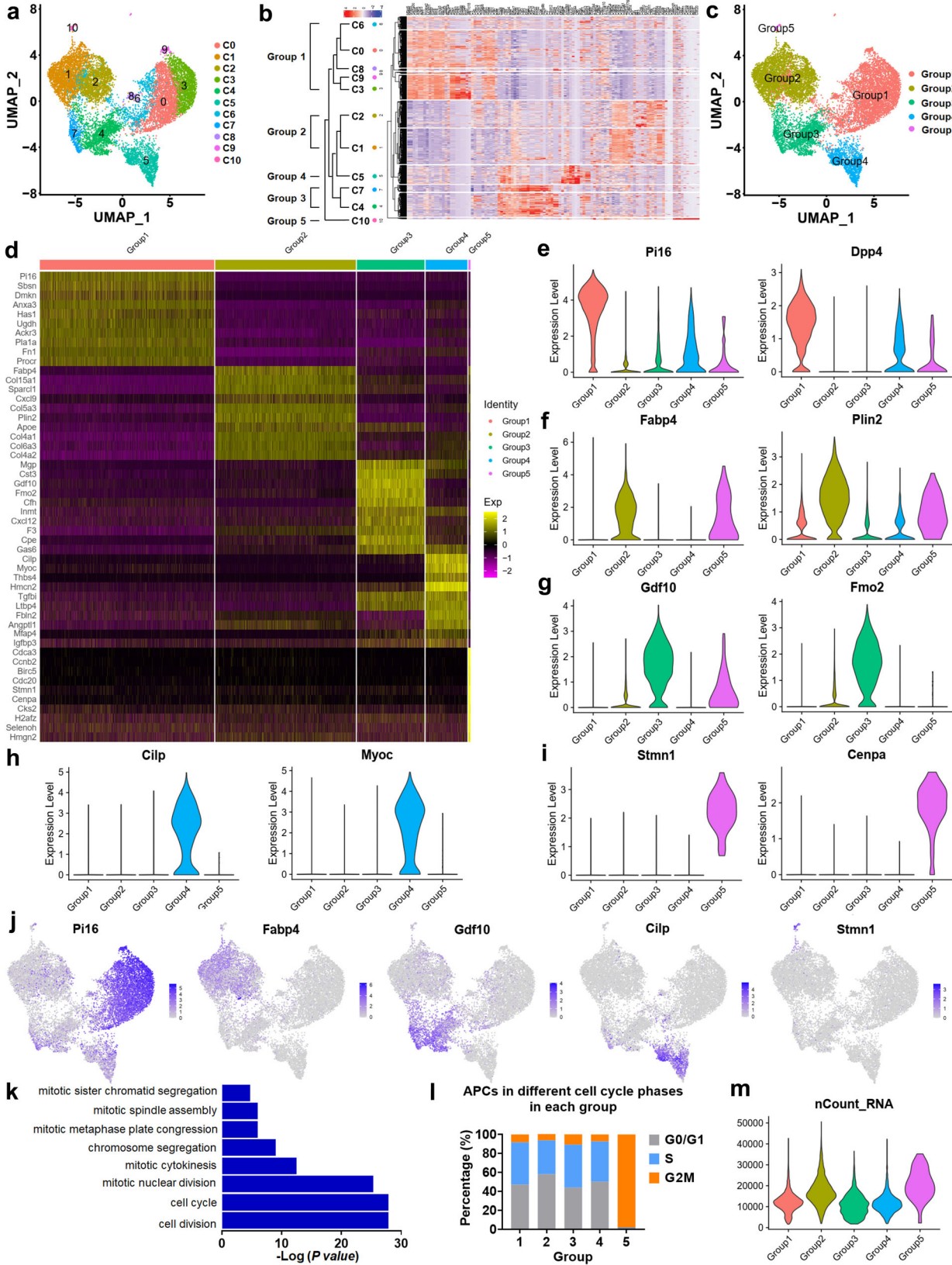

cells in different states, State 1 was enriched for the hallmarks of Group 1 (*Pi16*, etc.), State 2 was enriched for the marker genes of Group 3 (*Fmo2*, etc.), whereas the State 3 was enriched for marker genes of Group 2 (*Fabp4*, etc.) (Fig. 6e and Supplementary Data 4). Furthermore, pseudotime feature plots revealed that the expression of *Pi16* (marker for Group 1) was mainly enriched

towards the start of pseudotime, while the expression of *Gdf10* (marker for Group 3) and *Fabp4* (marker for Group 2) was primary enriched towards the end of pseudotime along the top and bottom left branches, respectively (Fig. 6f). In contrast, the expression of *Cilp* (marker for Group 4) was transiently expressed along the progression of pseudotime (Fig. 6f). Notably, the

Fig. 5 scRNA-seq analysis reveals cellular heterogeneity of APCs in iBAT. a UMAP plot presenting cellular heterogeneity of APCs in iBAT with 11 distinct clusters of cells identified and color-coded. b Unsupervised hierarchical clustering of gene signatures showing relatedness of clusters. c UMAP plot of APCs presenting 5 distinct groups. d Heatmap of differentially expressed genes (DEGs) between different groups. Top 10 genes and their scaled expression are shown. e–j Violin plots (e–i) or feature plots (j) of expression levels and distribution for top ranked (e–i, left and j) and selected (e–i, right) group-specific genes. k GO analysis of top 50 up-regulated genes of Group 5. l Cell cycle scoring for APCs in iBAT. Stacked bar chart showing relative abundance of cells from different cell cycle phases in each group. m Violin plot of total mRNA molecules in distinct groups. P values were calculated by hypergeometric distribution (two-tailed) for k. Source data are provided as a Source Data file.

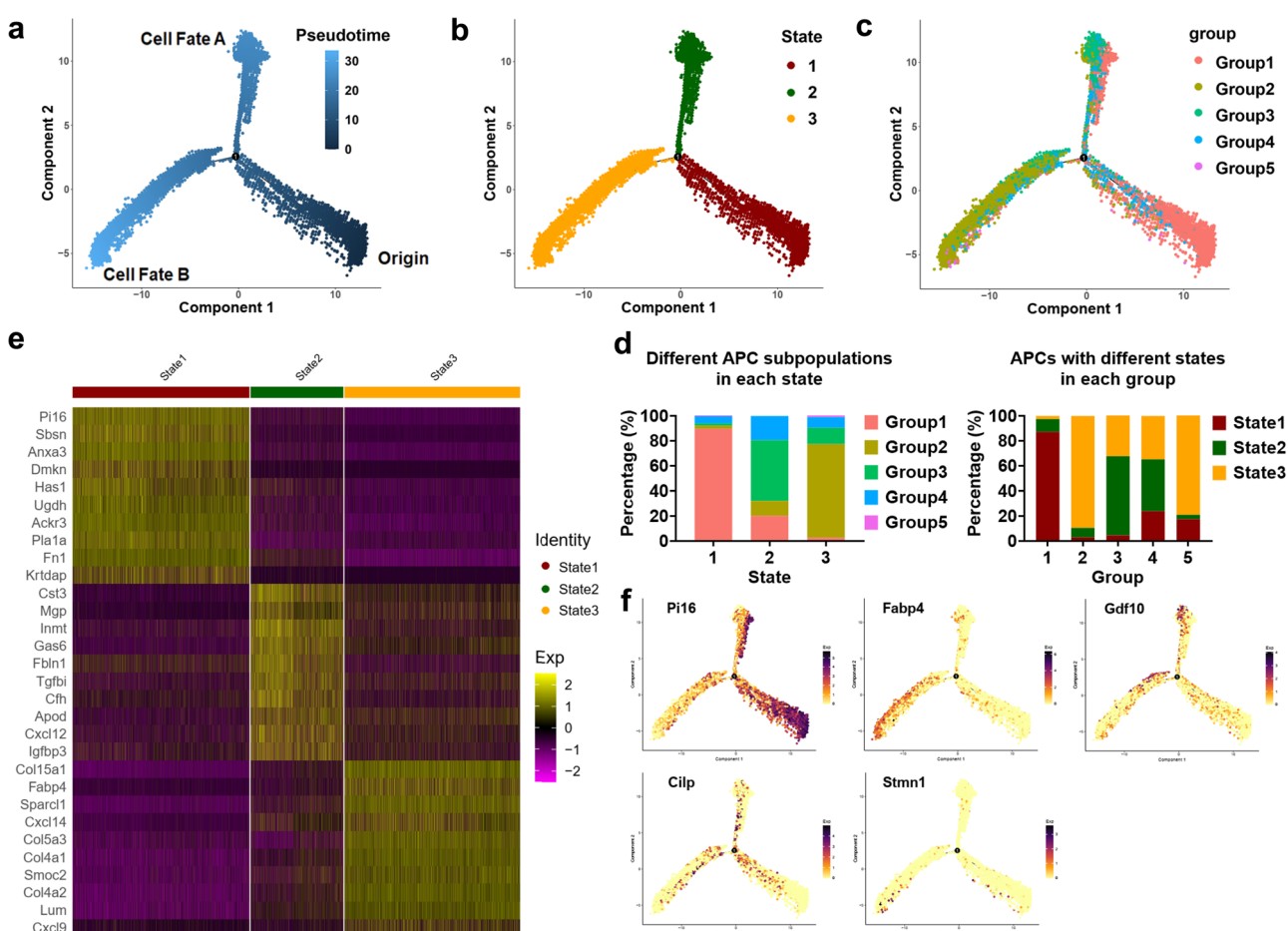

Fig. 6 Pseudotime developmental trajectories of APCs in iBAT. a–c Pseudotemporal cell ordering along trajectories of APCs in iBAT. Pseudotime is depicted from dark to light blue (a). Cells were classified into 3 states (b). Group identities were overlaid on pseudotime trajectory map (c). d Stacked bar chart showing the abundance of cells from distinct groups in 3 states or from distinct states in 5 groups. e Heatmap of DEGs between different states. Top 10 DEGs and their scaled expression are shown. f Feature plots of expression distribution for group markers across pseudotime. Source data are provided as a Source Data file.

expression of *Stmn1* (marker for Group 5) was primarily enriched towards the end of bottom left branch (Fig. 6f).

We also analyzed the trajectory of the Group 1 cells distributed primarily along the root segment of pseudotime, followed by analysis of markers of other groups. The expression levels of markers of other groups were all elevated along the progression of pseudotime, supporting the notion that Group 1 cells stand at the very root of adipogenesis and provide a source of other groups, thereby giving rise to distinct cell states (Supplementary Fig. 11a). By using similar approach, we speculated that Group 2 cells unlikely serve as the primary progenitors for Group 1, 3, and 4 (Supplementary Fig. 11b). Based on our above clustering and pseudotemporal analyses, we hypothesized that progenitors in Group 1 (the 'true' adipose stem cells) have two trajectories: Group 3 cells located in top branch (Cell Fate A) and Group 2

preadipocytes located in bottom left branch (Cell Fate B) (Fig. 6a and Supplementary Note 5).

To validate the pseudotime trajectories of APCs in iBAT, we carried out in vivo cell transplantation studies (Supplementary Fig. 14a, b and Supplementary Note 5). BODIPY staining revealed lipid droplet-containing tdTomato$^+$ cells in either Dpp4$^+$ or Icam1$^+$ (cell-surface markers for Group 1 and Group 2, respectively) APC-derived implants, suggesting that both sort-purified Dpp4$^+$ and Icam1$^+$ APCs can differentiate in vivo (Supplementary Fig. 14c). Moreover, we found that both Dpp4$^+$ APCs and Icam1$^+$ APCs could fully differentiate into matured adipocytes with large lipid droplets, when mice were exposed to cold environment, a potent adipogenic stimulus (Fig. 7a and Supplementary Fig. 14d, e), further supporting the notion that both Dpp4$^+$ and Icam1$^+$ APCs have differentiation potential.

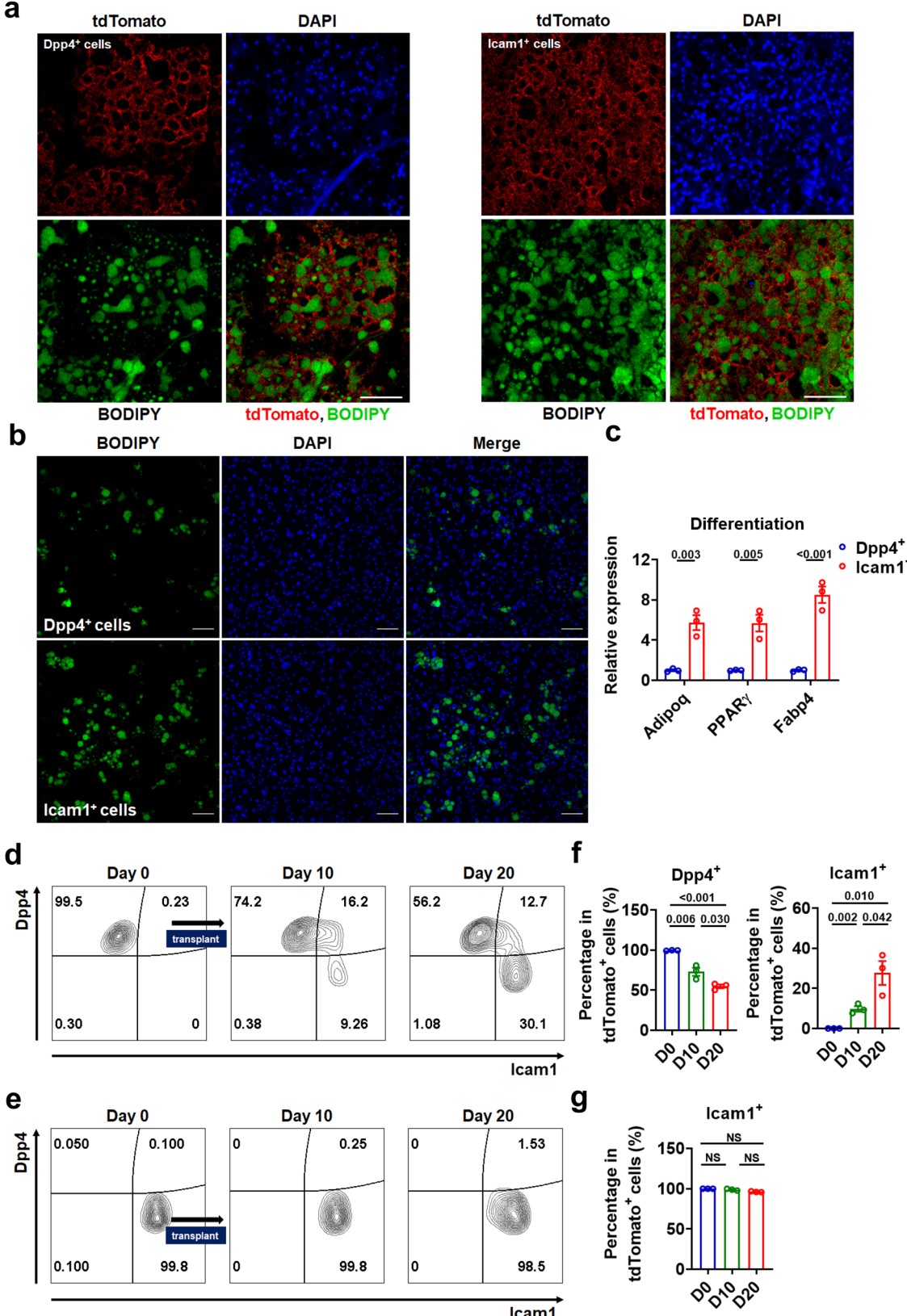

In agreement with our notion that Dpp4+ APCs are more primitive undifferentiated APCs, while Icam1+ APCs are more committed preadipocytes, more lipid droplet-containing adipocytes were observed from the transplanted Icam1+ cells after 20 days of the cell transplantation (Supplementary Fig. 14f) and more adipocytes with larger lipid droplets were detected from

transplanted Icam1+ cells at 30 days post-transplantation (Supplementary Fig. 14g) as compared to those from transplanted Dpp4+ cells at corresponding time points. Similar results could be obtained in cultured Dpp4+ APCs and Icam1+ APCs derived from iBAT under minimal adipogenic conditions in vitro. The Icam1+ cells-derived adipocytes tended to have larger lipid

**Fig. 7 In vivo fate and in vitro adipogenic differentiation capacity of Dpp4+ or Icam1+ progenitors. a** Immunofluorescence staining of tdTomato+Dpp4+ (left panels) or tdTomato+Icam1+ (right panels) APC-derived implants with BODIPY (green), tdTomato (red), and DAPI (blue). One week after cell transplantation, mice were exposed to cold stress for 3 weeks. $5 \times 10^5$ APCs were used for transplantation and the thickness of tissue sections is 12 μm. Scale bar, 100 μm. **b, c** Staining of adipocytes with BODIPY (**b**) and mRNA levels of adipocyte-specific genes (**c**, $n = 3$) in cultured Dpp4+ or Icam1+ cells from APCs of iBAT after minimal adipogenic induction for 8 d. Scale bar, 100 μm. **d, e** FACS plot of Dpp4+ APC-derived (**d**) or Icam1+ APC-derived (**e**) implants before (day 0) and after transplantation (day 10 and day 20). Cells were pre-gated on live tdTomato+ cells. **f, g** Percentages of Dpp4+ cells and Icam1+ cells in tdTomato+ cells from Dpp4+ APC-derived implants (**f**), and percentage of Icam1+ cells in tdTomato+ cells from Icam1+ APC-derived implants (**g**) before (day 0) and after transplantation (day 10 and day 20) ($n = 3$). Means ± SEM are shown. P values were calculated by two-tailed unpaired Student's $t$ test for **c**, **f**, and **g**. NS, not significant. Source data are provided as a Source Data file.

droplets than those derived from of Dpp4+ cells (Fig. 7b). Accordingly, differentiated Icam1+ cells displayed higher levels of adipocyte-specific genes than those in differentiated Dpp4+ cells (Fig. 7c). The findings that cultured Icam1+ APCs differentiated efficiently into adipocytes, while cultured Dpp4+ APCs displayed relatively low adipogenic competence (Fig. 7b, c), further support the notion that Icam1+ cells might be more committed preadipocytes.

To determine the in vivo fate of iBAT-derived Dpp4+ or Icam1+ APCs, we assessed the expression of Dpp4 and Icam1 in tdTomato+ cells after cell transplantation. Flow cytometry analysis of tdTomato+ cells from recipient mice showed that tdTomato+ Dpp4+ cells acquired Icam1 expression as early as day 10 post-transplantation (Fig. 7d). Meanwhile, a subset of these transplanted cells started to down-regulate the Dpp4 expression when they acquired the Icam1 expression (Fig. 7d). In contrast, transplanted tdTomato+ Icam1+ cells did not produce substantial numbers of Dpp4+ cells (Fig. 7e), suggesting that Dpp4+ APCs might serve as progenitors for Icam1+ APCs, which would further differentiate into mature adipocytes. Consistently, we found that the percentages of Dpp4+ cells (Fig. 7f, left) and Icam1+ cells (Fig. 7f, right) in tdTomato+ cells from Dpp4+ APC-derived implants were gradually decreased and increased, respectively, while the percentage of Icam1+ cells in tdTomato+ cells from Icam1+ APC-derived implants remained unaltered after transplantation (Fig. 7g). Collectively, these results support the hypothesis based on our pseudotemporal trajectory analysis that Dpp4+ progenitors (Group 1 cells) are more stem-like progenitors and Dpp4+ progenitors can adopt an adipogenic fate, giving rise to more committed Icam1+ progenitors (Group 2 cells) (Fig. 6). Although above transplant data are supportive of our hypothesis, further studies using lineage tracing are required to offer more definitive evidence.

**T3 promotes cell state transition and cell cycle progression in APCs of iBAT**. After delineating the heterogeneous nature and hierarchical trajectory of APCs in iBAT, we evaluated the T3 effect on APCs at a single-cell resolution. Segregation of the aggregated UMAP plot according to the treatment condition demonstrated that the cell numbers of different APC sub-populations (Group 1-5) were all increased after T3 treatment (Fig. 8a and Supplementary Fig. 15a), suggesting that all APC subpopulations might be proliferative in response to T3 treatment. Accordingly, EdU+Dpp4+ cells could be detected in the iBAT depot after T3 treatment (Supplementary Fig. 15b). Moreover, by performing BrdU labeling and FACS analysis, we found increases in the percentage of BrdU+ cells and Ki67 staining intensity in not only Dpp4+ APCs but also Icam1+ APCs from the iBAT of mice after T3 treatment, indicating they were both proliferative (Supplementary Fig. 15c–f).

Segregation of the aggregated UMAP plot according to the treatment condition also demonstrated that T3 changed the APC subpopulation composition in all major groups rather than altered the degree of overall heterogeneity of APCs in iBAT depot

(Fig. 8a, b and Supplementary Fig. 15g). After the T3 treatment, the proportions of Group 1 and Group 4 were decreased from 53.2% and 12.2% (MMI group) to 31.4% and 7.9% (T3 group), respectively (Fig. 8b). On the other hand, after T3 treatment, the proportions of Group 2, 3, and 5 were increased from 20.4%, 14.0%, and 0.2% (MMI group) to 42.6%, 17.5%, and 0.6% (T3 group), respectively (Fig. 8b). Accordingly, the percentage of cells in State 1 was decreased from 53.2% to 29.4%, while the percentage of cells in State 3 was increased from 20.8% to 53.7% after T3 treatment in MMI mice (Fig. 8b and Supplementary Fig. 15h).

To further test the T3 effect on adipogenic trajectory, we performed FACS analysis. In line with above findings, the percentage of Dpp4+ APCs in iBAT tended to increase at day 1 and then significantly decreased at day 4 after T3 treatment in MMI mice, albeit with a gradual increase in total number of Dpp4+ APCs (Fig. 8c). Accordingly, the percentage of Icam1+ APCs decreased at day 1 and then increased at day 4, while the total number of Icam1+ APCs was not altered at day 1 but markedly increased at day 4 after T3 treatment (Fig. 8d). Collectively, these results indicate that T3 treatment might promote the cell state transition from a more stem-like state towards a more committed adipogenic state.

To explore the underlying mechanisms, we performed GO analysis to classify the functions of DEGs between MMI and T3 group and identified the generation of precursor metabolites and energy (GO:0006091) as the most significantly enriched GO term ranked according to the $p$-value and counts (Fig. 8e and Supplementary Data 5). Interestingly, among the 42 DEGs associated with GO:0006091, 12 genes including *Myc* and its regulated genes were associated with glycolytic process (GO:0006096). The differential expression of these 12 glycolytic genes between MMI and T3 groups in all APC subpopulations from scRNA-seq was visualized by a heatmap (Fig. 8f). The frequency distribution of expression values from scRNA-seq for *Myc* and two downstream effectors of Myc in glycolytic pathway (*Hk2* and *Pgk1*) were additionally visualized by violin plots (Fig. 8g). These findings indicate that Myc-controlled glycolysis might be involved in the T3 action on APCs of iBAT.

Consistently, the effect of T3 on the protein expression of Myc and Hk2 could be observed in BAC cells (Supplementary Fig. 15i). Moreover, knockdown of *Myc* by specific siRNA could attenuate the T3 effect on Hk2 protein expression (Supplementary Fig. 15j). Accordingly, decreased protein levels of Myc were observed in cultured SVF cells from the iBAT of MTRαKO but not MTRβKO mice (Fig. 8h). Further, the time-dependent elevation of Myc protein expression by T3 treatment could not be observed in cultured SVF cells from the iBAT of MTRαKO mice (Fig. 8i). These data indicate that T3 regulates Myc expression not only in a cell-autonomous manner but also in a TRα-dependent manner. As the transcriptional regulation of *Myc* by T3 during Xenopus metamorphosis has been reported previously[21], we then tested whether T3 could control *Myc* expression directly at the transcription level in mammalian cells. Time-dependent elevation

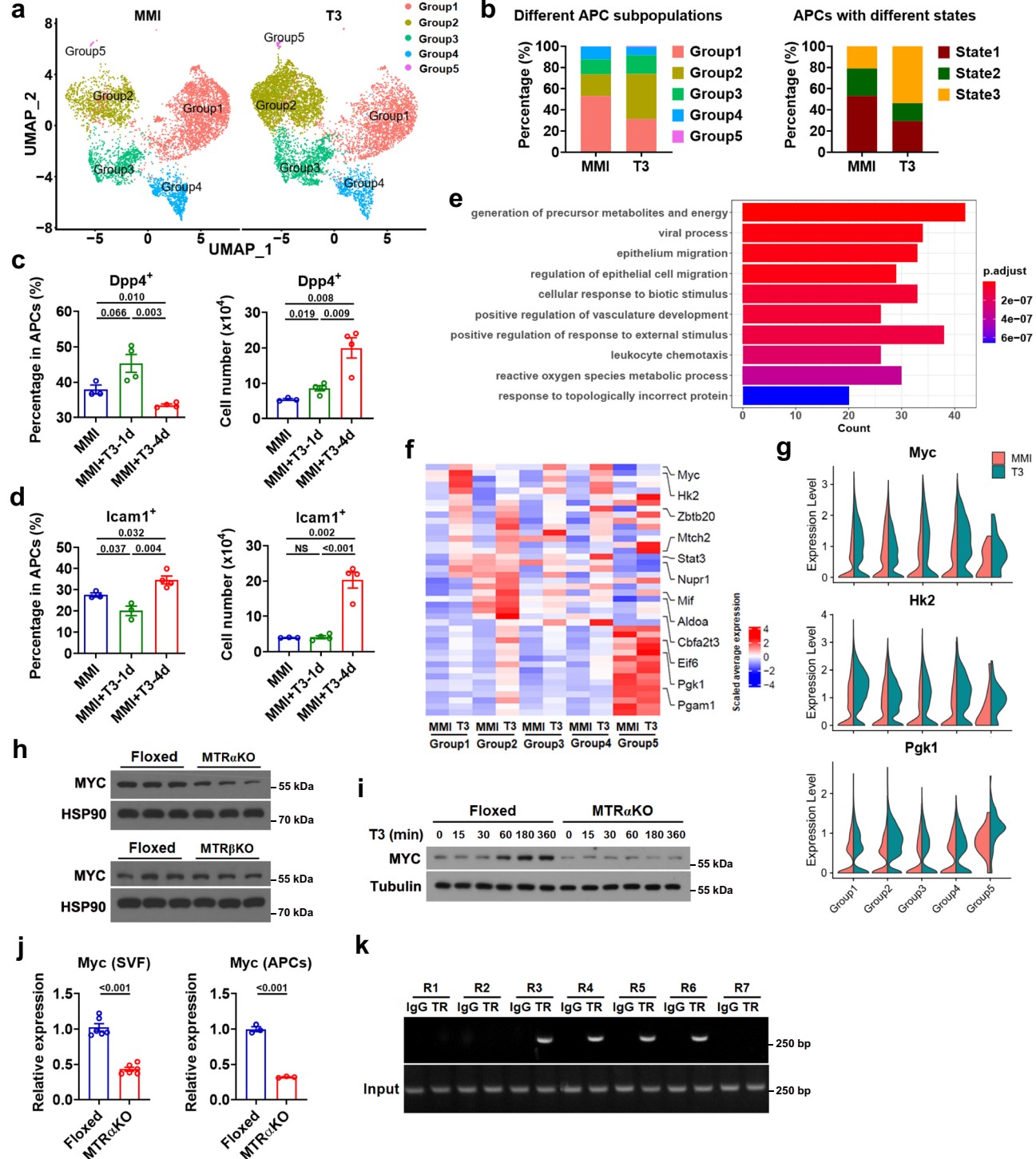

of *Myc* mRNA levels by T3 treatment could be observed in BAC cells (Supplementary Fig. 15k). The mRNA levels of *Myc* were decreased in either cultured APC-enriched SVF cells or APCs from the iBAT of MTRαKO mice (Fig. 8j). The transcriptional regulation of *Myc* by T3 was further confirmed by ChIP assay and luciferase assay, indicating that *Myc* is a direct target gene of T3 and T3 may control glycolysis by direct regulating the transcription of *Myc* in APCs of iBAT (Fig. 8k, Supplementary Fig. 15l–n and Supplementary Note 6).

As metabolism has a central role in regulating cell state transition, we then tested whether increased glycolysis is required for the cell state transition of APCs induced by T3

(Supplementary Fig. 16a, b and Supplementary Note 6). Treatment of either 2-deoxy-D-glucose (2DG), an inhibitor of glycolysis, or F4-10085 (F4), an inhibitor of Myc could blunt the effect of T3 on the mRNA expression of either *Pi16* and *Dpp4* or *Fabp4* and *Plin2* in Dpp4⁺ APCs (Supplementary Fig. 16b). We also employed a shRNA knockdown approach to reduce the expression of *Myc* or *Hk2* in the APCs of iBAT depot and found that knockdown of either *Myc* or *Hk2* could not only abolish the suppressive effect of T3 on the expression of Group 1 markers and attenuate the upregulation of Group 2 marker gene expression by T3 in FACS-sorted APCs derived from iBAT (Fig. 9a, b), but also totally diminish the T3 effect on either the

**Fig. 8 T3 promotes cell state transition and cell cycle progression and regulates *Myc* transcription. a** UMAP plots of APCs from iBAT of MMI mice (MMI group) and MMI mice treated with T3 at 0.5X standard dose for 4 d (T3 group). **b** Stacked bar chart showing relative abundance of cells from distinct groups (left) and from distinct states (right) in MMI group or T3 group. **c, d** Percentage of Dpp4+ APCs (**c**, left) and Icam1+ APCs (**d**, left) and number of Dpp4+ APCs (**c**, right) and Icam1+ APCs (**d**, right) (per mouse) in iBAT of MMI mice (n = 3) and MMI mice with daily T3 injection at 0.5X standard dose for 1 d (n = 4 for **c** and **d**, right; n = 3 for **d**, left) or 4 d (n = 4). **e** GO analysis of the up-regulated genes in T3 group vs MMI group. Bar chart showing the 10 most enriched biological processes. **f** Heatmap showing scaled average expression of genes belonging to GO:0006091 in Group 1–5 from MMI group and T3 group. Genes related to glycolysis are listed on the right. **g** Violin plots showing the expression levels and distribution of *Myc*, *Hk2* and *Pgk1*, split by "orig.ident" (MMI group and T3 group). **h** Western blot analysis of Myc in cultured SVF cells from the iBAT of MTRαKO and MTRβKO mice with their corresponding Floxed mice. **i** Western blot analysis of Myc in cultured SVF cells from the iBAT of Floxed and MTRαKO mice with T3 treatment for indicated time. **j** Relative mRNA expression of *Myc* in cultured SVF cells (n = 6) or APCs (n = 3) from the iBAT of Floxed and MTRαKO mice. **k** ChIP-PCR analysis of the recruitment of HA-TRα to mouse *Myc* promoter in BAC cells after T3 treatment for 24 h. Means ± SEM are shown. *P* values were calculated by two-tailed unpaired Student's *t* test for **c**, **d**, and **j**, and were calculated by hypergeometric distribution (two-tailed) and adjusted by Benjamini-Hochberg correction for **e**. NS, not significant. Source data are provided as a Source Data file.

percentage of Dpp4+ cells or the percentage of Icam1+ cells in these APCs (Fig. 9c). These data further suggest that T3 might promote cell state transition from a more stem-like state towards a more committed adipogenic state via enhancing Myc-mediated glycolysis.

As the greatest proportion change after T3 treatment was observed in Group 5 cells undergoing mitotic division, which showed a 3-fold increase from 0.2% to 0.6% (Fig. 8a, b), we speculated that T3 may also play a role in mitotic progression along adipogenic trajectory (Supplementary Note 6). At a single-cell level, we found that the expression of *Ccnb1*, the G2/M phase marker, was increased by T3 in Group 5 cells (Fig. 9d). Further analysis revealed that the percentage of *Ccnb1* positive cells was significantly increased after T3 treatment, suggesting that more APCs were committed to mitotic division after T3 treatment (Fig. 9e and Supplementary Fig. 16g).

Furthermore, lentivirus-mediated shRNA knockdown of *Myc* and *Hk2* in the APCs of iBAT could markedly attenuate the effect of T3 on the percentage of BrdU+ cells in these APCs (Fig. 9f, g). Accordingly, the T3 effect on the expression of Group 5 markers (*Stmn1* and *Cenpa*) and *Ccnb1* were attenuated by the knockdown of *Myc* and *Hk2* in the APCs of iBAT (Fig. 9h, i). We hypothesized that T3 might promote the cell cycle progression towards a mitotic state through a mechanism also involving Myc-mediated glycolysis, which is not surprising given that glycolysis generates intermediates for the generation of new biomass. Together, based on above findings, we propose that T3 may promote cell state transition and cell cycle progression in adipocyte progenitors via enhancing Myc-mediated glycolysis, hence promoting the hyperplastic growth of iBAT.

It has been suggested that adult human BAT resembles murine beige fat much more closely than it resembles murine classic BAT, cautioning the extrapolation of rodent studies to humans[22]. However, whether APCs resided in adult human BAT are similar to or different from APCs resided in murine beige fat and classic BAT has not been extensively studied. Our analysis indicates that APCs resided in adult human BAT might resemble APCs resided in murine classic BAT (Supplementary Note 6), prompting us to assess the physiopathological relevance of our findings in this study by analyzing the correlation between the T3 level and the number of APCs in human neck brown fat[5]. As T3 level normally declines during aging[23], we compared the number of APCs in human neck brown fat between young and old subjects. As expected, the T3 levels were relatively lower in old subjects than those in young subjects (Supplementary Fig. 16j). In agreement with our above findings, both the number of APCs per gram tissue or the percentage of APCs were lower in old subjects than those in young subjects (Supplementary Fig. 16j). Moreover, a correlation between the percentage of APCs and the T3 level was observed in this cohort (Supplementary Fig. 16k), suggesting that

normal T3 level might be essential for maintaining the APC homeostasis in human brown fat and the declined T3 levels due to aging might contribute to the reduction of APC pool in the brown fat of old subjects.

## Discussions

Less attention has been paid to the recruitment of brown adipocyte progenitors, since classic BAT is previously considered non-recruitable and only beige fat is classified recruitable. By measuring the total protein amount of UCP1, it has been demonstrated that the most recruitment of thermogenic capacity is indeed found in the classic BAT during cold acclimation, which is ultimately achieved by tissue hyperplasia[8,9]. The effects of TH on iBAT expansion were realized early, however, how TH regulates the recruitment of full thermogenic capacity of classic BAT is largely unknown. Here, we show that T3 increases the recruitment of thermogenic capacity of iBAT in male mice through tissue expansion by promoting APC proliferation. Using male mice lacking TRα or TRβ in Myf-5 lineage, we demonstrate that TRα is the major TR isoform responsible for the T3 effect on APC proliferation in iBAT. At a single-cell level, we demonstrate that APCs in iBAT are a heterogeneous cell population and exist in various cell states. T3 stimulates iBAT hyperplasia by promoting cell state transition and cell cycle progression in adipocyte progenitors via mechanisms involving the Myc action on glycolysis. Thus, we identified previously undescribed mechanisms underlying the profound effects of TH on adaptive thermogenesis, hence providing therapeutic targets or strategies for treating obesity or related metabolic diseases. A limitation was that generalizability of the findings to female mice has not been investigated.

ScRNA-seq provides unprecedented opportunities to understand the tissue function and homeostasis. Here, we took advantage of this technology to investigate how TH impacts the adipogenic commitment and mitotic proliferation of APCs in iBAT. Based on our pseudotemporal analysis, experimental data, and available relevant literatures focusing on WAT[15,16,18], we propose that APCs in iBAT depot comprise 5 major groups in various cell states, including Group 1 (most stem-like progenitors), Group 2 (more committed preadipocytes), Group 3 and Group 4 (two transient subpopulations with distinct cell fate potential), and Group 5 (dividing progenitors), further supporting the notion that APCs may exist in various cell states with different cell fate potential[24]. Notably, T3 induced a shift in subpopulation composition for all major groups rather than altered the degree of overall heterogeneity of APCs in iBAT depot, which led to a decrease in the percentage of cells in State 1 and an increase in the percentage of cells in State 3 (Fig. 8a, b, Supplementary Fig. 15a, g, h and Supplementary Discussion). Thus, we propose that T3 promotes not only cell state transition

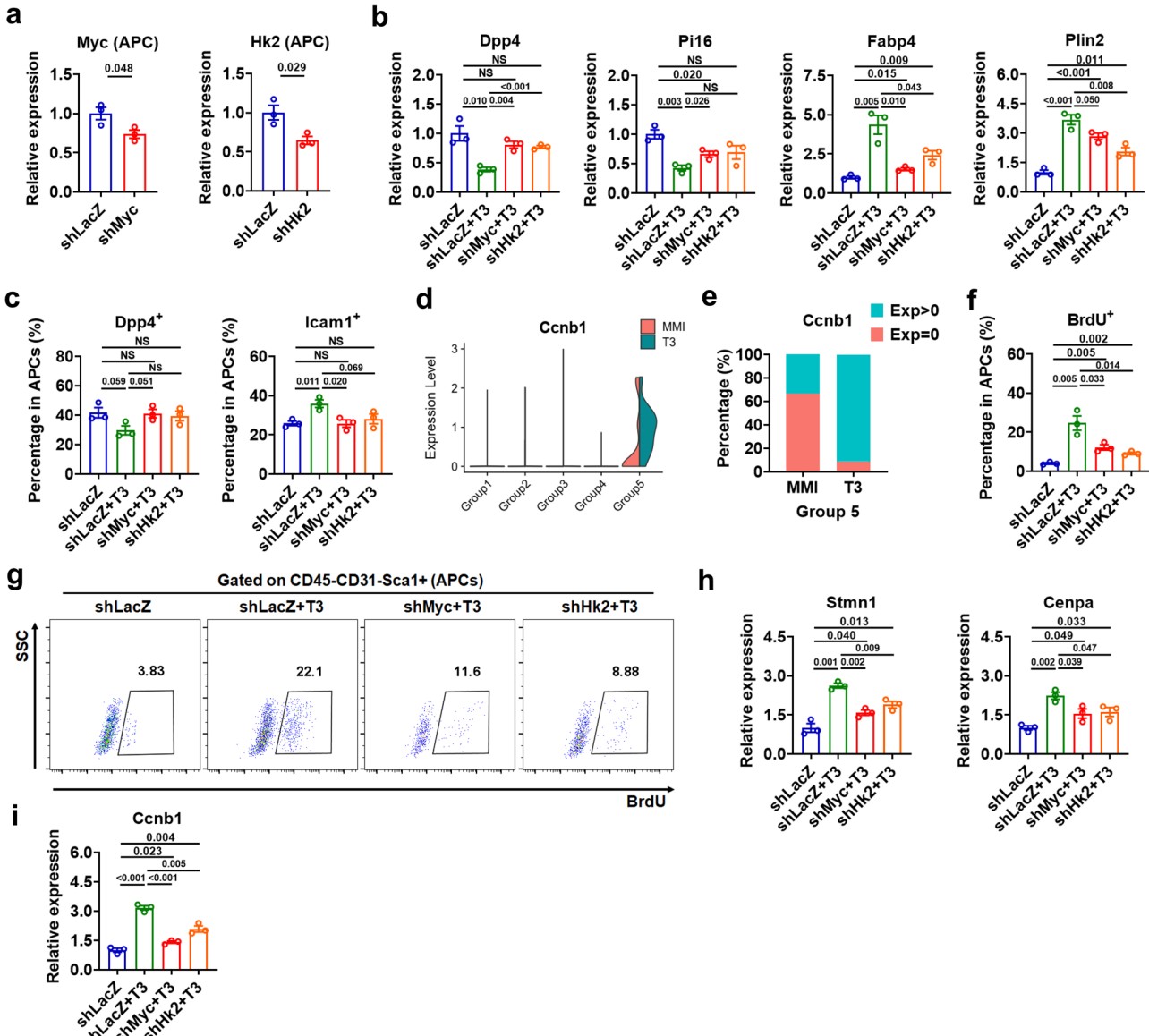

**Fig. 9 Knockdown of *Myc* or *Hk2* attenuates the T3 effect on cell state transition and cell cycle progression. a** qRT-PCR analysis of *Myc* or *Hk2* in APCs from iBAT of shRNA-injected mice ($n = 3$). **b** qRT-PCR analysis of selected Group 1-specific genes and Group 2-specific genes in APCs from iBAT of shRNA-injected mice with daily T3 injection for 4 d ($n = 3$). **c** Percentage of Dpp4⁺ APCs, Icam1⁺ APCs in iBAT of shRNA-injected mice with daily T3 injection for 4 d ($n = 3$). **d** Violin plots showing the expression levels and distribution of *Ccnb1*, split by "orig.ident" (MMI group and T3 group). **e** Comparison of the percentage of *Ccnb1* positive cells in Group 5 cells between MMI group and T3 group. **f** Percentage of BrdU⁺ APCs in iBAT of shRNA-injected mice with daily T3 injection for 4 d ($n = 3$). **g** FACS plot of the BrdU⁺ APCs in iBAT of shRNA-injected mice with daily T3 injection for 4 d ($n = 3$). Cells were pre-gated on live CD45⁻CD31⁻Sca1⁺ cells. **h, i** qRT-PCR analysis of Group 5 markers (*Stmn1* and *Cenpa*) (**h**) and G2/M phases marker (*Ccnb1*) (**i**) in APCs from iBAT of shRNA-injected mice with daily T3 injection for 4 d ($n = 3$). Means ± SEM are shown. *P* values were calculated by two-tailed unpaired Student's *t* test for **a–c**, **f**, **h**, and **i**. NS, not significant. Source data are provided as a Source Data file.

from a more stem-like state towards a more committed adipogenic state but also cell cycle progression towards a mitotic state in APCs of iBAT along the adipogenic trajectory.

Mitosis is a process of cell division that can be used for the organic growth of tissues. Cells spend most of their time in interphase (G1, S and G2 phases) and cycle through interphase and then M phase to proliferate. In general, cells must have adequate carbohydrates, nucleotides, amino acids and fatty acids for the replication of the entire contents of the cell. Intermediary metabolism, especially glycolysis, plays an essential role in cell cycle progression, as increased glycolysis generates glycolytic intermediates and ATP needed for biosynthetic pathways. Correlation between glycolytic capacity and proliferative potential has

been implicated in progenitor cells. In current study, upon T3 treatment, APCs were enriched for intermediates and energy producing pathways, including glycolysis, suggesting that metabolic reprogramming might be functionally involved in the regulation of APCs by T3 (Fig. 8e–g). Further analysis revealed that inhibition of glycolysis could attenuate the T3 effect not only on cell state transition from a more stem-like state towards a more committed adipogenic state but also cell cycle progression towards a mitotic state (Supplementary Fig. 16b, d–f).

Mechanistically, Myc, a master regulator of glycolytic enzymes, might mediate the T3 effect on both cell state transition and cell cycle progression (Supplementary Fig. 16b, d–f). Lentivirus-mediated shRNA knockdown of either *Myc* or *Hk2* in the APCs

of iBAT depot could not only abolish the effect of T3 on the expression of Group 1, 2 and Group 5 markers and *Ccnb1* (Fig. 9a, b, h, i) but also attenuate the T3 effect on the percentages of Dpp4+ cells and Icam1+ cells and the percentage of BrdU+ cells (Fig. 9c, f, g), further supporting the notion that Myc and its regulated glycolysis might participate in the regulation of cell state transition and cell cycle progression in response to T3. Our observations support not only the notion that glycolysis generates the intermediates for the generation of new biomass but also the notion that signals that stimulate the proliferation likely participate in the metabolic reprogramming that allows cells to proliferate. Thus, our study has greatly advanced our understanding of the regulatory mechanisms mediating the T3-induced hyperplastic growth of iBAT at the individual cell level.

## Methods

**Animal and human study.** Experiments involving mice were all in accordance with institutional guidelines for the care and use of animals. Animal protocols were approved by the Ethics Committee of Shanghai Institute of Nutrition and Health, Chinese Academy of Sciences Animal Care Committee (2015-AN-12, 2016-AN-1, SIBS-2019-YH-1). Male wild-type C57BL/6 mice and transgenic mice on the C57BL/6 background aged about 8–12 weeks were used. Age-matched mice or mice from same litters were randomly assigned to each group. Mice were housed under 12 h/12 h light/dark cycle and humidity 50–60% with free access to food (chow diet, MD17121, Mediscience Ltd, China) and water. TRαFloxed mice (Jax, Stock No: 024394), TRβFloxed mice (Shanghai Biomodel Organism Science & Technology Development), and Rosa26-tdTomato reporter mice (Jax, Stock No: 007909) were crossed with Myf5-Cre mice (Jax, Stock No:007893) or UCP1-Cre mice (a thankful gift from Prof. Kong Xingxing at UCLA) as indicated. TH action indicator (THAI) mice harboring a TH-responsive luciferase reporting system were developed previously[25]. To investigate T3 effect in vivo, mice were rendered hypothyroid by the addition of 0.1% methimazole (MMI, Sigma, M8506) and 1% NaClO₄ (Sigma, 410241) in their drinking water for 25–27 days as indicated (MMI mice). Some animals from this group were injected i.p. with T3 at a standard dose (1X) of 0.25 μg per gram body weight or 0.1X and 0.5X the standard dose per day for 2–6 days on day 20–21 of MMI/NaClO₄ treatment (MMI + T3-2/4/5/6d) and then euthanized 24 h after last T3 injection as indicated. Some MMI mice received a single T3 injection at a standard dose and then were sacrificed 4 h later as indicated (MMI + T3-4h). To visualize proliferating cells, euthyroid or hypothyroid mice received daily i.p. injections of T3 at a standard dose for indicated days. And half an hour after last T3 injection, these mice were given 4 i.p. injections of 50 mg/kg BrdU (Sigma, B5002) in sterile PBS every 6 h and then sacrificed 2 h later after BrdU injection as indicated.

Mice were acclimatized to room temperature (RT, 22°C) or 30°C for 20 days prior to T3 injection (0.5X). Mice received TRβ-selective agonist GC-1 (Tocris Bioscience) treatment at an equimolar dose to T3 (0.5X) for 5 days. ICV treatments were performed with a stereotaxic frame (Steolting, IL, USA) and body temperature of the animals was maintained with a heating pad. ICV cannulae were stereotaxically implanted into the lateral ventricle in the brain and fixed to the skull with dental cement. The mice were allowed to recover from anesthesia on a heat blanket and then were i.p. injected with antibiotics (Ceftriaxone sodium, 0.1 g/kg) for 3 days to prevent infection. After a 7 days recovery period, the mice were ICV injected with 1.0 μl of T3 (8 ng/day) or saline during 5 days. The dose of T3 was selected as reported previously[26]. THAI mice received local treatment of T3 (0.125X) to the iBAT region for 5 days through rolling microneedles. In this study, a derma rolling system (Derma Roller) with fine needles (MEVOS Medical Technology, Suzhou, China) was applied for transdermal delivery of T3 to the iBAT of mice. This model can be connected to syringe and the length of microneedles is 1.5 mm to achieve a transdermal delivery. After the interscapular region of anaesthetized mice was shaved, the skin was punctured with the roller. 60 μl of T3-containing saline was than added during rolling process either via a syringe or by manual pipetting. Mice received daily i.p. injection of SR59230A (3 mg/kg/day; MCE) or metoprolol (5 mg/kg/day; APExBIO) for 5 days three hours prior to T3 injection (0.5X).

For transplantation, APCs of iBAT from Rosa26-tdTomato reporter mice with Myf5-Cre were sorted by FACS as donor cells. 5 or 10 × 10⁵ tdTomato+ APCs were mixed 1:1 with Matrigel on ice. After mice were anesthetized and shaved, a small cutaneous incision was made to expose interscapular region, and then 5 μl of mixture was injected into iBAT depot (normally to one lobe of the iBAT) by using a microsyringe (Shanghai Gaoge Industry & Trade Co., LTD). The brown adipose pad containing the implants was harvested for histology procedures or FACS at the indicated timepoint. To enhance the differentiation of transplanted APCs, one week after transplantation, mice were exposed to cold stress (4°C) for 3 weeks as indicated.

For study using human samples, 16 patients (8 males and 8 females, aged between 29–68 years old) with normal free T3, free T4, and TSH levels were recruited before thyroidectomy due to papillary thyroid cancer. Adipose tissues were collected from the neck by an experienced surgeon during thyroidectomy. The study protocol was approved by the Ethics Committee of Zhongshan Hospital, Fudan University. Informed consent was obtained from all participants.

BAT temperature was measured at RT, using an infrared camera (MAG 384 × 288; Magnity Technologies) according to previous publications. Mice were anaesthetized using an i.p. injection of isoflurane. The images were captured from the backs of the mice and were displayed with the rainbow high contrast color palette with a temperature linear display between 26°C and 38°C. Indirect calorimetry measurements of squared hair-free interscapular area were exported to CSV files. Then surface temperature was calculated as the average of the highest 10% area of the interscapular area as reported by Crane et al[27]. Emissivity was 0.95.

**Lentivirus production and administration.** By using on-line siRNA software (broadinstitute.org) for siRNA sequences were designed against the mouse *Myc* and *Hk2* sequence. The details of the sequence are listed in Supplementary Data 6. Paired deoxyribonucleotide oligos encoding the shRNAs were synthesized, annealed, and cloned into the EcoRI and AgeI sites of the pLKO.1 vector. HEK293T cells (ATCC) were co-transfected with the constructs described above and the lentivirus packaging plasmids psPAX2 and pMD2.G. The virus in a conditioned medium was harvested, filtered and purification by Amicon® Ultra-15 Centrifugal Filter Unit (Millipore, UFC910096). Then, centrifuging twice at 1400 *g* for 40 min at 4°C. Direct lentivirus injection into brown adipose was performed according to the protocol reported by Aileen Balkow et.al[28].

**Cell isolation, culture, treatment, counting and FACS analysis.** To isolate SVF cells, tissues were minced into pieces and digesting in HBSS containing 1 mg/ml Collagenase Type I (Sigma) in a 37°C shaking water bath with appropriate time. For human samples, adipose tissue (200–500 mg) was minced and digested at 37°C for one hour. Digests were filtered through a 70 μm cell strainer (BD Biosciences) and centrifuged at 800 *g* for 10 min and then resuspended in red blood cell lysis buffer before further analysis. To count mature adipocytes and SVF cells, iBAT of mice was excised, minced, and digested at 37°C for 20 or 40 min and the homogenates were filtered through a 70 μm cell strainer. Mature adipocytes were collected from the top of tubes after centrifugal separation at 300 *g*. SVF cells were isolated from the bottom after centrifugal separation at 800 *g*, and incubated in red blood cell lysis buffer for 5 min on ice before counting.

For FACS analysis, SVF cells were centrifuged at 800 *g* for 5 min at 4°C after lysis of red blood cell, and then resuspended in staining buffer (0.1% BSA in PBS). Dead cells were stained with live and dead violet viability kit (Invitrogen), then gated out in analysis. Anti-mouse CD16/32 antibody (Invitrogen) was used for murine cells to block nonspecific binding before surface staining. Afterwards, cells were washed once with PBS followed by staining with surface antibodies. Following cell-surface antibodies were used in this study: FITC anti-mouse CD31 (1:200; Biolegend), PE/Cyanine7 anti-mouse CD31 (1:100; Biolegend), APC/Cyanine7 anti-mouse CD31 (1:200; Biolegend), APC/Cyanine7 anti-mouse CD45 (1:100; Biolegend), PE anti-mouse Ly-6A/E (Sca-1) (1:100; Invitrogen), FITC anti-mouse Ly-6A/E (Sca-1) (1:100; Invitrogen), APC anti-mouse CD26 (DPP-4) (1:200; Biolegend), PE/Cyanine7 anti-mouse CD54 (ICAM-1) (1:100; Biolegend), FITC anti-human CD31 (1:50; Invitrogen), APC/Cyanine7 anti-human CD45 (1:50; Biolegend) and PE/Cyanine7 anti-human CD90 (1:50; Biolegend). For nuclear Ki67 staining, cells were fixed and permeabilized using a Foxp3/Transcription Factor Staining Buffer Set (Invitrogen), then incubated with Alexa Fluor® 488 anti-mouse/human Ki-67 (1:200; Biolegend) for 40 min at 4°C. For BrdU analysis, BrdU-pulsed cells were washed, fixed and permeabilized using FITC BrdU Flow Kit (BD Pharmingen). Cells were then treated with DNase for 1 hour at 37°C, then stained with anti-BrdU antibody (1:100; FITC BrdU Flow Kit) for 20 min at room temperature. Flow cytometry data were collected using Gallios flow cytometer (Beckman). FACS sorting were carried out using Moflo Astrios EQ.

For culture of SVF cells, iBAT of newborn mice were excised, minced and digested at 37°C for 30 min. Cell suspensions were filtered through 100 μm filters and collected at 800 *g* for 5 min. The isolated cells were seeded in culture dishes and expanded in DMEM, supplemented with 10% FBS and 100 U/ml penicillin and 1,000 U/ml streptomycin at 37°C with 5% CO₂. Immortalized BAC was obtained after infection of SVF-derived preadipocytes from iBAT of newborn mice with SV40 retrovirus. For T3 treatment, 100 nM T3 was added to the TH-deficient (TD) medium containing of 10% TD serum and 90% DMEM. To analyze UCP1 protein stability, SVF-derived preadipocytes from iBAT were treated with induction medium (DMEM containing 10% FBS with 0.125 μM indomethacin, 0.5 mM IBMX, 1 μM dexamethasone, 5 μg/ml insulin and 5 μM rosiglitazone) for 2 days and differentiation medium (DMEM containing 10% FBS with 5 μg/m insulin and 5 μM rosiglitazone) for 4 more days. After adipogenic induction, cells were incubated with or without T3 and treated with 50 μg/mL cycloheximide for 6 or 12 h to inhibit protein synthesis. For adipogenic commitment induction, freshly isolated Dpp4+ APCs or Icam1+ APCs by FACS were cultured in growth medium until near confluence, then treated with Mixture of the Differentiation Inducers (MDI) for indicated time: DMEM containing 10% FBS with the addition of a minimal adipogenic cocktail (10 nM insulin, 1 nM rosiglitazone). Cultured Dpp4+ cells were maintained in minimal adipogenic induction medium and treated with T3 (100 nM), F4-10058 (100 μM) or 2DG (5 mM) as indicated. siRNAs for mouse *Myc* were transfected into BAC using Lipofectamine 2000 (Invitrogen), while non-

specific siRNA was used as negative controls. For BrdU analysis of BAC, cultured SVF cells or Dpp4+ cells, 10 μM of BrdU solution was added when cells were approximately 80% confluent. After 3 h of labeling, cells were collected and stained by using FITC BrdU Flow Kit as described above.

Luciferase assays were performed using Dual-Luciferase® Reporter Assay System (Promega). To analyze luciferase activity in tissue samples from THAI mice, proteins were collected from tissues by using luciferase lysis buffer and quantified using a BCA Protein Assay Kit. To analyze mouse *Myc* promoter activity, HEK293T cells were planted in 48-well plates and transfected with pGL3-Basic or a reporter containing ~6k promoter region of *Myc* and then treated with or without T3 (100 nM) for 24 h. The pRL-TK vector was used to normalize the luciferase activity. Cells were lysed 48 h after transfection and measured luciferase activity. ChIP assays of BAC cells were performed using an EZ Magna ChIP Gkit (Millipore) according to the manufacturer's protocol. BAC cells were transfected with HA-TRα and treated with T3 for 24 h. Immunoprecipitation was performed using an anti-HA antibody or with rabbit IgG (Cell Signaling Technology) as a negative control. PCR products were resolved by electrophoresis in a 2% Agarose-gel (Invitrogen). Primer sequences for ChIP assay are provided in Supplementary Data 6.

**Single-cell RNA sequencing**. Cells of iBAT were isolated from MMI mice (MMI group, *n* = 30 mice pooled) and MMI mice treated with T3 at 0.5X the standard dose for 4 days (T3 group, *n* = 8 mice pooled) and sorted by FACS to obtain APCs (Lin−CD31−Sca1+ cells). Live cells were resuspended in 0.04% BSA and counted using the automated cell counter (Countess). Single cell gel beads in emulsion (GEMs) generation, barcoding, post GEM-RT cleanup, cDNA amplification, and cDNA library construction were performed using Single Cell 3′ v3 chemistry (10x Genomics), according to manufacturer's protocol. The libraries were sequenced on an Illumina HiSeq X Ten (Illumina). The Cell Ranger Single Cell Software Suite v.3.0 was used to de-multiplex individual cells, process UMIs, and count UMIs per gene, following standard pipeline. Individual count tables were merged using CellRanger aggr function. Using cellranger mkfastq and cellranger count functions, FASTQ files were generated and aligned to the mm10 reference genome (10x Genomics), sequencing reads were filtered by base-calling quality scores, and then cell barcodes and UMIs were assigned to each read in the FASTQ files. Subsequent data analysis was carried out in R 3.5.1 and the Seurat package (v 3.2.2).

As a quality-control step, Seurat was used to trim datasets (min.cells = 3, 1000 < nFeature_RNA < 6000, percent.mt < 0.1). Overall, 12,715 cells (5513 cells from MMI group and 7202 cells from T3 group) were available for further analyses. After filtering, UMI count matrixes were normalized in log. To integrate the two groups of datasets, top 2000 variable features were used as input to perform canonical correlation analysis and create potential anchors with FindIntegrationAnchors function. Subsequently, IntegrateData function was used to integrate data. To reduce dimensionality, principal component analysis (PCA) was performed on data matrix. With JackStraw and ScoreJackStraw functions, the scores of 30 PCs were calculated and top 20 PCs were determined for downstream analysis. Cell clusters were identified with the FindClusters function, with resolution set as 0.5. Two-dimensional UMAP plots were used to visualize resulting clusters. Differential marker genes were identified using FindAllMarkers function with Wilcox test (only.pos = TRUE, min.pct = 0.25, logfc.threshold = 0.25). Violin plots, heatmaps and individual UMAP plots for given genes were generated by Seurat toolkit VlnPlot, DoHeatmap, and FeaturePlot functions, respectively. Cell cycle scoring was performed using CellCycleScoring function.

Trajectory analysis was performed using Monocle R package (v 2.18.0). Ordering genes were selected using a cutoff of expression in at least 10 cells and a combination of inter-cluster differential expression and dispersion with a *q* value cutoff of <1 × 10⁻³. Monocle feature plots and pseudotime kinetic curves were generated by using plot_cell_trajectory and plot_genes_in_pseudotime functions, respectively. After using branch expression analysis modelling (BEAM), split heatmap and kinetic curves were plotted by using plot_genes_branched_heatmap and plot_genes_branched_pseudotime functions respectively, and gene divergence between two kinetic curves was evaluated by a likelihood ratio test[29]. To identify the genes differentially expressed across states, the labels were transferred back to the Seurat dataset and differential expression analysis was performed as described above.

Single-nuclei RNA-seq count matrices from human deep-neck fat reported by Sun et al. were downloaded with accession ID of E-MTAB-8564 from ArrayExpress[30]. scRNA-seq raw data of mouse iWAT and eWAT reported by Burl et al. were downloaded with accession IDs of SRX4074084, SRX4074085, SRX4074088 and SRX4074089 from Sequence Read Archive[15]. Sequence alignment, quality control, normalization, data integration and cell clustering were performed using standard workflows as described above. Preadipocytes or APCs were selected based on marker genes provided by the authors. To performed a cross-species comparison analysis, mouse genome was used as the reference gene list, and human genes were transformed into mouse orthologs by homologene R package. If human's *GENE1* had two reads, and *GENE2* three reads, the mouse ortholog *Gene* received a total five reads[31]. Preadipocytes or APCs were then integrated and clustering by Seurat R package for combined analyses across species. 13 subpopulations were determined by appropriate PCs and resolution[32]. Spearman rank correlation of global expression signature was calculated by average

expression levels in each sample which was regarded as pseudo-bulk RNA-seq data[31]. Among the 13 subpopulations, 11 were identified in both human and mouse datasets. Spearman rank correlation of average expression levels in each matched subpopulation in human and mouse was calculated[33].

**Analysis of mRNA and protein expression**. Tissues or cells were immediately homogenized in TRIzol reagent (Invitrogen) to isolate the RNA, followed by the reverse transcription using RT Reagent Kit (Takara). Real-time PCRs were performed using a commercial master mix (Takara) and 7900 ABI Real-Time System (Applied Biosystems). Primers used in this study were provided (Supplementary Data 6). An average cycle threshold (Ct) value was calculated from the duplicate reactions and normalized to the expression of 18S, and the ΔΔCt value was then calculated. To quantify the expression of TR isoforms, absolute quantification of TRα and TRβ was performed. Two pairs of primers were designed to amplify fragments from mice cDNA, which were used as templates for standard curves. The other two pairs of primers were designed to perform regular real-time PCR to get specific copy numbers according to corresponding standard curves.

For western blot analysis, proteins were collected from tissues by using RIPA buffer plus proteinase and phosphatase inhibitor, quantified using a BCA Protein Assay Kit, and then subjected to immunoblotting. Primary antibodies against UCP1 (Abcam), Tyrosine hydroxylase (Millipore), MYC, HK2, HSP90, GAPDH and Tubulin (Cell Signaling Technology) were used. The intensity of protein bands was quantified using Image J. To determine the amount of UCP1 protein, proteins (5 μg per lane) were loaded onto a 10% SDS PAGE. The amount of UCP1 protein in the standard sample was designated as 1 a.u. of UCP1.

**Morphological and histological study**. Mouse tissues were fixed in 4% paraformaldehyde and embedded in paraffin. Sections were stained with hematoxylin and eosin according to standard protocols. Immunohistochemical staining of paraffin sections was carried out with 1:200 anti-UCP1 (Abcam). For Immunofluorescence, slides were fixed with 4% PFA and permeabilized in ice cold methanol. Heat-mediated antigen retrieval with a 0.01 M citric acid (pH 6.0) was performed for 5 min in a microwave. After blocking with 4% BSA, the sections were blocked with unconjugated AffiniPure Fab Fragment (1:100; Jackson), then incubated with primary antibodies anti-UCP1 (rabbit; 1:200; Abcam), anti-red fluorescent protein (rabbit; 1:500; Rockland), anti-PDGFRα (goat; 1:200; R&D) or anti-DPP4 (goat; 1:200; R&D) followed by detection with secondary antibodies Alexa Fluor® 594 conjugated goat antibody to rabbit IgG (1:1,000; Invitrogen) or Alexa Fluor® 647 conjugated donkey antibody to goat IgG (1:1,000; Invitrogen). Finally, the sections were incubated with DAPI or Hoechst. The adipocytes were stained with BODIPY® 493/503 (Invitrogen). EdU incorporation assay was performed following the instruction of Click-iT® Plus EdU Alexa Fluor® 488 Imaging Kit (Invitrogen).

**Statistics and reproducibility**. Representative images of at least three independent experiments with similar results were shown in Fig. 8h, i, k, and Supplementary Figs. 1a, c, 14a, 15b, i, j. Representative images of two independent experiments with similar results were shown in Figs. 2h, 7a, b, and Supplementary Figs. 1b, 1e, 3e, 3f, 14c–g. All remaining experiments were performed at least twice, and representative data are shown. Data are expressed as mean ± SEM. GraphPad Prism and two-tailed unpaired student's *t* test were applied as indicated. ZEN 2.3 (blue edition) software was used for acquisition of images from confocal microscope. ImageJ (National Institutes of Health) was used for Western blot densitometry analysis. FlowJo was used for FACS analysis. For scRNA-seq data analysis, differences in gene expression levels were tested with Wilcoxon rank sum test (using Seurat in R) and *P* values were adjusted for false discoveries. The *P* values between split kinetic curves were calculated by a likelihood ratio test (using Monocle in R). Two-way ANOVA was performed for the data in Fig. 3g–i additionally. A P value of less than 0.05 was considered significant.

**Reporting summary**. Further information on research design is available in the Nature Research Reporting Summary linked to this article.

## Data availability
Data supporting the findings of this study are available within the article and its Supplementary information files or from the corresponding author upon reasonable request. The scRNA-seq data generated in this study have been deposited in the Gene Expression Omnibus database under accession code GSE186188. The scRNA-seq data from other publications used in this study are available in the ArrayExpress under accession code E-MTAB-8564[30], and Sequence Read Archive under accession code SRX4074084, SRX4074085, SRX4074088 and SRX4074089[15]. Source data are provided with this paper.

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

## Acknowledgements

This work was supported by National Key Research and Development Program of China No. 2021YFA1100500 (H.Y.); National NSFC 91957205 (H.Y.), 81770865 (X.G.), 82070821 (J.J.), 81970748 (L.Z.), 81870541 (Z.L.); Pujiang Talent Program from STCSM 21PJ1416100 (Y.L.); Youth Innovation Promotion Association, CAS 2021261 (Y.L.); Laboratory for Marine Drugs and Bioproducts of Pilot National Laboratory for Marine Science and Technology (Qingdao) LMDBKF-2019-04 (H.Y.); NHC Key Laboratory of Food Safety Risk Assessment (2020K02 to Y.W.), Hungarian National Brain Research Program 2017-1.2.1-NKP-2017-00002 (C.F. and B.G.), NRDIO K125247 (B.G.), and Nature Science Foundation of Shanghai 202R1444400 (X.F.). The authors thank Prof. Yifu Qiu (Peking University, China), Prof. Jiqiu Wang (Shanghai Jiaotong University Medical School affiliated Ruijin Hospital, China), and Prof. Ju Qiu and Prof. Haibing Zhang (SINH, CAS, China) for guidance and expertise in scRNA-seq, flow cytometry, stem cell, adipose tissue biology, and ICV injection. The authors would like to thank Zhonghui Weng et al. from ICSTF of SINH, CAS for technical assistance.

## Author contributions

S.L., S.S., J.J. and H.Y. designed the experiments and analyzed the data; S.L. and S.S. carried out most of the experiments; Y.Y., C.S., Z.L., H.F., Y.M., Z.T., J.Y., Y.W., C.F., B.G., P.M., F.Y., M.S. and L.Z. provided the technical assistance, analyzed and interpreted the data; F.G., X.F., C.H., X.G., Y.L., M.S. and J.J. contributed to the discussion and supervised the project; S.L., S.S., Y.Y., J.J. and H.Y. wrote and revised the manuscript.

## Competing interests

The authors declare no competing interests.
