## [Peer Review File · Nature Communications]

REVIEWER COMMENTS

Reviewer #1 (Remarks to the Author):

Drs. Jiang and Ying's group took a closer look at the effect of thyroid hormones on the proliferation of APC in interscapular "classical" BAT. It has been known for many years that thyroid hormones positively regulate adipose tissue thermogenesis (e.g. Johann et al Cell Reports 2019), and although this is not novel per se, the phenotypic analysis of tissue-specific receptor knockout using two Cre models (Myf5 and Ucp1) and the detailed investigation of the effect of T3 by single cell analysis are important developments in the field. If the following requirements are met, I recommend publication in Nature Communications.

On the analysis of human neck brown fat.

It is known that human supraclavicular BAT resembles to mouse "recruitable" beige adipose tissue from studies by several groups (e.g. Shinoda et al, Nature Medicine 2015). Thus it is inappropriate to directly apply the authors' findings from mouse using "classical" BAT to human neck brown fat. In addition, CD90 is used to isolate APCs from human, but isn't CD140A (PDGFRa) more common (Oguri et al Cell 2020)? Are CD90 and CD140A expressed in the same population?

Regarding the analysis of single cell RNA-seq. The authors should show the mRNA levels of TR α /TR β and Myf5, the driver of Cre, in a dotted violin plot. They claim that knockout of TR β had no effect on APC proliferation, but isn't that because TR β doesn't significantly express in Myf5+ cells? There is already a paper showing that TR β is required for "beigeing" by thyroid hormones (Johann et al Cell Reports 2019). It would be interesting if, as the author points out, there is a different usage of the receptor for each thermogenic depot, with TR α being classical and TR β being beige. In this regard, the expression levels of each receptor should be shown for each subtype identified through scRNA-seq. A comparison with Burl et al (iWAT) would be also intriguing.

The authors conclude that the number of Ucp1+ cells is increasing because the expression level of UCP1 per mg tissue protein is decreasing (Figure 1e). Representative histology (H&E and UCP-IHC) and western blot results should be included in addition to the graph. In particular, morphological changes with the long-term T3 treatment should be described in detail.

Related to the above, is there a possibility that T3 alters the stability of UCP1 protein?

Recently, the creatine kinase responsible for futile creatine cycling was discovered in BAT (Rahbani et al Nature 2021). Is the expression of CKB regulated by short and long term T3 administration? It is also interesting to check Serca2b (Ikeda et al Nature Medicine 2017) as it is important for adipose tissue thermogenesis.

Kosaku Shinoda, Ph.D.

Albert Einstein College of Medicine

The Einstein-Mount Sinai Diabetes Research Center (ES-DRC)

Reviewer #2 (Remarks to the Author):

The manuscript by Liu et al. provides new information on the role of T3 and its receptors in brown adipose tissue neogenesis. The investigators provide data that T3 treatment restores proliferation in a model of chemically-induced hypothyroidism, and that this effect likely depends of THRA expression in stromal cells of the myf5 lineage and not in mature brown adipocytes. The authors additionally suggest that T3 promotes the transition of Pi16/Dpp4+ APCs into the adipogenic pathway and that proliferation involves mitotic expansion that depends on upregulation of myc and activation of glycolytic pathways.

This manuscript provides clear and convincing data that T3 promotes APC proliferation and potential differentiation compared to hypothyroid mice. Further, experiments that dissect cell types with Myf5-Cre and UCP-Cre nicely indicate a role of THRA in Myf5+ lineage vs mature adipocytes or THRB in the Myf5 lineage. These new data expand on previous work in rats and is of interest to the field of adipose tissue remodeling. However, there are a few concern regarding the generality of the results and (over) interpretation of the data.

1) Much of the results involve comparisons between chemically-induced hypothyroidism and induced hyperthyroidism. The systemic effects of hypothyroidism and T3 treatment complicate interpretation. The experiments with lineage-specific THRA addresses this concern somewhat, but not completely.

2) There little relevance to euthyroid conditions and normal physiology of BAT. For example, cold-induced neural activation leads to prominent increase in local T3 generation (via DIO2), and that T3 strongly regulates the sensitivity to adrenergic stimulation (which required for cold-induced proliferation). Do T3 effects require adrenergic activity; are effects observed at thermoneutral, or following sympathetic denervation?

3) The authors observe cellular expansion of both endothelial cells and immune cells, albeit somewhat less. There is a general concern about cell recovery and representation.

4) Data correlating T3 levels with APC in human sample is confounded by many factors, including age, and thus is not possible to interpret.

5) If each IBAT pad contains $>1 \times 10^5$ APCs, why were up to 30 mice pooled to obtain on 5×10^3 cells? Why was FACS used at all, since it greatly diminishes yield and greatly increases the time from dissection to cDNA synthesis. (What was the time between dissection and cDNA synthesis?). The whole point of scRNA is to avoid the use of FACS. The overall extremely low recovery raises questions regarding how representative the data are. This concern is magnified by the lack of histological verification.

6) Pseudotime analysis by Monocle will always produce a putative trajectory, which must be independently validated, for example by lineage tracing. The pseudotime trajectories of Figure 6 are highly speculative. It seems clear that the proliferating cluster is restricted to the committed preadipocyte cluster (group 2).

7) There are no data to show that myc is causally related, or a mechanistic connection to THRA signaling. Myc expression correlates with dozens of cell cycle genes.

Technical issues:

There is a general concern about cell recovery and representation. Collagenase digestion differentially releases cells and damages others (especially adipocytes). Brown adipose tissues contains at least 5 times more SVC than adipocytes, which is not reflected in the counts in figure 1G.

EdU is not an immunofluorescent technique. EdU tracing could have been performed using standard antibodies like PDGFRA, DPP4, ICAM etc. that would identify cell types in situ and avoid concerns about differential isolation. Fate mapping of flash-labeled cells could also be performed in vivo.

Skin temperature is not a direct measure of BAT thermogenesis, especially when comparing hypo- vs hyperthyroid mice.

The authors should consider that the gene profiles of DPP4+ (and ICAM+) cells is dramatically altered by isolation and growth on plastic in the presence of FBS, not to mention the effects of adipogenic induction. It is unclear why focus on these cells, since the scRNA data indicate they do not proliferated in response to T3.

Reviewer #3 (Remarks to the Author):

Ying and colleagues have investigated the effect of T3 on the recruitment capacity of interscapular BAT. They show that T3 increases BAT hyperplastic growth by promoting APCs proliferation. This effect is dependent on TRalpha (but not TRbeta) presence. They finally show that APCs is a heterogeneous population and that T3 drives adipogenesis and mitosis involving the action of Myc on glycolysis.

This study provides interesting and compelling evidence on the actions of T3 on BAT. The data are of interest and mainly support the conclusions. The experimental approach, which combines physiology, metabolic phenotyping, animal models and omics, is superb. Some points need to be addressed.

1. The data of MTRalpha KO are clear and the involvement of TRalpha in Myf5 lineage is demonstrated and out of question. However, MTRalpha KO still show responses to T3 on thermogenesis, tissue expansion and APC proliferation. The authors demonstrate that neither TRbeta nor TRalpha in matured (UCP1 expressing) adipocytes are involved in that effect. However, that question remains unsolved and it is of interest. One possibility could be a central action of T3 on the hypothalamus, mediated by the SNS, a mechanism that is demanding to address in the present study. This can be reached easily by several ways, for example by giving T3 centrally in euthyroid or hypothyroid mice (ICV or within the VMH) and also by examining the effect of SNS denervation or beta3 blockage (using SR59230A) in MTRalpha KO mice. This should be investigated, since it will really strengthen the study at a highest level. Also the key papers on central actions of T3 on BAT should be quoted.

2. While of 2DG experiments are of interest, mechanistically do not add to much value. This kind of top study would be benefic of a most sophisticated approach, for example by genetic targeting of glycolysis. I would not ask about generating new mice strains, since that it would be out of the scope, but a virogenetic approach (targeting key glycolytic enzymes) would be easily to perform.

3. IRT pictures must be improved. FLIR cameras (as the one used) provide several color ways to display images, another option should be chosen. Details of the model of the camera, landmarks for the analysis (was it squared? as shown), temperature selected (average, maximal etc.) and emissivity (exact value must be specified) are needed.

4. Page 1 Introduction/line 81: "... (iBAT), a classic brown fat depot [13]..."

Reviewer #1 (Remarks to the Author):

Drs. Jiang and Ying's group took a closer look at the effect of thyroid hormones on the proliferation of APC in interscapular "classical" BAT. It has been known for many years that thyroid hormones positively regulate adipose tissue thermogenesis (e.g. Johann et al Cell Reports 2019), and although this is not novel per se, the phenotypic analysis of tissue-specific receptor knockout using two Cre models (Myf5 and Ucp1) and the detailed investigation of the effect of T3 by single cell analysis are important developments in the field. If the following requirements are met, I recommend publication in Nature Communications.

1. On the analysis of human neck brown fat. It is known that human supraclavicular BAT resembles to mouse "recruitable" beige adipose tissue from studies by several groups (e.g. Shinoda et al, Nature Medicine 2015). Thus, it is inappropriate to directly apply the authors' findings from mouse using "classical" BAT to human neck brown fat. In addition, CD90 is used to isolate APCs from human, but isn't CD140A (PDGFR α) more common (Oguri et al Cell 2020)? Are CD90 and CD140A expressed in the same population?

Response to the Reviewer's Comments: We agree with the reviewer that it has been suggested that adult human supraclavicular BAT resembles murine beige fat much more closely than it resembles murine classic BAT, cautioning the extrapolation of rodent studies to humans [1]. However, whether APCs resided in adult human BAT are similar to or different from APCs resided in murine beige fat and classic BAT has not been extensively studied. We compared our scRNA-seq data (APCs from mouse iBAT (m-iBAT)) with those reported by Burl et al. (ASC 1, ASC 2, and Prolif. from mouse iWAT (m-iWAT); ASC 1, ASC 2, Pro. ASC and Diff. ASC from eWAT (m-eWAT)) [2], and Sun et al. (preadipocytes from human deep-neck fat (h-dnFAT)) [3] (Supplementary Fig. 9h,i). Correlation of global expression signature of preadipocytes from h-dnFAT and APCs of three types of adipocytes from mice was measured by spearman rank correlation coefficient (CC). As expected, the highest CC was observed between m-iWAT and m-eWAT (CC: 0.963). Interestingly, we found that the preadipocytes from h-dnFAT were more closely related to APCs from m-iBAT (CC: 0.61) than APCs from m-iWAT (CC: 0.569) and m-eWAT (CC: 0.565) (Supplementary Fig. 9h). According to a recently developed method to determine cross-species similarities based on scRNA-seq data [4, 5], we integrated the 4 datasets and assigned 11 cell subpopulations. Cross-species pairwise correlation analysis of matched subpopulations revealed a higher correlation between preadipocytes from h-dnFAT and APCs from m-iBAT (Supplementary Fig. 9i). However, as the experimental condition and methodology vary in different studies, further analyses are required to clarify the nature of APCs in adult human BAT and reveal whether there are functional differences between APCs in adult human BAT and other APCs derived from different murine fat depots. We removed the data related to human neck brown fat from the main figure, provided them as additional data (Supplementary Fig. 9j,k) and revised our manuscript accordingly.

We agree with the reviewer that CD140a (PDGFR α) had been used to isolate APCs from human samples in previous studies [6]. In fact, when we searched the literatures, we noticed that CD90 was more frequently used. We listed the most cited paper, a recently published paper, and a review paper regarding the usage of CD90 to isolate APCs for human samples as reference here [7-9]. As the reviewer suggested, we repeated the experiment by using freshly isolated human fat and determined the expression pattern of CD90 and CD140a. After excluding immune cells and endothelial cells,

the remaining cells were analyzed for CD90 and CD140a expression. We found that almost all (99%) cells expressing CD90 were CD140a positive, whereas about 88% of cells expressing CD140a were CD90 positive, suggesting that CD90 and CD140a are normally co-expressed in the same population (Supporting data Fig. 1a).

2. Regarding the analysis of single cell RNA-seq. The authors should show the mRNA levels of TR α /TR β and Myf5, the driver of Cre, in a dotted violin plot. They claim that knockout of TR β had no effect on APC proliferation, but isn't that because TR β doesn't significantly express in Myf5⁺ cells? There is already a paper showing that TR β is required for "beigeing" by thyroid hormones (Johann et al Cell Reports 2019). It would be interesting if, as the author points out, there is a different usage of the receptor for each thermogenic depot, with TR α being classical and TR β being beige. In this regard, the expression levels of each receptor should be shown for each subtype identified through scRNA-seq. A comparison with Burl et al (iWAT) would be also intriguing.

Response to the Reviewer's Comments: As suggested, we analyzed the scRNA-seq data and provided the data showing the mRNA levels of TR α and TR β in APCs from iBAT in a dotted violin plot (Supplementary Fig. 5g). The expression of Myf5 was not detected in our scRNA-seq data, which is in line with the notion that APCs lose the expression of Myf5 [10], although they are developed from the Myf5⁺ lineage. Consistent with the data in Fig. 3a, the mRNA levels of TR α were 9 times higher than those of TR β in APCs from iBAT according to the analysis of our scRNA-seq data (Supplementary Fig. 5g-i). These results suggest that TR α is the major TR isoform in APCs resided in iBAT, which might play a more important role than TR β in these APCs.

Regarding the role of TR β in TH-induced beigeing/browning, current evidence based on the studies using global TR β KO mice and TR β agonist GC-1 suggests that TR β plays a critical role in the browning of WAT [11, 12]. Our unpublished data recently obtained in mice with Adiponectin-Cre-mediated adipocyte-specific deletion of TR β (ATR β KO mice) also support the notion that adipocyte TR β is required for the browning induced by TH (Supporting data Fig. 1b). However, whether TR α is required for the WAT browning is currently unknown. Actually, most previous studies focused on the TH effect on the expression of thermogenic genes in beige fat, whether TH plays a role in APCs resided in WAT during browning and whether the effect of TH on these APCs contributes to the processes of browning remain to be clarified [13]. As suggested, we analyzed the scRNA-seq data reported by Burl et al. and found that the mRNA levels of TR α were 18 times and 14 times higher than those of TR β in the APCs resided in iWAT and eWAT, respectively, suggesting that TR α might also be critically involved in the regulation of APCs resided in WAT (Supplementary Fig. 5g-i).

It is also worth noting that the mRNA levels of TR α and TR β are similar in mature adipocytes differentiated from iBAT-derived SVF, suggesting that both TR α and TR β are important for the function of mature adipocytes (Fig. 3a). Based on our current knowledge and findings, we speculate that TR α might be more important for proliferation than TR β in adipose progenitor cells, while both TR α and TR β might be important for adipocyte maturation and function. Given that we do not have data showing whether TR α is critical for the beigeing, for the time being, we decide not to conclude that there is a different usage of the receptor for each thermogenic depot, with TR α being classic and TR β being beige in this manuscript. We believe that, after we explore the role of TR α and TR β in either APCs or mature adipocytes in different fat depots, we will provide more evidence showing whether TR α and TR β play a specific or redundant role in certain cell types, which may strategize

future therapeutic interventions by targeting specific TR isoform in certain cell types.

3. The authors conclude that the number of Ucp1⁺ cells is increasing because the expression level of UCP1 per mg tissue protein is decreasing (Figure 1e). Representative histology (H&E and UCP-IHC) and western blot results should be included in addition to the graph. In particular, morphological changes with the long-term T3 treatment should be described in detail.

Response to the Reviewer's Comments: Thanks for the reviewer's comment. We did not make the conclusion that the number of Ucp1⁺ cells were increasing according to the observation that UCP1 mRNA levels were decreasing after long-term T3 treatment (the right panel in Supplementary Fig. 1d, which was submitted as Supplementary Fig. 1b in previous version) or the expression level of UCP1 per mg tissue protein was decreasing (actually it was increasing as shown in Fig. 1d). In fact, we speculate that the number of Ucp1-expressing cells per depot was increasing after long-term T3 treatment based on the fact that the UCP1 protein levels per mg tissue protein were not further increased by long-term T3 treatment compared to short-term treatment (Fig. 1d), meanwhile the total amount of UCP1 protein per depot was further increased by long-term T3 treatment (Fig. 1e). We have revised the manuscript to avoid this confusion. As suggested, we provided representative H&E and UCP1-IHC staining and western blot results (Supplementary Fig. 1a-c) and described morphological changes in our revised manuscript (Page 4, Line 112-117) and also as follows. H&E staining of iBAT sections revealed that the lipid droplets maintain their typical multilocular appearance after either short-term or long-term T3 treatment. Consistent with a role for T3 in promoting lipid utilization in brown adipocytes [14], the size of lipid droplet was slightly decreased after short-term T3 treatment. In agreement with a role for T3 in driving systemic fat mobilization and lipogenesis in brown adipocytes [15], lipid droplets became larger after long-term T3 treatment, which might partially contribute to the enlargement of iBAT. It has been suggested that, when UCP1 levels are saturated in fully differentiated brown adipocytes, the increase in thermogenic capacity eventually results from acquisition of more brown adipocytes by promoting the proliferation and subsequent differentiation of APCs within BAT depots [16, 17]. As we found that T3 could induce BAT hyperplasia by promoting the APC proliferation (Fig. 1f-k, 2a-k, etc.), we mainly focused on the effects of TH on APCs in the iBAT depot in this study.

4. Related to the above, is there a possibility that T3 alters the stability of UCP1 protein?

Response to the Reviewer's Comments: To test this possibility, we treated the iBAT-SVF-derived mature adipocytes with T3 in the presence of cycloheximide (CHX), a protein synthesis inhibitor, and found that T3 treatment might not alter the stability of UCP1 protein in brown adipocytes (Supplementary Fig. 1e and Supporting data Fig. 1c).

5. Recently, the creatine kinase responsible for futile creatine cycling was discovered in BAT (Rahbani et al Nature 2021). Is the expression of CKB regulated by short and long term T3 administration? It is also interesting to check Serca2b (Ikeda et al Nature Medicine 2017) as it is important for adipose tissue thermogenesis.

Response to the Reviewer's Comments: As the reviewer suggested, we analyzed the expression of creatine kinase (Ckb) and Serca2b (Atp2a2). We found that the mRNA levels of Ckb were not altered after short-term T3 treatment but increased after long-term T3 administration (Supporting data Fig. 1d), while the protein levels of Ckb were markedly elevated after both short-term and long-

term T3 treatment (Supporting data Fig. 1e). The mRNA levels of Serca2b were not altered after either short-term or long-term T3 treatment (Supporting data Fig. 1f). Unfortunately, because we could not purchase the Serca2b antibody from Thermo Fisher (2A7-A1) due to customs restrictions, we do not know whether the Serca2b protein levels can be regulated by T3 or not. We also analyzed the expression of Ckb and Serca2b in the APCs of iBAT based on our scRNA-seq data and found that both Ckb and Serca2b (Atp2a2) expression are expressed in all five groups identified in this study (Supporting data Fig. 1g,h). The expression of Ckb was decreased (Ckb: p_val_adj=3.52E-17, avg_logFC=-0.08), while the expression of Serca2b (Atp2a2) was increased (Serca2b: p_val_adj=8.82E-137, avg_logFC=0.22) in these APCs after long-term T3 treatment (Supporting data Fig. 1i,j). Although the roles of Ckb and Serca2b (Atp2a2) in thermogenesis have been reported [18, 19], whether they are able to regulate thermogenesis through their actions in APCs resided in iBAT requires further investigation.

Reviewer #2 (Remarks to the Author):

The manuscript by Liu et al. provides new information on the role of T3 and its receptors in brown adipose tissue neogenesis. The investigators provide data that T3 treatment restores proliferation in a model of chemically-induced hypothyroidism, and that this effect likely depends of THRA expression in stromal cells of the myf5 lineage and not in mature brown adipocytes. The authors additionally suggest that T3 promotes the transition of Pi16/Dpp4+ APCs into the adipogenic pathway and that proliferation involves mitotic expansion that depends on upregulation of myc and activation of glycolytic pathways.

This manuscript provides clear and convincing data that T3 promotes APC proliferation and potential differentiation compared to hypothyroid mice. Further, experiments that dissect cell types with Myf5-Cre and UCP-Cre nicely indicate a role of THRA in Myf5+ lineage vs mature adipocytes or THRB in the Myf5 lineage. These new data expand on previous work in rats and is of interest to the field of adipose tissue remodeling. However, there are a few concerns regarding the generality of the results and (over) interpretation of the data.

1. Much of the results involve comparisons between chemically-induced hypothyroidism and induced hyperthyroidism. The systemic effects of hypothyroidism and T3 treatment complicate interpretation. The experiments with lineage-specific THRA addresses this concern somewhat, but not completely.

Response to the Reviewer's Comments: We totally agree with the reviewer that systemic effects of hypothyroidism and T3 treatment may complicate interpretation. In current study, mice were rendered hypothyroid by the addition of MMI and NaClO₄ in their drinking water first. Then hypothyroid mice with T3 treatment were compared directly to hypothyroid mice without T3 treatment to minimize the effects of potential confounding factors. In this research field, as the expression of TR isoforms is usually tissue-specific, an alternative approach to delineate the tissue-specific T3 action is to investigate the role of major TR isoform in certain tissue types. For instance, as TR β is the major TR isoform in liver, we can investigate the role of hepatic TR β by using liver-targeted and/or TR β -selective agonist (MB07811 or GC-1). Given that TR α -selective agonist or APC-targeted approach has not yet been developed, to study the TR-mediated TH action in APCs resided in iBAT, we applied the Cre/LoxP system to delete the endogenous TR α or TR β in APCs developed from Myf5⁺ lineage.

To avoid systemic effects that may complicate interpretation, we employed an approach developed recently in our lab by using rolling microneedles for local delivery of T3 to the iBAT region. The tissue-specific T3 delivery was evaluated by using TH action indicator (THAI) mouse harboring a TH-responsive luciferase reporting system developed previously [20]. Consistent with the published data, i.p. injection of T3 could elevate the luciferase activity in T3 target tissues, including iBAT, iWAT, liver, and brain (Supporting data Fig. 2a). In contrast, local injection of T3 into the iBAT region for 5 days only increased the luciferase activity in iBAT but not in other T3 target tissues, suggesting that this rolling microneedle-based approach could achieve an “iBAT-specific” T3 administration (Supplementary Fig. 2q). In line with the data obtained in mice with i.p. injection of T3, we found that “iBAT-specific” T3 treatment could also increase the iBAT mass and the number of either SVF cells or APCs in the iBAT depot (Supplementary Fig. 2r). Although we cannot exclude the possibility that other factors or other tissues might also be involved in the regulation of APC proliferation by the elevation of circulating T3 levels after T3 i.p. injection, our new results together with our previous *in vivo* and *in vitro* data suggest that T3 is able to target the APCs resided in iBAT depot directly and promote the cell proliferation in a cell-autonomous manner.

2) There little relevance to euthyroid conditions and normal physiology of BAT. For example, cold-induced neural activation leads to prominent increase in local T3 generation (via DIO2), and that T3 strongly regulates the sensitivity to adrenergic stimulation (which required for cold-induced proliferation). Do T3 effects require adrenergic activity; are effects observed at thermoneutral, or following sympathetic denervation?

Response to the Reviewer's Comments: We agree with the reviewer that cold-induced neural activation increases the local T3 generation and T3 regulates the sensitivity to adrenergic stimulation [21-24]. However, whether the regulation of APC proliferation by T3 observed in this study requires adrenergic activity had not been investigated. As suggested, to test whether the T3 effect on APC proliferation requires adrenergic activity, we performed the experiments under a thermoneutral condition or by using adrenergic receptor (AR) antagonists. We found that i.p. injection of T3 could increase the iBAT mass, number of APCs, and percentage of APCs in iBAT depot of euthyroid mice under a thermoneutral condition (30°C) in which thermal stress was eliminated (Supplementary Fig. 2j). We also compared the T3 effect between MMI mice housed at room temperature (RT) and thermoneutral temperature (30°C). Consistently, the effect of T3 on the number and percentage of APCs could be observed either at RT, which is a sub-thermoneutral temperature, or at 30°C (Supplementary Fig. 2k). We noticed a small but significant reduction in the number and percentage of APCs in T3-treated MMI mice at RT as compared to those T3-treated MMI mice at 30°C (Supplementary Fig. 2k). Given that the cold-induced thermogenesis is nullified and the sympathetic drive to BAT is minimal at thermoneutrality, these results not only indicate that the cold-induced sympathetic activation might be dispensable for the regulation of APC proliferation by T3, but also suggest that the sympathetic activation might be able to potentiate the action of T3 and might be required for T3 to achieve its maximal effect on APC proliferation in iBAT.

As surgical denervation would affect nonsympathetic nerves, any other efferent nerves, and afferent sensory nerves running in the same nerve bundles, to further explore the involvement of sympathetic activity, we employed SR59230A (SR), a selective β_3 -AR antagonist, and metoprolol (Met), a selective β_1 -AR antagonist. We found that the effect of i.p. injection of T3 on APC proliferation in iBAT could be observed after either SR or Met treatment (Supplementary Fig. 2l,m). Decreases

could also be detected in the number and percentage of APCs in the iBAT of T3-treated MMI mice with either SR or Met treatment as compared to those in the iBAT of T3-treated MMI mice without SR or Met treatment (Supplementary Fig. 2l,m). These results suggest that, although either the β 3 or β 1-AR-mediated sympathetic activation might be dispensable for the regulation of APC proliferation by T3, both of them might be capable of potentiating the T3 action and might be required for T3 to achieve its maximal effect on APC proliferation in iBAT. Given that β 1-AR (*Adrb1*) and β 3-AR (*Adrb3*) have differential expression pattern (Supporting data Fig. 2b) and distinct functions in adipocyte progenitor and mature adipocytes, respectively [25-27], these results imply that both β 1 and β 3 adrenergic signaling are involved in the regulation of APC proliferation by T3 but the underlying mechanisms differ. As T3 regulates APC proliferation in a cell-autonomous manner (Fig. 2l and Supplementary Fig. 2i) and either β 3 or β 1-AR-mediated sympathetic activation is not required for the regulation of APC proliferation by T3, we speculate that either β 1 or β 3 adrenergic signaling contributes to the profound effect of T3 on APC proliferation but both of them are not indispensable in iBAT.

As we know, the *Dio2* expression is enriched in mature brown adipocytes as compared to that in brown preadipocytes before differentiation [28]. We also observed that the mRNA levels of *Dio2* was gradually increased during the differentiation of iBAT-derived SVF (Supporting data Fig. 2c). Accordingly, we found that the abundance of *Dio2* was low in APCs (Supplementary Fig. 5j and Supporting data Fig. 2d). These results indicate that cold-induced neural activation might increase the T3 generation in mature adipocytes via *Dio2*, leading to an elevation of local T3 levels, thereby promoting the APC proliferation within the iBAT depot. As hypothyroid mice had fewer APCs as compared to euthyroid mice (Supplementary Fig. 2g), we speculate that, at least at room temperature, normal TH levels are required for maintaining the stem cell pool. Therefore, we propose that a euthyroid condition is critical for the normal physiology of BAT. We also would like to point out that, in this study, we employed euthyroid or hypothyroid mice with or without T3 treatment as a model to investigate the regulation of APC proliferation by T3, which has shed new light on the physiological role of T3, which are produced by mature brown adipocytes upon cold stress, in regulating the recruitment process of BAT. In our revised manuscript, we provided more discussion based on our above newly obtained data (Page 22, Line 759-775).

3) The authors observe cellular expansion of both endothelial cells and immune cells, albeit somewhat less. There is a general concern about cell recovery and representation.

Response to the Reviewer's Comments: Thanks for the reviewer's comments. The recovery efficiency of immune cells, endothelial cells, and APCs were expected to be the same for both experimental group and control group since tissues from experimental group and control group were processed side by side using the same protocol. It is also worth noting that 3-5 biological repeats from experimental group and control group respectively were used to calculate the statistical difference of numbers or proportions of immune cells, endothelial cells, and APCs in adipose tissues, taking processing-caused variance from different samples into account. Thus, differential recovery efficiency is less likely to be a confounding factor for comparing the numbers or percentages of immune cells, endothelial cells, and APCs in fat tissues between the two groups. It is also known that, although the procedure for dissociation of adipose tissue has been improved, it still suffers from some inherent limitations. A highly effective dissociation with good biocompatibility, i.e., maintaining high cell viability and functionality after digestion and purification is still a huge

challenge. As the protocols published previously were normally used for WAT but not BAT, we modified the broadly used protocols accordingly in order to achieve high cell recovery rate. Our flow cytometry analysis revealed that we could obtain ~15% immune cells, ~61% endothelial cells, and ~17% APCs in live SVF cells from one fat pad of iBAT (Supporting data Fig. 2e), which is in agreement with a previous finding reported by Steinbring et al (Supporting data Fig. 2e) [29].

4) Data correlating T3 levels with APC in human sample is confounded by many factors, including age, and thus is not possible to interpret.

Response to the Reviewer's Comments: We agree the reviewer that many factors can confound the data interpretation. As the reviewer suggested, we removed the data related to human neck brown fat from main figure, provided them as additional data (Supplementary Fig. 9j,k), and revised our manuscript accordingly.

5) If each iBAT pad contains $>1 \times 10^5$ APCs, why were up to 30 mice pooled to obtain on 5×10^3 cells? Why was FACS used at all, since it greatly diminishes yield and greatly increases the time from dissection to cDNA synthesis. (What was the time between dissection and cDNA synthesis?). The whole point of scRNA is to avoid the use of FACS. The overall extremely low recovery raises questions regarding how representative the data are. This concern is magnified by the lack of histological verification.

Response to the Reviewer's Comments: Thanks for the comments. Sorry for the confusion caused. The number of APCs per iBAT pad was determined by the number of SVF cells which was counted immediately after digestion and the percentage of APCs in live SVF cells was measured by flow cytometry analysis. As for sample preparation for scRNA-seq, it is recommended by the staff of our core facility that the optimal cell concentration is in the range of ~800 to 1,000 cells/ μ l. Although only 2.5 to 15 μ l cell suspension will be used for library construction, it is recommended by either the Cell Preparation Guide from 10x Genomics Single Cell Protocols or the facility staff that the cell suspension volumes should be larger than 50 μ l. Therefore, we have to prepare 5×10^4 cells for each sample. Cell loss is inevitable from dissociation, washing, resuspension, and sorting steps. Given that MMI mice have fewer APCs and the cell recovery efficiency vary in each experiment, we used 30 MMI mice in this study and obtained cell suspension with at a concentration of 1000 cell/ μ l in a total volume of 79 μ l. After strict quality control to filter low quality cells, 5,513 high-quality cells in MMI group were obtained for downstream analysis.

Regarding the use of FACS, we agree with the reviewer that, although the current FACS has a quite accurate cell sorting capability, it still suffers from some inherent limitations. Given that the percentages of endothelial cells and immune cells add up to more than 60% in the SVF of iBAT, since we focus on the regulation of APCs by T3 in this study, we used FACS to enrich APCs to better understand the heterogeneity of APCs. As a matter of fact, FACS-sorted cells, including adipocyte SVF and satellite cells etc., have been frequently used for scRNA-seq to address specific scientific questions or analyze cellular heterogeneity [30-33], including identifying new subsets and developmental trajectories of adipose-resident immune cells and revealing the heterogeneous nature of human satellite cells, etc.

Regarding the recovery rate, as the available protocols were normally used for WAT but not BAT, we modified the broadly used protocols accordingly in order to achieve high cell recovery rate of

APCs. As we mentioned above (Response to the Reviewer's Comments #3:), our flow cytometry analysis revealed that we could obtain ~15% immune cells, ~61% endothelial cells, and ~17% APCs in live SVF cells from one fat pad of iBAT (Supporting data Fig. 2e), which is in agreement with a previous finding reported by Steinbring et al (Supporting data Fig. 2e) [29].

Regarding how representative the data are, despite differences in the utilized methodologies for cell isolation, given that the majority of iBAT-derived APC subpopulations described in our study represent similar or highly overlapping subpopulations to those APC subpopulations identified in WAT-derived progenitors in previously published works from Burl et al. [2], in which magnetic beads were used for cell isolation, and Schwalie et al. [34] and Hepler et al. [35], in which FACS were employed for cell isolation, we believe that our strategy to obtain APCs by FACS isolation is reliable, although it might have limitations.

6) Pseudotime analysis by Monocle will always produce a putative trajectory, which must be independently validated, for example by lineage tracing. The pseudotime trajectories of Figure 6 are highly speculative. It seems clear that the proliferating cluster is restricted to the committed preadipocyte cluster (group 2).

Response to the Reviewer's Comments: We agree with the reviewer that pseudotime trajectories of APCs shown in Fig. 6 are speculative, which needs validation. We transplanted the iBAT-derived tdTomato-expressing APCs into the iBAT depot of mice. BODIPY staining of the transplants after 2 weeks revealed the development of tdTomato⁺ mature adipocytes (Supplementary Fig. 7a), suggesting that sort-purified APCs from the iBAT of tdTomato⁺ donor mice are capable of giving rise to mature adipocytes *in vivo*. We also transplanted the tdTomato-expressing Dpp4⁺ APCs and Icam1⁺ APCs into the iBAT depot, respectively, for indicated period of time. Immunofluorescent staining of the transplants revealed the development of tdTomato⁺ mature adipocytes from either Dpp4⁺ APCs or Icam1⁺ APCs, suggesting that both sort-purified Dpp4⁺ APCs and Icam1⁺ APCs can give rise to mature adipocytes *in vivo* (Fig. 7a and Supplementary Fig. 7c,d). In agreement with our notion that Dpp4⁺ APCs are more primitive undifferentiated APCs, while Icam1⁺ APCs are more committed preadipocytes, more mature adipocytes were observed from transplanted Icam1⁺ cells at 20 days post-transplantation (Supplementary Fig. 7c) and more mature adipocytes with larger lipid droplets were detected from transplanted Icam1⁺ cells after 30 days of cell transplantation (Supplementary Fig. 7d) as compared to those from transplanted Dpp4⁺ APCs at corresponding time points. Similar results could be obtained in cultured Dpp4⁺ APCs and Icam1⁺ APCs derived from the iBAT depot under minimal adipogenic conditions *in vitro*. The Icam1⁺ cells-derived adipocytes tended to have larger lipid droplets than those derived from of Dpp4⁺ cells (Fig. 7b). Accordingly, differentiated Icam1⁺ cells displayed higher levels of adipocyte-specific genes than those in differentiated Dpp4⁺ cells (Fig. 7c). The findings that cultured Icam1⁺ APCs differentiated efficiently into adipocytes, while cultured Dpp4⁺ APCs displayed relatively low adipogenic competence (Fig. 7b,c), further support the notion that Icam1⁺ cells might be more committed preadipocytes.

To further determine the *in vivo* fate of Dpp4⁺ or Icam1⁺ APCs, we transplanted iBAT-derived tdTomato-expressing Dpp4⁺ or Icam1⁺ APCs into the iBAT depot of mice. Flow cytometry analysis of APCs from recipient mice showed that tdTomato⁺Dpp4⁺ cells acquired Icam1 expression as early as day 10 post-transplantation (Fig. 7d). Meanwhile, a subset of transplanted cells started to down-

regulate Dpp4 expression when they acquired Icam1 expression (Fig. 7d). In contrast, transplanted tdTomato⁺Icam1⁺ cells did not produce substantial numbers of Dpp4⁺ cells (Fig. 7e), suggesting that Dpp4⁺ APCs might serve as progenitors for Icam1⁺ APCs, which would further differentiate into mature adipocytes. Consistently, we found that the percentages of Dpp4⁺ cells (Fig. 7f, left) and Icam1⁺ cells (Fig. 7f, right) in tdTomato⁺ cells from Dpp4⁺ APC-derived implants were gradually decreased and increased, respectively, while the percentage of Icam1⁺ cells in tdTomato⁺ cells from Icam1⁺ APC-derived implants remained unaltered after transplantation (Fig. 7g). Collectively, these results support the hypothesis based on our pseudotemporal trajectory analysis (Fig. 6) that Dpp4⁺ progenitors are more stem-like progenitors and Dpp4⁺ progenitors are able to adopt an adipogenic fate, giving rise to more committed Icam1⁺ progenitors.

Regarding the proliferating group, based on our findings, we speculate that it is not restricted to the committed preadipocyte group (group 2). As we mentioned in our manuscript, cells spend most of their time in interphase (G1, S and G2 phases) and cycle through interphase and then M phase to proliferate. The cell cycle phase scoring (Fig. 5l) revealed a similar cell cycle phase distribution pattern among the other four groups (Group 1-4). Notably, Group 1-4 cells were in different cell cycle phases and most of them were in G0/G1 and S phases, suggesting that a considerable number of cells in Group 1-4 are preparing for division including replicating DNA (Fig. 5l). Group 5 cells were enriched for cell cycle- or mitosis-related genes (Fig. 5k, Supplementary Fig. 5c and Supplementary Table 2,3), suggesting that Group 5 cells are actively undergoing mitotic division. The results of cell cycle phase scoring and calculation of the total mRNA molecules per cell further suggest that Group 5 cells are cycling cells in G2/M phase (Fig. 5l,m). In line with these data, segregation of the aggregated UMAP plot according to the treatment condition demonstrated that the cell numbers of different APC subpopulations (Group 1-5) were all increased after T3 treatment (Fig. 8a and Supplementary Fig. 8a), suggesting that all APC subpopulations might be proliferative in response to T3 treatment. Accordingly, EdU⁺Dpp4⁺ cells could be detected in the iBAT depot after T3 treatment (Supplementary Fig. 8b). Moreover, by performing BrdU labeling and FACS analysis, we found increases in the percentage of BrdU⁺ cells and Ki67 staining intensity in not only Dpp4⁺ APCs (Group 1 cells) but also Icam1⁺ APCs (Group 2 cells) from the iBAT of mice after T3 treatment, indicating they were both proliferative (Supplementary Fig. 8c-f). Although we did not have any corresponding evidence for Group 3-5 due to the lacking of antibodies to distinguish these subpopulations *in vivo*, based on our results in this study, we speculate that Group 1-4 cells in interphase are preparing for division; Group 1 and Group 2 cells are both proliferative, while Group 2 cells are about to enter into mitotic phase; Group 5 cells are undergoing division. Further investigation is required in the future to substantiate our hypothesis.

7) There are no data to show that myc is causally related, or a mechanistic connection to THRA signaling. Myc expression correlates with dozens of cell cycle genes.

Response to the Reviewer's Comments: Thanks for the reviewer's comments. The regulation of Myc by T3 during *Xenopus* metamorphosis has been reported in a JBC paper by Yunbo Shi's lab at NICHD [36]. We tested whether T3 could control the Myc expression directly at the transcription level in mammalian cells. Time-dependent elevation of Myc mRNA levels by T3 treatment could be observed in brown adipocyte precursor cells (Supplementary Fig. 8k). The mRNA levels of Myc were decreased in either cultured APC-enriched SVF cells or APCs from the iBAT of MTR α KO (Fig. 8j). To test whether TR regulates the Myc expression through direct binding to its promoter,

we searched for T3 response elements (TREs) in the promoter region of mouse *Myc* gene (up to 6-kb upstream of the start site) and found multiple putative sites, including one site (site 3) conserved between mouse and human (Supplementary Fig. 8l). To further test which specific promoter regions might be responsible to the recruitment of TR α , we performed ChIP-PCR (Supplementary Fig. 8l,m) and found that sites 3-7 but not sites 1 and 2 might be responsible to the recruitment of TR α (Fig. 8k). Accordingly, luciferase assay showed that T3 could stimulate the activity of the reporter containing 6-kb mouse *Myc* promoter fragment (Supplementary Fig. 8n). Collectively, these results indicate that *Myc* is a direct target gene of T3 and T3 may control glycolysis by direct regulating the transcription of *Myc* in the APCs of iBAT.

As suggested, to test whether *Myc* is critically involved in the regulation of cell cycle progression by T3 treatment, we transfected si-*Myc* into the brown adipocyte precursor cells. In agreement with the finding reported in a PNAS paper that *Ccnb1* is a direct target gene of *Myc* [37], we found that the upregulation of *Ccnb1*, which controls cell cycle G2/M transition, by T3 treatment was attenuated after *Myc* expression was reduced, suggesting that *Myc* might mediate the T3 effect on the expression of *Ccnb1* (Supporting data Fig. 2f, left). In other words, these data indicate that T3 might increase *Ccnb1* expression by direct targeting *Myc*, thereby promoting the G2/M transition. In contrast, the expression of *Ccne1*, which is required for the entry into S phase, was not affected by either T3 treatment or *Myc* knockdown in brown adipocyte precursor cells (Supporting data Fig. 2f, right). Therefore, we speculate that T3 might potentiate the entry into S phase by increasing the *Myc* expression and *Myc*-controlled glycolysis, rather than directly regulating the transcription of all cell cycle genes, such as *Ccne1*. Therefore, we speculate that the mechanisms underlying the regulation of cell cycle by *Myc*-mediated T3 action includes, but not limited to the *Myc*-mediated transcriptional regulation.

Technical issues:

1) There is a general concern about cell recovery and representation. Collagenase digestion differentially releases cells and damages others (especially adipocytes). Brown adipose tissue contains at least 5 times more SVC than adipocytes, which is not reflected in the counts in figure 1G.

Response to the Reviewer's Comments: We agree with the reviewer that collagenase digestion differentially releases cells and damages others, especially the adipocytes. To dissociate the tissue into single cells and obtain more SVF cells from the tissue, we normally digest iBAT for 30-60 minutes depending on the size of minced tissue pieces (Fig. 1g, right panel, in which the iBAT was digested for 40 minutes). To avoid the damage to the mature adipocytes and obtain more precise number of mature adipocytes in iBAT, we digested the iBAT for only 20 minutes (previous version: Fig. 1g, left panel). In our revised manuscript, we provided corresponding results of these two experiments. As expected, less SVF cells were obtained when the tissues were digested for 20 minutes (Supplementary Fig. 1g, right panel), while less mature adipocytes were obtained when the tissues were digested for 40 minutes (revised version: Fig. 1g, left panel). Sorry for not describing the details of the experiment settings clearly. In our revised manuscript, we provided the data from the same experiment in one panel. As we mainly focus on progenitors in this study, we provided the data obtained using longer digestion time (40 minutes) in Fig. 1g, provided those obtained using shorter digestion time (20 minutes) in Supplementary Fig. 1g, and revised the results, legend, and

materials and methods section by providing more details accordingly.

Regarding the ratio of SVF cells to mature adipocytes, the percentage of adipocytes and non-adipocytes in iBAT varies among published reports, depending on the method used for digestion, counting, and calculation. One paper reported that non-adipocytes make up at least 50% of cells in the fat pad [38], while the other papers reported that the fraction of adipocytes in iBAT is 25-30% and ~65%, respectively [39, 40]. In our case, using Type 1 Collagenase from Sigma would achieve a better yield of SVF cells, especially APCs, than using Type 1 Collagenase from Roche (Supporting data Fig. 2g,h). Although differential effects on different cell types were observed here, we think the differences are small (Supporting data Fig. 2g,h).

2) EdU is not an immunofluorescent technique. EdU tracing could have been performed using standard antibodies like PDGFRA, DPP4, ICAM etc. that would identify cell types *in situ* and avoid concerns about differential isolation. Fate mapping of flash-labeled cells could also be performed *in vivo*.

Response to the Reviewer's Comments: Thanks for the reviewer for pointing out our mistakes when we described the results using EdU. We agree with the reviewer that EdU is not an immunofluorescent technique. We have corrected this sentence in our revised manuscript. As the reviewer suggested, we performed EdU tracing using antibodies for those markers mentioned above. Consistent with our previous results, both EdU-labeled PDGFR α ⁺ cells and EdU-labeled Dpp4⁺ cells could be observed *in situ* in iBAT depot (Supporting data Fig. 2i and Supplementary Fig. 8b), supporting the notion that T3 promotes the proliferation of APCs, including Dpp4⁺ APCs (Group 1 cells). Unfortunately, we could not obtain specific staining of Icam1 by using the antibody from Proteintech (Cat. FITC-60299). To obtain more evidence showing whether the subpopulations identified in this study are proliferative, we performed *in vivo* BrdU labeling followed by flow cytometry analysis. Both BrdU⁺Dpp4⁺ APCs and BrdU⁺Icam1⁺ APCs could be detected in the iBAT of mice. In agreement with our notion, T3 treatment could markedly increase the percentage of BrdU⁺ cells in either Dpp4⁺ APCs or Icam1⁺ APCs (Supplementary Fig. 8c,d). Accordingly, the Ki67 expression was elevated in both Dpp4⁺ APCs and Icam1⁺ APCs (Supplementary Fig. 8e,f).

Regarding the fate mapping, as PDGFR α is a marker for APC and expressed in all subpopulations identified in this study (Supplementary Fig. 5a), we could not perform PDGFR α -Cre-based fate mapping to trace the APC subpopulations, including Dpp4⁺ APCs or Icam1⁺ APCs. Since Dpp4-Cre and Icam1-Cre mice are currently not available, we may need to generate these mice to perform fate mapping in the future. To test whether Dpp4⁺ APCs can give rise to Icam1⁺ APCs *in vivo*, we performed cell transplantation studies. As mentioned above in Response to the reviewer's comment #6, we found that all FACS sorted-APCs, Dpp4⁺ APCs, and Icam1⁺ APCs formed mature adipocytes within the iBAT depot (Supplementary Fig. 7a,c,d). More mature adipocytes were observed from transplanted Icam1⁺ cells at 20 days post-transplantation (Supplementary Fig. 7c) and more mature adipocytes with larger lipid droplets were detected from transplanted Icam1⁺ cells at 30 days post-transplantation (Supplementary Fig. 7d) as compared to those from transplanted Dpp4⁺ cells at corresponding time points. Similar results could be obtained in cultured Dpp4⁺ APCs and Icam1⁺ APCs derived from iBAT under minimal adipogenic conditions *in vitro* (Fig. 7b,c).

As mentioned above in Response to the reviewer's comment #6, to further determine the *in vivo* fate of iBAT-derived Dpp4⁺ or Icam1⁺ APCs, we assessed the expression of Dpp4 and Icam1 in

tdTomato⁺ cells after cell transplantation. Flow cytometry analysis of tdTomato⁺ cells from recipient mice showed that tdTomato⁺ Dpp4⁺ cells acquired Icam1 expression as early as day 10 post-transplantation (Fig. 7d). Meanwhile, a subset of these transplanted cells started to down-regulate the Dpp4 expression when they acquired the Icam1 expression (Fig. 7d). In contrast, transplanted tdTomato⁺ Icam1⁺ cells did not produce substantial numbers of Dpp4⁺ cells (Fig. 7e), suggesting that Dpp4⁺ APCs might serve as progenitors for Icam1⁺ APCs, which would further differentiate into mature adipocytes. Consistently, we found that the percentages of Dpp4⁺ cells (Fig. 7f, left) and Icam1⁺ cells (Fig. 7f, right) in tdTomato⁺ cells from Dpp4⁺ APC-derived implants were gradually decreased and increased, respectively, while the percentage of Icam1⁺ cells in tdTomato⁺ cells from Icam1⁺ APC-derived implants remained unaltered after transplantation (Fig. 7g). Collectively, these results support the hypothesis based on our pseudotemporal trajectory analysis that Dpp4⁺ progenitors (Group 1 cells) are more stem-like progenitors and Dpp4⁺ progenitors are able to adopt an adipogenic fate, giving rise to more committed Icam1⁺ progenitors (Group 2 cells) (Fig. 6).

3) Skin temperature is not a direct measure of BAT thermogenesis, especially when comparing hypo- vs hyperthyroid mice.

Response to the Reviewer's Comments: Thanks for the reviewer's comments. Several nuclear imaging methods have been employed to measure BAT activity in rodents, such as PET/CT, fMRI, or 13C-MRS that provide estimates of adipose tissue metabolism [41-43]. However, these methods rely on the assumption that circulating substrate uptake (FDG-PET/CT, 13C-MRS) or blood flow (fMRI) reflects BAT activity. In our lab, PET/CT had been used to measure the thermogenesis after T3 (Supporting data Fig. 2j, left panel), however, as the Glut4 is a direct target gene of T3 [44] (Supporting data Fig. 2j, right panel), therefore, the PET/CT might not reflect the thermogenic capacity of the iBAT as the glucose uptake was regulated by T3 directly. In this study, we used infrared imaging that has been widely used to measure thermogenesis. Importantly, Crane et al have validated the use of infrared thermography as a means to assess BAT-derived thermogenesis [45]. Moreover, several studies have measured surface body temperature using infrared imaging in rodents to evaluate the thermogenic activity of iBAT under basal or stimulated conditions [46-49]. Additionally, as TH can regulate the thermogenesis in muscle and adipocytes, measurement of the rectal temperature and heat production was also not suitable for our study. Given that Crane et al provided evidence that the skin temperature can represent the BAT activity, we decided to use the skin temperature measured by infrared thermography in our study, especially when comparing hypo- vs hyperthyroid mice.

4) The authors should consider that the gene profiles of DPP4+ (and ICAM+) cells is dramatically altered by isolation and growth on plastic in the presence of FBS, not to mention the effects of adipogenic induction. It is unclear why focus on these cells, since the scRNA data indicate they do not proliferated in response to T3.

Response to the Reviewer's Comments: Thanks for the reviewer's comments. We agree with the reviewer that the expression profiles of stem cells would be altered by isolation and growth on plastic. Based on our current knowledge, we also believe that the culture environment or altered microenvironment makes the cell cultures are less physiologically relevant and cannot represent *in vivo* tissue well. For example, it is easily to understand why transporter or membrane-bound protein cannot be correctly located when cells are cultured *in vitro*. However, we also believe that certain

mechanisms underlying the transcriptional regulation might still exist after isolation and growth. In fact, the mRNA expression of Dpp4 and Icam1 could be detected by qPCR (Supporting data Fig. 2k), while the protein expression of Dpp4 and Icam1 could not be detected by FACS after isolation and growth on plastic. We also would like to point that, as evident by the noted difference in Cycle threshold (Ct) values, the mRNA expression of Dpp4 and Icam1 remained higher and lower, respectively, in cultured Dpp4⁺ cells, while the mRNA levels of Dpp4 and Icam1 remained lower and higher, respectively in cultured Icam1⁺ cells (Supporting data Fig. 2k), suggesting that the transcriptional regulation might not be totally disrupted findings and the findings based on these cultured cells might still remain physiologically relevant.

Regarding the proliferation issue, as described in detail in Response to the reviewer's comment #6 (the last paragraph), we speculate that proliferating cluster is not restricted to the committed preadipocyte group (group 2), indeed, Group 1-4 cells in interphase are preparing for division; Group 1 and Group 2 cells are both proliferative, while Group 2 cells are about to enter into mitotic phase; Group 5 cells are undergoing division. Further investigation is required in the future to substantiate our hypothesis.

Reviewer #3 (Remarks to the Author):

Ying and colleagues have investigated the effect of T3 on the recruitment capacity of interscapular BAT. They show that T3 increases BAT hyperplastic growth by promoting APCs proliferation. This effect is dependent on TRalpha (but not TRbeta) presence. They finally show that APCs is a heterogenous population and that T3 drives adipogenesis and mitosis involving the action of Myc on glycolysis.

This study provides interesting and compelling evidence on the actions of T3 on BAT. The data are of interest and mainly support the conclusions. The experimental approach, which combines physiology, metabolic phenotyping, animal models and omics, is superb. Some points need to be addressed.

1. The data of MTRalpha KO are clear and the involvement of TRalpha in Myf5 lineage is demonstrated and out of question. However, MTRalpha KO still show responses to T3 on thermogenesis, tissue expansion and APC proliferation. The authors demonstrate that neither TRbeta nor TRalpha in matured (UCP1 expressing) adipocytes are involved in that effect. However, that question remains unsolved and it is of interest. One possibility could be a central action of T3 on the hypothalamus, mediated by the SNS, a mechanism that is demanding to address in the present study. This can be reached easily by several ways, for example by giving T3 centrally in euthyroid or hypothyroid mice (ICV or within the VMH) and also by examining the effect of SNS denervation or beta3 blockage (using SR59230A) in MTRalpha KO mice. This should be investigated, since it will really strengthen the study at a highest level. Also the key papers on central actions of T3 on BAT should be quoted.

Response to the Reviewer's Comments: Thanks for the comments and suggestion. To address these issues, as suggested, we performed ICV injection of T3. Consisted with previous reports [48, 50, 51], ICV injection of T3 for 5 days increase the size of brown adipocytes in iBAT depot and decrease the size of white adipocytes in iWAT (Supplementary Fig. 2n). Moreover, increased UCP1 and tyrosine hydroxylase protein expression could be observed in both iBAT and iWAT after ICV

injection of T3 (Supplementary Fig. 2o). These results suggest that the ICV injection of T3 was successful. Interestingly, we found that the number of APCs in iBAT was not significantly altered, while a decrease in the percentage of APCs was detected in iBAT, probably due to an increase in the percentage of endothelial cells (ECs) after ICV injection of T3 (Supplementary Fig. 2p). These results indicate that the central action of T3 might predominantly contribute to the regulation of thermogenesis in iBAT depot through increasing the UCP1 expression in mature brown adipocytes, it might not contribute to the T3-induced APC proliferation in iBAT, or it might not participate directly in the regulation of APC proliferation in iBAT by circulating T3 or local T3 generated in mature brown adipocytes via Dio2, which catalyzes the conversion of T4 to T3.

Furthermore, as suggested, we employed specific antagonist to block the function of adrenergic receptor (AR) in our study. The effect of i.p. injection of T3 on APC proliferation in iBAT could be observed after the treatment with either SR59230A (SR), a β 3-AR antagonist, or metoprolol (Met), a β 1-AR antagonist (Supplementary Fig. 2l,m). Decreases could also be detected in the number and percentage of APCs in iBAT of T3-treated MMI mice with either SR or Met treatment as compared to those in iBAT of T3-treated MMI mice without SR or Met treatment (Supplementary Fig. 2l,m). These results suggest that, although either the β 3 or β 1-AR-mediated sympathetic activation might be dispensable for the regulation of APC proliferation by T3, both of them might be capable of potentiating the T3 action and might be required for T3 to achieve its maximal effect on APC proliferation in iBAT. Given that β 1-AR and β 3-AR have differential expression pattern (Supporting data Fig. 2b) and distinct functions in adipocyte progenitors and mature adipocytes, respectively [25-27], these results imply that both β 1 and β 3 adrenergic signaling are involved in the regulation of APC proliferation by T3 but the underlying mechanisms differ. As T3 regulates APC proliferation in a cell-autonomous manner (Fig. 2l and Supplementary Fig. 2i) and either β 3 or β 1-AR-mediated sympathetic activation is not required for the regulation of APC proliferation by T3, we speculate that either β 1 or β 3 adrenergic signaling contributes to the profound T3 effect on APC proliferation but both of them are not indispensable in iBAT. Additionally, we also speculate that β 3 or β 1-AR-mediated sympathetic activation potentiate the T3 action and the β 1 or β 3 adrenergic signaling is required for T3 to achieve its maximal effect on APC proliferation in iBAT.

As the reviewer suggested, we also treated MTR α KO mice with SR59230A (SR). Consistent with above results and our findings in this study, the SR treatment could totally diminish the effect of i.p. injection of T3 on APC proliferation in the iBAT of MTR α KO mice, we speculated that β 3-AR mediated adrenergic activation is indispensable for the preserved T3 action in MTR α KO mice (Supplementary Fig. 3i). These results also indicate that β 3 adrenergic signaling involved here is independent of the TR α -mediated direct action of T3 on APC proliferation in the iBAT depot.

As suggested, we cited several key papers on central actions of T3 in our revised manuscript (Page 7, Line 231).

2. While of 2DG experiments are of interest, mechanistically do not add to much value. This kind of top study would be benefic of a most sophisticated approach, for example by genetic targeting of glycolysis. I would not ask about generating new mice strains, since that it would be out of the scope, but a virogenetic approach (targeting key glycolytic enzymes) would be easily to perform.

Response to the Reviewer's Comments: We thank for the reviewer's suggestion. To test whether glycolysis is critically involved in the regulation of APC by T3, we employed a lentivirus-mediated

shRNA knockdown approach developed recently to reduce the expression of Myc or Hk2 in the APCs of iBAT depot. We found that shRNA knockdown of either Myc or Hk2 could not only abolish the suppressive effect of T3 on the expression of Group 1 markers but also attenuate the upregulation of Group 2 marker gene expression by T3 in FACS-sorted APCs derived from iBAT (Fig. 9a,b). Furthermore, flow cytometry analysis revealed that knockdown of Myc and Hk2 by their specific shRNAs in the APCs of iBAT depot could totally diminish the T3 effect on either the percentage of Dpp4⁺ cells or the percentage of Icam1⁺ cells in these APCs (Fig. 9c). Similar results could be obtained in cultured iBAT-derived APCs (Supporting data Fig. 3a,b). These data further support the notion that T3 might promote the cell state transition from a more stem-like state towards a more committed adipogenic state via enhancing Myc-mediated glycolysis.

On the other hand, by performing *in vivo* BrdU labeling and flow cytometry analysis, we found that lentivirus-mediated shRNA knockdown of Myc and Hk2 in the APCs of iBAT could markedly attenuate the effect of T3 on the percentage of BrdU⁺ cells in these APCs (Fig. 9f,g). Similar results could be obtained in cultured cells derived from Dpp4⁺ APCs in iBAT (Supporting data Fig. 3c). Consistently, the T3 effect on the expression of Group 5 markers (Stmn1 and Cenpa) and Ccnb1 were attenuated by the knockdown of Myc and Hk2 in the APCs of iBAT of mice (Fig. 9h,i). Similar results could be obtained in cultured cells derived from Dpp4⁺ APCs in iBAT (Supporting data Fig. 3d). We then hypothesized that T3 might promote the cell cycle progression towards a mitotic state through a mechanism also involving Myc-mediated glycolysis, which actually is not surprising given that glycolysis generates intermediates for the generation of new biomass.

Together, based on above findings, we propose that T3 acts to facilitate both adipogenic and mitotic commitment in adipocyte progenitors via enhancing Myc-mediated glycolysis, hence promoting the hyperplastic growth of iBAT.

3. IRT pictures must be improved. FLIR cameras (as the one used) provide several color ways to display images, another option should be chosen. Details of the model of the camera, landmarks for the analysis (was it squared? as shown), temperature selected (average, maximal etc.) and emissivity (exact value must be specified) are needed.

Response to the Reviewer's Comments: The model of the FLIR infrared camera used previously in this study is E6 (160 x 120), which cannot achieve high resolution. To improve the quality of these IRT pictures, we repeated these experiments by using a high-resolution infrared camera from Magnity Technologies, which is the latest model (MAG 384 x 288) (Fig. 1b, 3d and Supplementary Fig. 1k). The details of the experiments are described briefly as below. Mice were anaesthetized using an i.p. injection of Isoflurane. The images were captured from the backs of the mice and were displayed with the rainbow high contrast color palette with a temperature linear display between 26°C and 38°C. Emissivity was 0.95. Indirect calorimetry measurements of squared hair-free interscapular area were exported to CSV files. Then surface temperature of iBAT is calculated as the average of the highest 10% area of the interscapular area as reported by Gregory R. Steinberg et.al [52].

4. Page 1 Introduction/line 81: "... (iBAT), a classic brown fat depot [13]..."

Response to the Reviewer's Comments: We thank the reviewer for pointing this out. We've changed "a classic BAT" to "a classic brown fat depot".

Notes for Supporting Data:

We apologized that we did not include all newly obtained data in our revised manuscript. We are willing to integrate the “Supporting data” for review into the supplementary figures upon request.

Supporting data Figure 1-3

Supporting data Figure 1.

(a) FACS plot of the CD90⁺CD140a⁺ cells in neck fat from human subjects. Cells were pre-gated on live CD45⁺CD31⁻ cells. (b) Representative H&E staining of iWAT from TRβFloxed and ATRβKO mice treated with T3 for 5 d. (c) Densitometry analysis of UCP1 protein in mature iBAT-SVF-derived adipocytes treated with cycloheximide (CHX) and T3 for 0, 6 and 12h as indicated (d) qRT-PCR analysis of Ckb in iBAT of MMI mice treated with T3 for 4 h or 5 d (n=5). (e) Western blot analysis and quantification (n=3) of CKB protein in iBAT of MMI mice with different T3 treatments. (f) qRT-PCR analysis of Serca2b in iBAT of MMI mice treated with T3 for 4 h or 5 d (n=5). (g-h) Feature plots and violin plots showing the expression levels and distribution of Ckb (g) and Atp2a2 (Serca2b) (h) in APCs of iBAT. (i-j) Comparison of the expression levels of Ckb (i) and Atp2a2 (Serca2b) (j) between MMI group and T3 group in APCs of iBAT.

Supporting data Figure 2.

(a) Luciferase activity in iBAT, iWAT, liver, and brain from THAI mice i.p. injected with T3 for 5 d (n=3). (b) Feature plots and dotted violin plots showing the levels of Adrb1 and Adrb3 in APCs of iBAT. (c) qRT-PCR analysis of Dio2 in cultured SVF cells from iBAT after adipogenic induction for indicated time (n=3). (d) Dotted violin plot showing the levels of Dio2 in APCs of iBAT. (e) Percentage of endothelial cells (ECs), hematopoietic cells and APCs in live SVF cells of iBAT from wild-type mice (WT mice) in this study (Liu et al.) and murine iBAT reported by Steinbring et al. (f) qRT-PCR analysis of Ccnb1 and Ccne1 in BAC cells transfected with Myc siRNA. Cells were treated with or without T3 for 24 h (n=6). (g and h) Number of SVF cells in iBAT and number of APCs, CD45⁺ cells and ECs (g) and percentage of APCs, CD45⁺ cells and ECs (h) in live SVF cells of iBAT digested with collagenase from Roche or Sigma (n=3). (i) Staining of PDGFR α and Hoechst with EdU labeling on iBAT sections from MMI mice with daily T3 injection (0.5X) for 5 d. (j) ¹⁸F-FDG uptake analysis of iBAT (left) and qRT-PCR analysis of Glut4 in the iBAT (right) of MMI mice treated with T3 for 4 h (n=3). (k) qRT-PCR analysis of Dpp4 and Icam1 in cultured Dpp4⁺ and Icam1⁺ APCs, respectively. Cycle threshold (Ct) values for Dpp4 and Icam1 are shown (n=3).

Supporting data Figure 3.

(a) Western blot analysis of Hk2 or Myc after its corresponding shRNA-mediated knockdown in cultured Dpp4⁺ cells. (b-d) qRT-PCR analysis of selected Group 1-specific genes (Dpp4 and Pi16) and Group 2-specific genes (Fabp4 and Plin2) (b), BrdU incorporation analysis (c) and qRT-PCR analysis of Group 5 markers (Stmn1 and Cenpa) and G2/M phases marker (Ccnb1) (d) in shRNA-treated Dpp4⁺ cells after minimal adipogenic induction along with T3 for 24 h (n=3).

Materials and Methods for Supporting data

Mice study

Mice with a targeted deletion of TR β in adipose tissues (ATR β KO mice) were generated by crossing the TR β Floxed mice (Shanghai Biomodel Organism Science & Technology Development) with Adiponectin-Cre mice (a thankful gift from Prof. Liu Yong at Wuhan University). PET-CT imaging was performed by Siemens Inveon PET-CT Multimodality System under room temperature. MMI mice were received a single T3 injection at a standard dose in advance. Three hours later, mice were lightly anesthetized using 3% isoflurane, and injected with approximately 150 μ Ci of ¹⁸F-FDG into the tail vein. After that, the animal was permitted to roam freely in the cage for one hour to uptake ¹⁸F-FDG. Subsequently, the animal was placed onto the imaging bed under 2% isoflurane anesthesia for the duration of imaging. Tissues of mice after PET/CT imaging were ex vivo measured with γ counter (SN-695 γ RIA Counter) and corrected with tissue weight.

Cell treatment and FACS analysis

For Myc or Hk2 knockdown *in vitro*, cultured Dpp4⁺ cells derived from APCs of iBAT were infected with shRNA lentivirus of LacZ, Myc or Hk2. Dpp4⁺ cells were maintained in minimal adipogenic induction medium and treated with T3 (100 nM) for 24 hours. Following antibodies were used for analysis of APCs in neck fat from human subjects: FITC anti-human CD31 (Invitrogen), APC/Cyanine7 anti-human CD45 (Biolegend), PE/Cyanine7 anti-human CD90 (Biolegend), PE anti-human CD140a (PDGFR α) (Biolegend). Collagenase Type 1 from Sigma and Roche (1mg/mL) were used to compare the efficiency for mouse iBAT digestion.

Analysis of mRNA and protein expression

For qRT-PCR analysis, primers for Ckb, Serca2b, Glut4 and Ccne1 are show below: mCkb-F, GCC TCA CTC AGA TCG AAA CTC; mCkb-R, GGC ATG TGA GGA TGT AGC CC; mSerca2b-F, ACC TTT GCC GCT CAT TTT CCA G; mSerca2b-R, AGG CTG CAC ACA CTC TTT ACC; mGlut4-F, GTG ACTG GAA CAC TGG TCC TA; mGlut4-R, CCA GCC ACG TTG CAT TGT AG; mCcne1-F, GTG GCT CCG ACC TTT CAG TC; mCcne1-R, CAC AGT CTT GTC AAT CTT GGC A. For western blot analysis, primary antibody against CKB (ABclonal) was used. The other materials and methods were described in the paper.

Reference:

1. Shinoda, K., et al., *Genetic and functional characterization of clonally derived adult human brown adipocytes*. Nat Med, 2015. **21**(4): p. 389-94.
2. Burl, R.B., et al., *Deconstructing Adipogenesis Induced by beta3-Adrenergic Receptor Activation with Single-Cell Expression Profiling*. Cell Metab, 2018. **28**(2): p. 300-309 e4.
3. Sun, W., et al., *snRNA-seq reveals a subpopulation of adipocytes that regulates thermogenesis*. Nature, 2020. **587**(7832): p. 98-102.
4. Shafer, M.E.R., *Cross-Species Analysis of Single-Cell Transcriptomic Data*. Frontiers in Cell and Developmental Biology, 2019. **7**.
5. Wang, X., et al., *Comparative analysis of cell lineage differentiation during hepatogenesis in humans and mice at the single-cell transcriptome level*. Cell Research, 2020. **30**(12): p. 1109-1126.
6. Oguri, Y., et al., *CD81 Controls Beige Fat Progenitor Cell Growth and Energy Balance via FAK Signaling*. Cell, 2020. **182**(3): p. 563-577 e20.
7. Katz, A.J., et al., *Cell surface and transcriptional characterization of human adipose-derived adherent stromal (hADAS) cells*. Stem Cells, 2005. **23**(3): p. 412-423.
8. Huang, S.J., et al., *Adipose-Derived Stem Cells: Isolation, Characterization, and Differentiation Potential*. Cell Transplantation, 2013. **22**(4): p. 701-709.
9. Zhu, Y.Z., et al., *Extracellular vesicles derived from human adipose-derived stem cells promote the exogenous angiogenesis of fat grafts via the let-7/AGO1/VEGF signalling pathway*. Scientific Reports, 2020. **10**(1).
10. An, Y.T., et al., *A Molecular Switch Regulating Cell Fate Choice between Muscle Progenitor Cells and Brown Adipocytes*. Developmental Cell, 2017. **41**(4): p. 382-+.
11. Johann, K., et al., *Thyroid-Hormone-Induced Browning of White Adipose Tissue Does Not Contribute to Thermogenesis and Glucose Consumption*. Cell Rep, 2019. **27**(11): p. 3385-3400 e3.
12. Lin, J.Z., et al., *Pharmacological Activation of Thyroid Hormone Receptors Elicits a Functional Conversion of White to Brown Fat*. Cell Rep, 2015. **13**(8): p. 1528-37.
13. Weiner, J., et al., *Thyroid hormones and browning of adipose tissue*. Mol Cell Endocrinol, 2017. **458**: p. 156-159.
14. Obregon, M.J., *Thyroid hormone and adipocyte differentiation*. Thyroid, 2008. **18**(2): p. 185-95.
15. Yeh, W.J., P. Leahy, and H.C. Freake, *Regulation of brown adipose tissue lipogenesis by thyroid hormone and the sympathetic nervous system*. Am J Physiol, 1993. **265**(2 Pt 1): p. E252-8.
16. Kalinovich, A.V., et al., *UCP1 in adipose tissues: two steps to full browning*. Biochimie, 2017. **134**: p. 127-137.
17. Nedergaard, J., Y. Wang, and B. Cannon, *Cell proliferation and apoptosis inhibition: essential processes for recruitment of the full thermogenic capacity of brown adipose tissue*. Biochim Biophys Acta Mol Cell Biol Lipids, 2019. **1864**(1): p. 51-58.
18. Rahbani, J.F., et al., *Creatine kinase B controls futile creatine cycling in thermogenic fat*. Nature, 2021. **590**(7846): p. 480-485.
19. Ikeda, K., et al., *UCP1-independent signaling involving SERCA2b-mediated calcium cycling regulates beige fat thermogenesis and systemic glucose homeostasis*. Nat Med,

2017. **23**(12): p. 1454-1465.
20. Mohacsik, P., et al., *A Transgenic Mouse Model for Detection of Tissue-Specific Thyroid Hormone Action*. *Endocrinology*, 2018. **159**(2): p. 1159-1171.
 21. Silva, J.E. and P.R. Larsen, *Potential of brown adipose tissue type II thyroxine 5'-deiodinase as a local and systemic source of triiodothyronine in rats*. *J Clin Invest*, 1985. **76**(6): p. 2296-305.
 22. Fernandez, J.A., et al., *Direct assessment of brown adipose tissue as a site of systemic triiodothyronine production in the rat*. *Biochem J*, 1987. **243**(1): p. 281-4.
 23. Rothwell, N.J., M.E. Saville, and M.J. Stock, *Sympathetic and thyroid influences on metabolic rate in fed, fasted, and refed rats*. *Am J Physiol*, 1982. **243**(3): p. R339-46.
 24. Yen, P.M., *Physiological and molecular basis of thyroid hormone action*. *Physiol Rev*, 2001. **81**(3): p. 1097-142.
 25. Bronnikov, G., et al., *beta1 to beta3 switch in control of cyclic adenosine monophosphate during brown adipocyte development explains distinct beta-adrenoceptor subtype mediation of proliferation and differentiation*. *Endocrinology*, 1999. **140**(9): p. 4185-97.
 26. Lee, Y.H., et al., *Cellular origins of cold-induced brown adipocytes in adult mice*. *FASEB J*, 2015. **29**(1): p. 286-99.
 27. Evans, B.A., et al., *Adrenoceptors in white, brown, and brite adipocytes*. *Br J Pharmacol*, 2019. **176**(14): p. 2416-2432.
 28. Jiao, Y., et al., *Ad36 promotes differentiation of hADSCs into brown adipocytes by up-regulating LncRNA ROR*. *Life Sciences*, 2021. **265**.
 29. Steinbring, J., et al., *Flow Cytometric Isolation and Differentiation of Adipogenic Progenitor Cells into Brown and Brite/Beige Adipocytes*. *Methods Mol Biol*, 2017. **1566**: p. 25-36.
 30. Barruet, E., et al., *Functionally heterogeneous human satellite cells identified by single cell RNA sequencing*. *Elife*, 2020. **9**.
 31. Xia, J., et al., *A single-cell resolution developmental atlas of hematopoietic stem and progenitor cell expansion in zebrafish*. *Proc Natl Acad Sci U S A*, 2021. **118**(14).
 32. Wanjalla, C.N., et al., *Single-cell analysis shows that adipose tissue of persons with both HIV and diabetes is enriched for clonal, cytotoxic, and CMV-specific CD4+ T cells*. *Cell Rep Med*, 2021. **2**(2): p. 100205.
 33. Hildreth, A.D., et al., *Single-cell sequencing of human white adipose tissue identifies new cell states in health and obesity*. *Nat Immunol*, 2021. **22**(5): p. 639-653.
 34. Schwalie, P.C., et al., *A stromal cell population that inhibits adipogenesis in mammalian fat depots*. *Nature*, 2018. **559**(7712): p. 103-108.
 35. Hepler, C., et al., *Identification of functionally distinct fibro-inflammatory and adipogenic stromal subpopulations in visceral adipose tissue of adult mice*. *Elife*, 2018. **7**.
 36. Fujimoto, K., et al., *Thyroid hormone activates protein arginine methyltransferase 1 expression by directly inducing c-Myc transcription during Xenopus intestinal stem cell development*. *J Biol Chem*, 2012. **287**(13): p. 10039-10050.
 37. Menssen, A. and H. Hermeking, *Characterization of the c-MYC-regulated transcriptome by SAGE: identification and analysis of c-MYC target genes*. *Proc Natl Acad Sci U S A*, 2002. **99**(9): p. 6274-9.
 38. Roh, H.C., et al., *Warming Induces Significant Reprogramming of Beige, but Not Brown,*

- Adipocyte Cellular Identity*. Cell Metab, 2018. **27**(5): p. 1121-1137 e5.
39. Rosenwald, M., et al., *Bi-directional interconversion of brite and white adipocytes*. Nature Cell Biology, 2013. **15**(6): p. 659-+.
 40. Roh, H.C., et al., *Simultaneous Transcriptional and Epigenomic Profiling from Specific Cell Types within Heterogeneous Tissues In Vivo*. Cell Reports, 2017. **18**(4): p. 1048-1061.
 41. Mirbolooki, M.R., et al., *Quantitative assessment of brown adipose tissue metabolic activity and volume using 18F-FDG PET/CT and β -adrenergic receptor activation*. EJNMMI Res, 2011. **1**(1): p. 30.
 42. Chen, Y.I., et al., *Anatomical and functional assessment of brown adipose tissue by magnetic resonance imaging*. Obesity (Silver Spring), 2012. **20**(7): p. 1519-26.
 43. Bonetti, P.O., et al., *Noninvasive identification of patients with early coronary atherosclerosis by assessment of digital reactive hyperemia*. J Am Coll Cardiol, 2004. **44**(11): p. 2137-41.
 44. Torrance, C.J., et al., *Effects of thyroid hormone on GLUT4 glucose transporter gene expression and NIDDM in rats*. Endocrinology, 1997. **138**(3): p. 1204-14.
 45. Lau, A.Z., et al., *Noninvasive identification and assessment of functional brown adipose tissue in rodents using hyperpolarized (1)(3)C imaging*. Int J Obes (Lond), 2014. **38**(1): p. 126-31.
 46. Whittle, A.J., et al., *BMP8B increases brown adipose tissue thermogenesis through both central and peripheral actions*. Cell, 2012. **149**(4): p. 871-85.
 47. Schulz, T.J., et al., *Brown-fat paucity due to impaired BMP signalling induces compensatory browning of white fat*. Nature, 2013. **495**(7441): p. 379-383.
 48. Martinez-Sanchez, N., et al., *Hypothalamic AMPK-ER Stress-JNK1 Axis Mediates the Central Actions of Thyroid Hormones on Energy Balance*. Cell Metabolism, 2017. **26**(1): p. 212-+.
 49. de Morentin, P.B.M., et al., *Estradiol Regulates Brown Adipose Tissue Thermogenesis via Hypothalamic AMPK*. Cell Metabolism, 2014. **20**(1): p. 41-53.
 50. Alvarez-Crespo, M., et al., *Essential role of UCP1 modulating the central effects of thyroid hormones on energy balance*. Molecular Metabolism, 2016. **5**(4): p. 271-282.
 51. Lopez, M., et al., *Hypothalamic AMPK and fatty acid metabolism mediate thyroid regulation of energy balance*. Nat Med, 2010. **16**(9): p. 1001-8.
 52. Crane, J.D., et al., *A standardized infrared imaging technique that specifically detects UCP1-mediated thermogenesis in vivo*. Mol Metab, 2014. **3**(4): p. 490-4.

REVIEWER COMMENTS

Reviewer #1 (Remarks to the Author):

The authors responded appropriately to almost all of the referee's comments, and one point I would like to add is that the analysis to look at the similarity of human deep-neck APC to mouse APCs with correlation coefficient (CC), which was newly added in this revision, should be explained in more detail. Were the CCs obtained by averaging the single cells within each sample across all? Since the results of this analysis are also included in Figure 9, more information about the methodology is needed.

Reviewer #2 (Remarks to the Author):

The authors have addressed many of the concerns raised in the initial review. However, several issues remaining, notably lack of certain experimental details, over reliance on speculative pseudotime analysis, and what appears to be internal inconsistencies between FACs and scRNA analyses.

Rolling microneedle procedure not described or referenced.

Several experiments should be evaluated by 2-way ANOVA (figure 3G, G, H I) to see if there is a significant interaction effect, as concluded.

It seems clear from the Umap, heat map and violin plots (fig 5) that the proliferating population is much more closely related to group 2 than any other group.

What are the relative numbers of group 5 cells in Figure 8a (the figure label obscures this group). Proliferating cells cluster away from others owing to the highly differential expression of cell cycle genes. Nonetheless the proliferating cell types can be identified by co-expression of informative ASC DEGs. It appears that cluster 5 is most closely related to cluster 2 and not any other. The scRNAseq data in figure 8 does not seem to support the claim in supple 8 that that 15 to 30% of clusters 1 and 2 are actively proliferating. What % of cluster 1 or 2 cells express proliferation makers and conversely what % of cluster 5 have evidence of differential ASC recruitment?

Pseudotime separates by gene expression and does not demonstrate lineage trajectories. The analysis is purely descriptive, speculative and lacks true in vivo validation. Please explain the meaning of the lines and how the lines are statistically evaluated. How much of the variance in gene expression is accounted for by the presumed trajectory?

The authors seem to be claiming that that the 5 ul Matrigel mixture was injected into the iBAT pad and that these cells were recovered from the total depot. Please provide more detail of the procedure. Are the authors claiming that the injected cells somehow incorporated in to the existing pad or did they recover the Matrigel "pad" that was injected in the region of the iBAT? At a minimum, should show low magnification images that encompass the transplanted material. What is shown has very little differentiation, none of which is quantified. While the data certainly suggest that Dpp4+ cells can give rise to Icam1+ cells under some conditions (and not the reverse), neither the "pseudotime analysis" nor the transplant experiments demonstrate that this occurs for endogenous cells.

The BODIPY staining in 7A (lipid) is clearly an artifact as it matches almost perfectly the tdTomato (protein and does not stain obvious lipid droplets. Any true staining that showed incorporation into the parenchyma would show massive staining of endogenous adipocytes.

Reviewer #3 (Remarks to the Author):

All my comments have been perfectly addressed. I have no further suggestions.

REVIEWER COMMENTS

Reviewer #1 (Remarks to the Author):

The authors responded appropriately to almost all of the referee's comments, and one point I would like to add is that the analysis to look at the similarity of human deep-neck APC to mouse APCs with correlation coefficient (CC), which was newly added in this revision, should be explained in more detail. Were the CCs obtained by averaging the single cells within each sample across all? Since the results of this analysis are also included in Figure 9, more information about the methodology is needed.

Response to the Reviewer's Comments: We thank the reviewer for the comments. Correlation coefficient of global expression signature was obtained based on averaged expression values for each scRNA-seq dataset analyzed by the *AverageExpression* function of Seurat R package, in which case each sample was regarded as pseudo-bulk RNA-seq data. As suggested, we provided more information about the methodology, including references, and explained it in more detail in Material and Method (Page 29, Line 1007-1017) and Result (Page 19, Line 646-653) sections, respectively.

Material and Method: To performed a cross-species comparison analysis, mouse genome was used as the reference gene list, and human genes were transformed into mouse orthologs by homologene R package. If human's "GENE1" had two reads, and "GENE2" three reads, the mouse ortholog "Gene" received a total five reads¹. Preadipocytes or APCs were then integrated and clustering by Seurat R package for combined analyses across species. 13 subpopulations were determined by appropriate PCs and resolution². Spearman rank correlation of global expression signature was calculated by average expression levels in each sample which was regarded as pseudo-bulk RNA-seq data¹. Among the 13 subpopulations, 11 were identified in both human and mouse datasets. Spearman rank correlation of average expression levels in each matched subpopulation in human and mouse was calculated³.

Results: We also used a recently developed method to determine the cross-species similarities based on scRNA-seq data, which requires cell types to be matched between species before correlations are calculated^{2,3}. We combined the four datasets to perform annotation and assigned 11 matched cell subpopulations between human and mouse. Spearman rank correlation of average expression levels in each matched subpopulation revealed a higher correlation between preadipocytes from human deep-neck fat and APCs from mouse iBAT (Supplementary Fig. 9i).

Reviewer #2 (Remarks to the Author):

The authors have addressed many of the concerns raised in the initial review. However, several issues remaining, notably lack of certain experimental details, over reliance on speculative pseudotime analysis, and what appears to be internal inconsistencies between FACs and scRNA analyses.

Response to the Reviewer's Comments: We thank the reviewer for the comments. We agree with the reviewer that the conclusions from speculative pseudotime analysis needs further experimental validation. Here, we tried our best to address the remaining issues raised by the reviewers as follows and revised our manuscript accordingly.

Rolling microneedle procedure not described or referenced.

Response to the Reviewer's Comments: We apologize for not providing the detailed information for this procedure. As the reviewer requested, we provided more detailed information in our revised manuscript (Page 24-25, Line 839-847). Indeed, a variety of techniques have been developed to deliver drugs or compounds into the dermis or subcutaneous layer, such as microneedle technique, which is non-surgical and less invasive. Different kinds of medical needles, including multi-needle (32/34G), microneedle, and prick needle, have been tested in our lab. In this study, as the procedure needed to be repeated daily for 5 days, to minimize the pain, a derma rolling system (Derma Roller) with fine needles from MEVOS Medical Technology, Suzhou, China, which has been used for cellulite reduction, skin rejuvenation, and wrinkle removal, was applied for the transdermal delivery of T3 to the iBAT of mice. This model can be connected to syringe and the length of microneedles used in this study is 1.5 mm in order to achieve a transdermal delivery. After the interscapular region of anaesthetized mice was shaved, the skin was punctured with the roller. 60 μ l of T3-containing saline (0.125X) was then added during the rolling process either via a syringe or by manual pipetting. We confirmed that the transdermal delivery was successful, as it could be visualized by using a blue colored solution (ink diluted with saline) or using GFP virus (Supporting data Fig. 4a,b). We found that appreciable amounts of colored solution penetrated across intact murine skin and diffuse into interscapular region (Supporting data Fig. 4a). We also detected positive GFP signals in iBAT after infection (Supporting data Fig. 4b). As the data from THAI mice indicate that the reporter activities were increased in iBAT but not in other tissue after T3 treatment (Supplementary Fig. 2q), we speculated that the delivery is relatively specific. Nevertheless, we could not totally rule out the possibility that the delivery of T3 to other tissues might also have regulatory effects, confounding the interpretations of results.

Several experiments should be evaluated by 2-way ANOVA (figure 3G, G, H I) to see if there is a significant interaction effect, as concluded.

Response to the Reviewer's Comments: As the reviewer requested, we evaluated the data in Figure 3G, H, I by two-way ANOVA. A significant genotype-by-treatment interaction concerning the cell number and Ki67 staining of APCs was observed in Panel 3g ($F_{1,18}=4.526$, $P=0.0475$) and 3i ($F_{1,18}=4.920$, $P=0.0397$). A trend for a genotype-by-treatment interaction concerning the percentage of APCs was found in Panel 3h ($F_{1,18}=4.397$, $P=0.0504$), although it did not reach statistical significance. The two-way ANOVA analysis revealed that the treatment-related differences in APC proliferation depended on the genotype, further supporting our notion that loss of TR α could attenuate the T3 effects on APC proliferation. We provided the results of analysis in the legend in our revised manuscript.

It seems clear from the Umap, heat map and violin plots (fig 5) that the proliferating population is much more closely related to group 2 than any other group. What are the relative numbers of group 5 cells in Figure 8a (the figure label obscures this group). Proliferating cells cluster away from others owing to the highly differential expression of cell cycle genes. Nonetheless the proliferating cell types can be identified by co-expression of informative ASC DEGs. It appears that cluster 5 is most closely related to cluster 2 and not any other. The scRNAseq data in figure 8 does not seem to support the claim in supple 8 that that 15 to 30% of clusters 1 and 2 are actively proliferating. What % of cluster 1 or 2 cells express proliferation makers and conversely what % of cluster 5 have evidence of differential ASC recruitment?

Response to the Reviewer's Comments: We agree with the reviewer that the Group 5 is much more closely related to Group 2 than any other groups. We prefer to call Group 5 cells “dividing progenitors” or “dividing population” rather than “proliferating population” in this study. As we mentioned in our manuscript, mitosis is a process of cell division used for the organic growth of tissues, and cells spend most of their time in interphase (G1, S and G2 phases) and cycle through interphase and then M phase to proliferate. As the number of APCs can increase and some of the APCs could be stained with BrdU under certain conditions, we would like to call them proliferative APCs, which means APCs are capable of or engaged in proliferation. At a certain timepoint, some APCs are proliferating and stay in different phases of cell cycle. As Group 5 cells express cell cycle markers (cell cycle or mitosis-related genes) (Fig. 5k-m, Supplementary Fig. 5c, and Supplementary Table 2,3), particularly G2M phase marker *Ccnb1*, we speculate that Group 5 cells are cells currently undergoing cell division. Therefore, we think that using “dividing” is better than using “proliferating” to describe this group of cells. Given that Group 5 is much more closely related to Group 2 than any other groups, we speculate that Group 2 cells are more likely to accumulate adequate carbohydrates, nucleotides, amino acids, and fatty acids and meet the conditions to entering M phase.

The number of group 5 cells in Figure 8a is 12 for MMI group and 45 for T3 group. We relabeled the Group 5 in our revised manuscript to avoid obscuration. As Group 5 cells are only referred to as cells currently undergoing cell division, therefore, only a very few APCs belong to Group 5 (~0.45%). BrdU incorporation assay showed that the percentage of newly generated Group 1 and 2 cells was about 2% and 6% in MMI group and about 16% and 30% in T3 group, respectively. I would like to point out that, as mice were given 4 i.p. injections of BrdU every 6 hours and then sacrificed 2 hours after later BrdU injection, BrdU⁺ cells represented all newly generated cells in 20 hours during the period of BrdU treatment. Therefore, it is not surprising that the percentage of BrdU⁺ cells was higher than the percentage of Group 5 cells.

We agree with the reviewer that dividing cells cluster away from others owing to the highly differential expression of cell cycle genes and the dividing cell types can be identified by co-expression of informative ASC DEGs. As suggested, we determined the expression of G2M phase markers in Group 1 or 2 cells and the expression of adipogenic markers in Group 5 cells. As Seurat R package contains a list for cell cycle phase marker genes (54 genes for G2M phase and 43 genes for S phase), we used this list and *CellCycleScoring* function to assign each cell a score, based on its expression of G2M and S phase markers (Fig. 5i). If $G2M.score > 0$ and $G2M.score > S.score$, the cell was considered to be in G2M phase. If $S.score > 0$ and $S.score > G2M.score$, the cell was considered to be in S phase. If $S.score < 0$ and $G2M.score < 0$, the cell was considered to be in G0/G1 phase. Here, we found that 98.25% Group 5 cells were enriched for G2M phase markers, while 8.43% Group 1 cells and 6.30% of Group 2 cells expressed G2M phase markers, respectively (Fig. 5i and Supporting data Fig. 4c). Notably, S phase markers were enriched in 44.36% Group 1 cells and 35.68% Group 2 cells, respectively (Fig. 5i and Supporting data Fig. 4c). These results further support our notion that Group 1 and Group 2 cells were proliferative APCs staying in different cell cycle phases. On the other hand, as expected, we found that adipogenic markers were expressed in 14-52% Group 1 cells and 62-93 % Group 2 cells, respectively (Supporting data Fig. 4d). To be noted, adipogenic markers were found to be expressed in 59-85% Group 5 cells (Supporting data Fig. 4d), suggesting that Group 5 cells belong to APCs and exhibit an adipogenic potential.

Pseudotime separates by gene expression and does not demonstrate lineage trajectories. The analysis is purely descriptive, speculative and lacks true *in vivo* validation. Please explain the meaning of the lines and how the lines are statistically evaluated. How much of the variance in gene expression is accounted for by the presumed trajectory?

Response to the Reviewer's Comments: We agree with the reviewer that the above-mentioned analysis is speculative and requires true *in vivo* validation. Our *in vivo* data from cell transplantation studies shown in Fig. 7 and Supplementary Fig. 7 support the conclusion based on pseudotemporal trajectory analysis that Dpp4⁺ progenitors (Group 1 cells) are more stem-like progenitors and Dpp4⁺ progenitors can adopt an adipogenic fate, giving rise to more committed Icam1⁺ progenitors (Group 2 cells), on the other hand, Group 2 cells unlikely serve as primary progenitors for Group 1. However, we could not employ this approach to test whether Group 1 progenitors can adopt Fate A and retain stem-like properties as suggested in Fig. 6g,h in our previously submitted manuscript. We also could not test whether Group 1 progenitors can give rise to the Group 3 cells due to the lack of surface markers for Group 3 cells (Fig. 6g,h and Supplementary Fig. 6a-c in our previously submitted manuscript). Nevertheless, as the results from pseudotemporal trajectory analysis are informative and may guide future studies, therefore, we removed the Fig. 6g,h from Figure 6 and provided them as supplementary figures in our revised manuscript. Regarding the meaning of lines, how the lines are evaluated, and how much of the variance is accounted for by the presumed trajectory, we provided more details of this analysis in our revise manuscript (Page 29, Line 993-997 and Page 31, Line 1067-1069).

Briefly, split kinetic curves in Fig.7h were plotted by *plot_genes_branched_pseudotime* function from the monocle packages. In order to make the bifurcation plot, the function first duplicated the progenitor states and by default stretch each branch into maturation level 0-100, and then fit two nature spline curves for each branch. Logarithm was used to normalize the expression of genes. The lines represent kinetic trends of gene expression from the root of trajectory to Cell Fate A (solid line) or to Cell Fate B (dashed line)⁴. Regarding statistical analysis, we used Branch expression analysis modeling (BEAM), a statistical approach for finding genes that are regulated in a manner that depends on the branch, to evaluate gene divergence between two fitted spline curves⁴. The p-value calculated by a likelihood ratio test reflected all the genes shown in figures (previously submitted as Fig. 6h, now provided as Supplementary Fig. 6d) differ between the two cell fates. The trends of these representative DEGs of each group along the pseudotime trajectory and their differences between cell fates indicate the evolution of each group to some extent. Thus, we hypothesized that Group 1 progenitors can adopt two fate trajectories along the pseudotime, giving rise to either the more committed Group 2 progenitors or the Group 3 cells.

The authors seem to be claiming that that the 5 ul Matrigel mixture was injected into the iBAT pad and that these cells were recovered from the total depot. Please provide more detail of the procedure. Are the authors claiming that the injected cells somehow incorporated in to the existing pad or did they recover the Matrigel “pad” that was injected in the region of the iBAT? At a minimum, should show low magnification images that encompass the transplanted material. What is shown has very little differentiation, none of which is quantified. While the data certainly suggest that Dpp4⁺ cells can give rise to Icam1⁺ cells under some conditions (and not the reverse), neither the “pseudotime analysis” nor the transplant experiments demonstrate that this occurs for endogenous cells.

Response to the Reviewer's Comments: We performed the transplantation according the methods published by Merrick et al. in 2019⁵. As suggested, we provided more detail of the procedure in our revised manuscript (Page 25, Line 850-859). Briefly, for cell transplantation, APCs of iBAT from Rosa26-tdTomato reporter mice with Myf5-Cre were sorted by FACS as donor cells. 5 or 10×10⁵ tdTomato⁺ APCs as indicted in the figure legend were mixed 1:1 with Matrigel on ice. C57BL/6 mice aged 21-28 days were anesthetized with isoflurane flow and back hair was shaved. A small cutaneous incision was made to expose interscapular region, and 5 µl of mixture was injected into iBAT depot (normally to one lobe of the iBAT) by using a microsyringe (Shanghai Gaoge Industry & Trade Co., LTD). The brown adipose pad containing the implants was harvested for histology procedures or FACS at the indicated timepoint as mentioned in the figure legend. Regarding the issue of differentiation, we did observe the formation of lipid droplet in transplanted cells, suggesting that these cells are capable to differentiate *in vivo*. However, as the lipid droplets observed after transplantation were normally small (Supplementary Fig. 7a,c), we speculate that transplanted APCs were not fully differentiated under normal experimental condition and potent adipogenic stimuli might be needed to potentiate the *in vivo* differentiation. To enhance the differentiation efficiency of transplanted APCs, one week after transplantation, mice were exposed to cold stress (4°C) for 3 weeks. We found that, in the presence of the potent adipogenic stimulus, fully differentiated cells with large lipid droplets could be observed (Fig. 7a and Supplementary Fig. 7d,e). These data indicate that Group 1 and 2 cells can differentiate into matured adipocyte *in vivo* with potent adipogenic stimuli. We provided these new data in our revised manuscript.

Normally, injected Matrigel pad could not be distinguished from the iBAT when we collected it for FACS or histology analysis. As the donor cells all express tdTomato and can be sorted out easily by FACS and the iBAT is relatively small, we recovered one lobe of the iBAT containing the injected Matrigel pad for further analysis. Here, we injected the 5 µl of Matrigel containing trypan blue as an indicator into one lobe of the iBAT as described above and provided a figure to illustrate how we collected the samples for FACS and histology analysis (Supporting data Fig. 4e). As the reviewer suggested, we provided low magnification images that encompass the transplanted material (Supplementary Fig. 7d,e and Supporting data Fig. 4f).

The BODIPY staining in 7A (lipid) is clearly an artifact as it matches almost perfectly the tdTomato (protein and does not stain obvious lipid droplets. Any true staining that showed incorporation into the parenchyma would show massive staining of endogenous adipocytes.

Response to the Reviewer's Comments: We thank the reviewer very much for pointing out this mistake. We repeated the experiment and replaced these problematic data in Fig. 7a with new data (Supplementary Fig. 7c). Moreover, as mentioned above, to enhance the *in vivo* differentiation efficiency, we treated mice with potent adipogenic stimuli (cold exposure) and observed fully differentiated adipocytes stained with BODIPY and tdTomato antibody *in vivo* (Fig. 7a and Supplementary Fig. 7d,e), further supporting out notion that both sort-purified Dpp4⁺ and Icam1⁺ APCs can give rise to mature adipocytes *in vivo*.

Reviewer #3 (Remarks to the Author):

All my comments have been perfectly addressed. I have no further suggestions.

Response to the Reviewer's Comments: We thank the reviewer for helping us to improve our

manuscript.

Reference

1. Geirsdottir L, *et al.* Cross-Species Single-Cell Analysis Reveals Divergence of the Primate Microglia Program. *Cell* **179**, 1609-1622 e1616 (2019).
2. Shafer MER. Cross-Species Analysis of Single-Cell Transcriptomic Data. *Front Cell Dev Biol* **7**, 175 (2019).
3. Wang X, *et al.* Comparative analysis of cell lineage differentiation during hepatogenesis in humans and mice at the single-cell transcriptome level. *Cell Res* **30**, 1109-1126 (2020).
4. Qiu X, Hill A, Packer J, Lin D, Ma YA, Trapnell C. Single-cell mRNA quantification and differential analysis with Census. *Nat Methods* **14**, 309-315 (2017).
5. Merrick D, *et al.* Identification of a mesenchymal progenitor cell hierarchy in adipose tissue. *Science* **364**, (2019).

Supporting data Figure 4

Supporting data Figure 4.

(a and b) A blue colored solution (ink diluted with saline) (a) and GFP virus (b) were used to confirm the transdermal delivery using the Derma Roller. (c) The percentage of cells expressing G2M or S phase markers in Group 1, 2, and 5 as indicated. (d) The percentage of cells expressing adipogenic markers (Pparg, Fabp4, Plin2, and Cebpa) in Group 1, 2, and 5 as indicated. (e) An illustration showing the procedure of sample collection after cell transplantation. 5 μ l of Matrigel containing trypan blue as an indicator was injected into one lobe of the iBAT. The entire lobe containing the Matrigel pad was collected for subsequent FACS and histology analysis as indicated. White dashed line indicates the boundary between Matrigel pad and iBAT pad. (f) Immunofluorescence staining of tdTomato⁺ APC-derived implants with BODIPY (green) and tdTomato (red). One week after transplantation, mice were exposed to cold environment for 3 weeks. White dashed line indicates the boundary between implants and iBAT.

Materials and Methods for Supporting data

Imaging

Images in Supporting data Fig. 4b were acquired using ZEISS Stereo Discovery.V20. Image in Supporting data Fig. 4f was acquired using Revolve Fluorescence Microscope by ECHO.

REVIEWERS' COMMENTS

Reviewer #1 (Remarks to the Author):

All my comments have been addressed. I have no further suggestions.

Reviewer #2 (Remarks to the Author):

Analysis of variance is performed to partition the variance for the reader, not to report the (marginal) significance of an interaction. The meaning of *** is not stated and nor how it is to be interpreted in the light of the marginally significant interaction in the ANOVA. Please report in the supplemental the ANOVA table that states the percentage of the variance accounted for by the main effects and interaction.

REVIEWER COMMENTS

Reviewer #1 (Remarks to the Author):

All my comments have been addressed. I have no further suggestions.

Response to the Reviewer's Comments: We thank the reviewer for helping us to improve our manuscript.

Reviewer #2 (Remarks to the Author):

Analysis of variance is performed to partition the variance for the reader, not to report the (marginal) significance of an interaction. The meaning of *** is not stated and nor how it is to be interpreted in the light of the marginally significant interaction in the ANOVA. Please report in the supplemental the ANOVA table that states the percentage of the variance accounted for by the main effects and interaction.

Response to the Reviewer's Comments: Thanks for the comments. To comply with the journal policies, formatting requirements, and editorial requests, we have provided the exact p value for those bar charts in our figures and provided the meaning of p in the corresponding legends in our revised manuscript. As the reviewer suggested, ANOVA tables that state the percentage of the variance accounted for by the main effects and interaction for the data in Fig. 3g-i were provided in Supplementary Table 1-3 in our revised manuscript.